# The selection landscape and genetic legacy of ancient Eurasians

Evan K. Irving-Pease[1,25]✉, Alba Refoyo-Martínez[1,25], William Barrie[2,25], Andrés Ingason[1,3,25], Alice Pearson[4,5,25], Anders Fischer[1,6,7,25], Karl-Göran Sjögren[6], Alma S. Halgren[8], Ruairidh Macleod[2,9], Fabrice Demeter[1,10], Rasmus A. Henriksen[1], Tharsika Vimala[1], Hugh McColl[1], Andrew H. Vaughn[11], Leo Speidel[9,12], Aaron J. Stern[11], Gabriele Scorrano[1], Abigail Ramsøe[1], Andrew J. Schork[3,13], Anders Rosengren[1,3], Lei Zhao[1], Kristian Kristiansen[1,6], Astrid K. N. Iversen[14,15], Lars Fugger[14,16,17], Peter H. Sudmant[8,11], Daniel J. Lawson[18,26], Richard Durbin[4,19,26], Thorfinn Korneliussen[1,26], Thomas Werge[1,20,21,26], Morten E. Allentoft[1,22,26], Martin Sikora[1,26], Rasmus Nielsen[1,23,26]✉, Fernando Racimo[1,26]✉ & Eske Willerslev[1,2,24,26]✉

The Holocene (beginning around 12,000 years ago) encompassed some of the most significant changes in human evolution, with far-reaching consequences for the dietary, physical and mental health of present-day populations. Using a dataset of more than 1,600 imputed ancient genomes[1], we modelled the selection landscape during the transition from hunting and gathering, to farming and pastoralism across West Eurasia. We identify key selection signals related to metabolism, including that selection at the FADS cluster began earlier than previously reported and that selection near the *LCT* locus predates the emergence of the lactase persistence allele by thousands of years. We also find strong selection in the HLA region, possibly due to increased exposure to pathogens during the Bronze Age. Using ancient individuals to infer local ancestry tracts in over 400,000 samples from the UK Biobank, we identify widespread differences in the distribution of Mesolithic, Neolithic and Bronze Age ancestries across Eurasia. By calculating ancestry-specific polygenic risk scores, we show that height differences between Northern and Southern Europe are associated with differential Steppe ancestry, rather than selection, and that risk alleles for mood-related phenotypes are enriched for Neolithic farmer ancestry, whereas risk alleles for diabetes and Alzheimer's disease are enriched for Western hunter-gatherer ancestry. Our results indicate that ancient selection and migration were large contributors to the distribution of phenotypic diversity in present-day Europeans.

One of the central goals of human evolutionary genetics is to understand how natural selection has shaped the genomes of present-day people in response to changes in culture and environment. The transition from hunter-gatherers to farmers, and subsequently pastoralists, during the Holocene in Eurasia, involved some of the most dramatic changes in diet, health and social organization experienced during recent human evolution. These changes represent big shifts in environmental exposure, impacting the evolutionary forces acting on the human gene pool and imposing a series of heterogeneous selection pressures. As human lifestyles changed, close contact with domestic animals and higher population densities are likely to have increased exposure to infectious diseases, introducing new challenges to our immune system[2,3].

Our understanding of the genetic architecture of complex traits in humans has been substantially advanced by genome-wide association studies (GWAS), which have identified many genetic variants associated with phenotypes of interest[4,5]. However, the extent to which

[1]Lundbeck Foundation GeoGenetics Centre, Globe Institute, University of Copenhagen, Copenhagen, Denmark. [2]GeoGenetics Group, Department of Zoology, University of Cambridge, Cambridge, UK. [3]Institute of Biological Psychiatry, Mental Health Services, Copenhagen University Hospital, Roskilde, Denmark. [4]Department of Genetics, University of Cambridge, Cambridge, UK. [5]Department of Zoology, University of Cambridge, Cambridge, UK. [6]Department of Historical Studies, University of Gothenburg, Gothenburg, Sweden. [7]Sealand Archaeology, Kalundborg, Denmark. [8]Department of Integrative Biology, University of California Berkeley, Berkeley, CA, USA. [9]UCL Genetics Institute, University College London, London, UK. [10]Eco-anthropologie, Muséum national d'Histoire naturelle, CNRS, Université Paris Cité, Musée de l'Homme, Paris, France. [11]Center for Computational Biology, University of California, Berkeley, CA, USA. [12]Ancient Genomics Laboratory, The Francis Crick Institute, London, UK. [13]Neurogenomics Division, The Translational Genomics Research Institute (TGEN), Phoenix, AZ, USA. [14]Oxford Centre for Neuroinflammation, Nuffield Department of Clinical Neurosciences, John Radcliffe Hospital, University of Oxford, Oxford, UK. [15]Nuffield Department of Clinical Neurosciences, John Radcliffe Hospital, University of Oxford, Oxford, UK. [16]Department of Clinical Medicine, Aarhus University Hospital, Aarhus, Denmark. [17]MRC Human Immunology Unit, John Radcliffe Hospital, University of Oxford, Oxford, UK. [18]Institute of Statistical Sciences, School of Mathematics, University of Bristol, Bristol, UK. [19]Wellcome Sanger Institute, Cambridge, UK. [20]Department of Clinical Medicine, University of Copenhagen, Copenhagen, Denmark. [21]Institute of Biological Psychiatry, Mental Health Center Sct Hans, Copenhagen University Hospital, Copenhagen, Denmark. [22]Trace and Environmental DNA (TrEnD) Laboratory, School of Molecular and Life Science, Curtin University, Perth, Western Australia, Australia. [23]Departments of Integrative Biology and Statistics, UC Berkeley, Berkeley, CA, USA. [24]MARUM Center for Marine Environmental Sciences and Faculty of Geosciences, University of Bremen, Bremen, Germany. [25]These authors contributed equally: Evan K. Irving-Pease, Alba Refoyo-Martínez, William Barrie, Andrés Ingason, Alice Pearson, Anders Fischer. [26]These authors jointly supervised this work: Daniel J. Lawson, Richard Durbin, Thorfinn Korneliussen, Thomas Werge, Morten E. Allentoft, Martin Sikora, Rasmus Nielsen, Fernando Racimo, Eske Willerslev. ✉e-mail: evan.irvingpease@gmail.com; rasmus_nielsen@berkeley.edu; fracimo@sund.ku.dk; ew482@cam.ac.uk

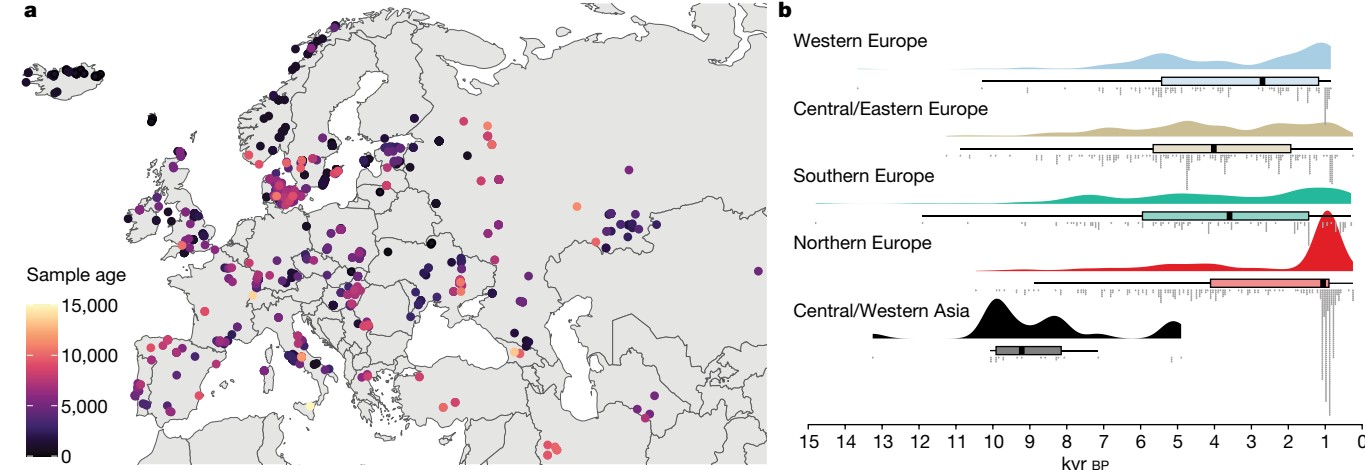

**Fig. 1 | Geographic and temporal distribution of the 1,015 ancient genomes from West Eurasia. a**, Map of West Eurasia showing sampling locations and ages of the ancient samples. **b**, Raincloud plot of the sample ages, grouped by sampling region: Western Europe (*n* = 156), Central/Eastern Europe (*n* = 268), Southern Europe (*n* = 136), Northern Europe (*n* = 432) and Central/Western Asia (*n* = 23). Boxplot shows the median and first and third quartiles of the sample ages and whiskers extend to the largest value no further than 1.5× the interquartile range.

these variants have been under directional selection during recent human evolution remains unclear. Although signatures of selection can be identified from patterns of genetic diversity in extant populations[6], this can be challenging in humans as they have been exposed to highly diverse and dynamic local environments through time and space. In the complex mosaic of genetic affinities that constitute a present-day human genome, any putative signatures of selection may misrepresent the timing and magnitude of the selective process. For example, episodes of admixture between ancestral populations can result in present-day haplotypes that contain no evidence of selective processes occurring further back in time. Ancient DNA (aDNA) provides the potential to resolve these issues, by directly observing changes in trait-associated allele frequencies over time.

Whilst numerous previous studies have used aDNA to infer patterns of selection in Eurasia during the Holocene (for example, refs. 7–9), many key questions remain unanswered. To what extent are present-day genetic differences due to natural selection or to differential patterns of admixture? What are the genetic legacies of Mesolithic, Neolithic and Bronze Age populations in present-day complex traits? How has the complex admixture history of Holocene Eurasia affected our ability to detect natural selection in genetic data? To investigate these questions, we tested for traces of divergent selection in health- and lifestyle-related genetic variants using three broad approaches. First, we looked for evidence of selection by identifying strong differentiation in allele frequencies between ancient populations. Second, we reconstructed the allele frequency trajectories and selection coefficients of tens of thousands of trait-associated variants, using a new chromosome painting technique to model ancestry-specific allele frequency trajectories through time. This allowed us to identify many trait-associated variants with new evidence for directional selection and to answer long-standing questions about the timing of selection for key health-, dietary- and pigmentation-associated loci. Finally, we used ancient genomes to infer local ancestry tracts in more than 400,000 present-day samples from the UK Biobank (UKB)[5] and calculated ancestry-specific polygenic risk scores for 35 complex traits. This allowed us to characterize the genetic legacy of Mesolithic, Neolithic and Bronze Age populations in present-day phenotypes.

## Samples and data

Our analyses are undertaken on a large collection of shotgun-sequenced ancient genomes presented in ref. 1. This dataset comprises 1,664 imputed diploid ancient genomes and more than 8.5 million single nucleotide polymorphisms (SNPs), with an estimated imputation error rate of 1.9% and a phasing switch error rate of 2.0% for 1X genomes. Full details of the validation and benchmarking of the imputation and phasing of this dataset are provided in ref. 10. These samples represent a considerable transect of Eurasia, ranging longitudinally from the Atlantic coast to Lake Baikal and latitudinally from Scandinavia to the Middle East (Fig. 1). The included genomes constitute a thorough temporal sequence from 11,000 to 1,000 cal BP. This dataset allowed us to characterize in fine detail the changes in selective pressures exerted by major transitions in human culture and environment.

## Genetic legacy of ancient Eurasians

We began our analysis by inferring local ancestry tracts in present-day populations by chromosome 'painting'[11] the UKB with Mesolithic, Neolithic and Bronze Age individuals as tract sources. We used a pipeline adapted from GLOBETROTTER[12] and estimated admixture proportions by means of non-negative least squares (Supplementary Note 2). In total, we painted 433,395 present-day samples, including 24,511 from individuals born outside the United Kingdom, from 126 countries (Supplementary Note 1). Our results show that none of the Mesolithic, Neolithic or Bronze Age ancestries are homogeneously distributed among present-day Eurasian populations (Fig. 2). Western hunter-gatherer (WHG)-related ancestries are highest in present-day individuals from the Baltic States, Belarus, Poland and Russia; Eastern hunter-gatherer (EHG)-related ancestries are highest in Mongolia, Finland, Estonia and Central Asia; and Caucasus hunter-gatherer (CHG)-related ancestries are highest in countries east of the Caucasus, in Pakistan, India, Afghanistan and Iran, in accordance with previous results[13]. The CHG-related ancestries probably reflect affinities to both CHG and Iranian Neolithic individuals, explaining the relatively high levels in South Asia[14]. Consistent with expectations[15], Neolithic Anatolian-related farmer ancestries are concentrated around the Mediterranean basin, with high levels in southern Europe, the Near East and North Africa, including the Horn of Africa, but are less frequent in Northern Europe. This is in direct contrast to the Steppe-related ancestries, which are found in high levels in northern Europe, peaking in Ireland, Iceland, Norway and Sweden and decreasing further south. There is also evidence for their spread into southern Asia. Overall, these results refine global patterns of spatial

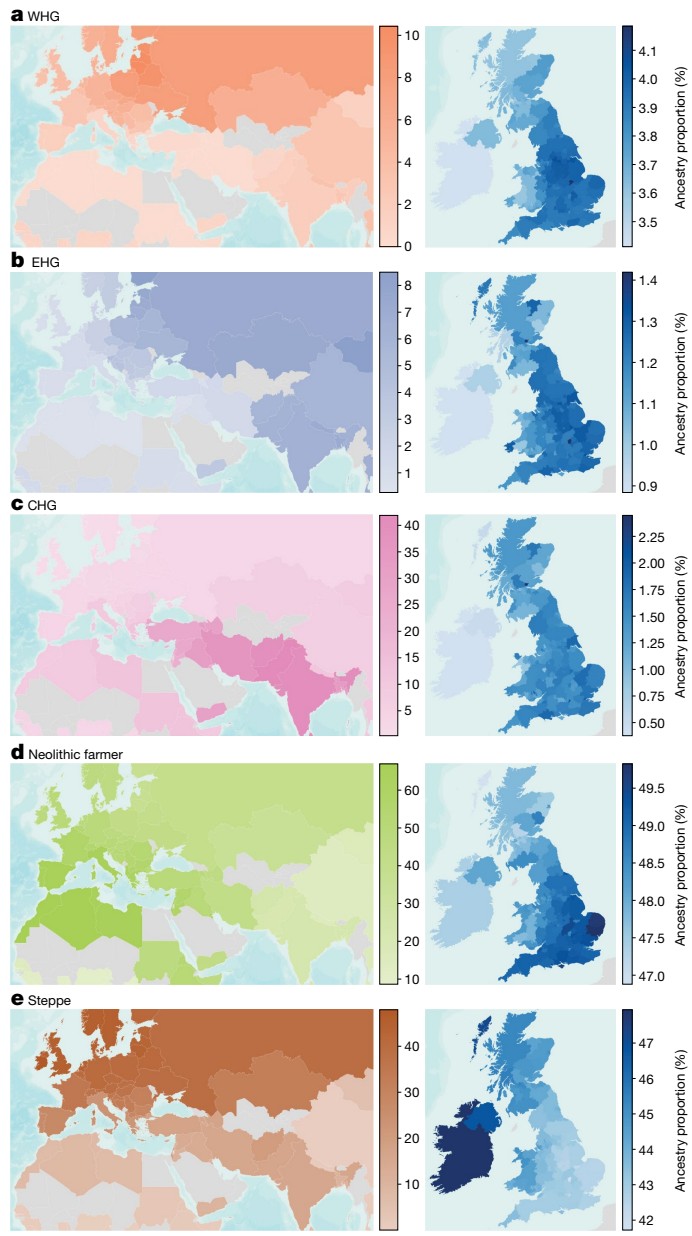

**Fig. 2 | The genetic legacy of ancient Eurasian ancestries in present-day populations. a–e**, Maps showing the average ancestry of: WHG (**a**); EHG (**b**); CHG (**c**); Neolithic farmer (**d**); and Steppe pastoralist (**e**) ancestry components per country (left) and per county or unitary authority within Great Britain and per country for the Republic of Ireland and Northern Ireland (right). Estimation was performed using ChromoPainter and NNLS, on samples of a 'typical ancestral background' for each non-UK country (*n* = 24,511) and Northern Ireland. For Great Britain, an average of self-identified 'white British' samples was used to represent each UK county and unitary authority, based on place of birth (*n* = 408,884). Countries with less than 4 and counties with less than 15 samples are shown in grey. Map uses ArcGIS layers World Countries Generalized and World Terrain.

distributions of ancient ancestries amongst present-day individuals. We caution, however, that absolute admixture proportions should be interpreted with caution in regions where our ancient source populations are less directly related to present-day individuals, such as in Africa and East Asia. Although these values are dependent on the reference samples used, as well as the treatment of pre- or post-admixture drift, the relative geographical variation and associations should remain consistent.

The availability of many present-day samples (*n* = 408,884) from self-identified 'white British' individuals who share similar positions on a principal component analysis[5] allowed us to further examine the distribution of ancient ancestries at high resolution in present-day Britain (Supplementary Note 2). Although regional ancestry distributions differ by only a few percentage points, we find clear evidence of geographical heterogeneity across the United Kingdom. This can be visualized by averaging ancestry proportions per county on the basis of place of birth (Fig. 2). The proportion of Neolithic farmer ancestries is highest in southern and eastern England today and lower in Scotland, Wales and Cornwall. Steppe-related ancestries are inversely distributed, peaking in the Outer Hebrides and Ireland, a pattern only previously described for Scotland[16]. This regional pattern was already evident in the Pre-Roman Iron Age and persists to the present day even though immigrating Anglo-Saxons had relatively less affinities to Neolithic farmers than the Iron Age individuals of southwest Britain. Although this Neolithic farmer/Steppe-related dichotomy mirrors the modern 'Anglo-Saxon'/'Celtic' ethnic divide, its origins are older, resulting from continuous migration from a continental population relatively enriched in Neolithic farmer ancestries, starting as early as the Late Bronze Age[17,18]. By measuring haplotypes from these ancestries in present-day individuals, we show that these patterns differentiate Wales and Cornwall as well as Scotland from England. We also find higher levels of WHG-related ancestries in central and northern England. These results demonstrate clear ancestry differences within an 'ethnic group' (white British), highlighting the need to account for subtle population structure when using resources such as the UKB[19].

## Ancestry-stratified selective sweeps

Having identified that significant differences in ancestries persist in seemingly homogeneous present-day populations, we sought to disentangle these effects by developing a chromosome painting technique that allows us to label haplotypes on the basis of their genetic affinities to ancient individuals. To achieve this, we built a quantitative admixture graph model (Fig. 3 and Supplementary Note 3) that represents the four main ancestry flows contributing to present-day European genomes over the last 50,000 years[20]. We used this model to simulate genomes at time periods and in sample sizes equivalent to our empirical dataset and inferred tree sequences using Relate[21,22]. We trained a neural network classifier to estimate the path backwards in time through the population structure taken by each simulated individual, at each position in the genome. Our trained classifier was then used to infer the ancestral paths taken at each site, using 1,015 imputed ancient genomes from West Eurasia that passed quality filters. Using simulations, we show that our chromosome painting method has an average accuracy of 94.6% for the four ancestral paths leading to present-day Europeans and is robust to model misspecification.

We then adapted CLUES[23] to model aDNA time-series data (Supplementary Notes 4 and 5) and used it to infer allele frequency trajectories and selection coefficients for 33,341 quality-controlled trait-associated variants from the GWAS Catalog[24]. An equal number of putatively neutral, frequency-paired variants were used as a control set (Supplementary Note 4). To control for possible confounders, we built a causal model to distinguish direct effects of age on allele frequency from indirect effects mediated by read depth, read length and/or error rates (Supplementary Note 6) and developed a mapping bias test used to evaluate systematic differences between data from ancient and present-day populations (Supplementary Note 4). Because admixture between groups with differing allele frequencies can confound interpretation of allele frequency changes through time, we used the local ancestry paths from our chromosome painting model to stratify haplotypes in our selection tests. By conditioning on these path labels, we are able to infer selection trajectories while controlling for changes in admixture proportions through time.

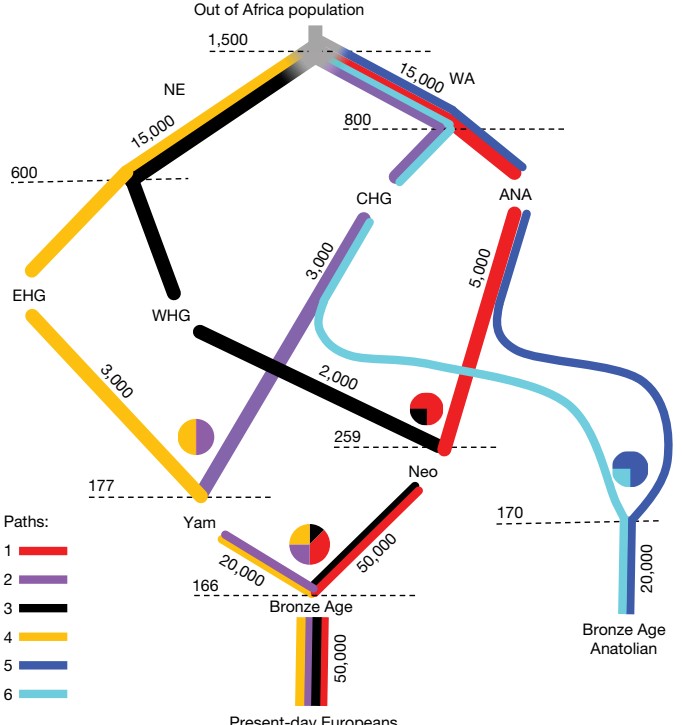

## Fig. 3 | A schematic of the model of population structure in Europe.

Quantitative admixture model used to simulate genomes to train the local ancestry neural network classifier. The model begins with the Out-of-Africa population, before splitting into basal Northern Europeans (NE) and West Asians (WA), who further split into EHG, WHG, CHG and ANA. These then admix to form Steppe pastoralist (Yam) and Neolithic farmer (Neo) populations. Moving down the figure is forwards in time and the population split times and admixture times are given in generations ago. Each branch is labelled with the effective population size of the population. Coloured lines represent the populations declared in the simulation that extend through time.

Our analysis identified no genome-wide significant ($P < 5 \times 10^{-8}$) selective sweeps when using genomes from present-day individuals alone (1000 Genomes Project populations GBR, FIN and TSI[25]), although trait-associated variants were enriched for evidence of selection compared to the control group ($P < 7.29 \times 10^{-35}$, Wilcoxon signed-rank test). By contrast, when using imputed aDNA genotype probabilities, we identified 11 genome-wide significant selective sweeps in the GWAS group ($n = 476$ SNPs with $P < 5 \times 10^{-8}$) and no sweeps in the control group, despite some SNPs exhibiting evidence of selection ($n = 51$). These results are consistent with selection preferentially acting on trait-associated variants. We then conditioned our selection analysis on each of our four local ancestry pathways—that is, local ancestry tracts passing through WHG, EHG, CHG or Anatolian farmers (ANA)—and identified 21 genome-wide significant selection peaks (Fig. 4 and Extended Data Figs. 1–10). This suggests that admixture between ancestral populations has masked evidence of selection at many trait-associated loci in Eurasian populations[26].

## Selection on diet-associated loci

We find strong changes in selection associated with lactose digestion after the introduction of farming but before the expansion of the Steppe pastoralists into Europe around 5,000 years ago[27,28], the timing of which is a long-standing controversy[29–32]. The strongest overall signal of selection in the pan-ancestry analysis is observed at the *MCM6/LCT* locus (rs4988235: A; $P = 1.68 \times 10^{-59}$; $s = 0.0194$), where the derived allele results in lactase persistence[33]. The trajectory inferred from the

pan-ancestry analysis indicates that the lactase persistence allele began increasing in frequency about 6,000 years ago and has continued to increase up to the present (Fig. 4). In the ancestry-stratified analyses, this signal is driven primarily by sweeps in two of the ancestral backgrounds, associated with EHG and CHG. We also observed that many selected SNPs within this locus exhibited earlier evidence of selection than at rs4988235, suggesting that selection at the *MCM6/LCT* locus is more complex than previously thought. To investigate this further, we expanded our selection scan to include all SNPs within the ~2.6 megabase (Mb)-wide sweep locus ($n = 5,608$) and checked for the earliest evidence of selection. We observed that most genome-wide significant SNPs at this locus began rising in frequency earlier than rs4988235, indicating that strong positive selection at this locus predates the emergence of the lactase persistence allele by thousands of years. Among the alleles showing much earlier frequency rises was rs1438307: T ($P = 9.77 \times 10^{-24}$; $s = 0.0146$), which began rising in frequency about 12,000 years ago (Fig. 4). This allele has been shown to regulate energy expenditure and contribute to metabolic disease and it has been suggested to be an ancient adaptation to famine[34]. The high linkage disequilibrium between rs1438307 and rs4988235 in present-day individuals ($R^2 = 0.89$ in GBR) may explain the recently observed correlation between frequency rises in the lactase persistence allele and archaeological proxies for famine and increased pathogen exposure[35]. To control for potential bias introduced by imputation, we replicated these results using genotype likelihoods, called directly from the aDNA sequencing reads, and with publicly available 1240k capture array data from the Allen Ancient DNA Resource v.52.2 (ref. 36) (Supplementary Note 4).

We also found strong selection in the FADS gene cluster—*FADS1* (rs174546: C; $P = 4.41 \times 10^{-19}$; $s = 0.0126$) and *FADS2* (rs174581: G; $P = 2.21 \times 10^{-19}$; $s = 0.0138$)—which are associated with fatty acid metabolism and known to respond to changes in diet from a more/less vegetarian to a more/less carnivorous diet[37–41]. In contrast to previous results[39–41], we find that much of the selection associated with a more vegetarian diet occurred in Neolithic populations before they arrived in Europe, then continued during the Neolithic (Fig. 4). The strong signal of selection in this region in the pan-ancestry analysis is driven primarily by a sweep occurring across the EHG, WHG and ANA haplotypic backgrounds (Fig. 4). Interestingly, we do not find statistically significant evidence of selection at this locus in the CHG background but most of the allele frequency rise in the EHG background occurs after their admixture with CHG (around 8,000 years ago[42]), within whom the selected alleles were already close to present-day frequencies. This suggests that the selected alleles may already have existed at substantial frequencies in early farmer populations in the Middle East and among Caucasus hunter-gatherers (associated with the ANA and CHG backgrounds, respectively) and were subject to continued selection as eastern groups moved northwards and westwards during the late Neolithic and Bronze Age periods.

When specifically comparing selection signatures differentiating ancient hunter-gatherer and farmer populations[43], we also observe many regions associated with lipid and sugar metabolism and various metabolic disorders (Supplementary Note 7). These include, for example, a region in chromosome 22 containing *PATZ1*, which regulates the expression of *FADS1* and *MORC2*, which plays an important role in cellular lipid metabolism[44]. Another region in chromosome 3 overlaps with *GPR15*, which is both related to immune tolerance and to intestinal homoeostasis[45,46]. Finally, in chromosome 18, we recover a selection candidate region spanning *SMAD7*, which is associated with inflammatory bowel diseases such as Crohn's disease[47]. Taken together these results indicate that the transition to agriculture imposed a substantial amount of selection for humans to adapt to a new diet and lifestyle and that the prevalence of some diseases observed today may be a consequence of these selective processes.

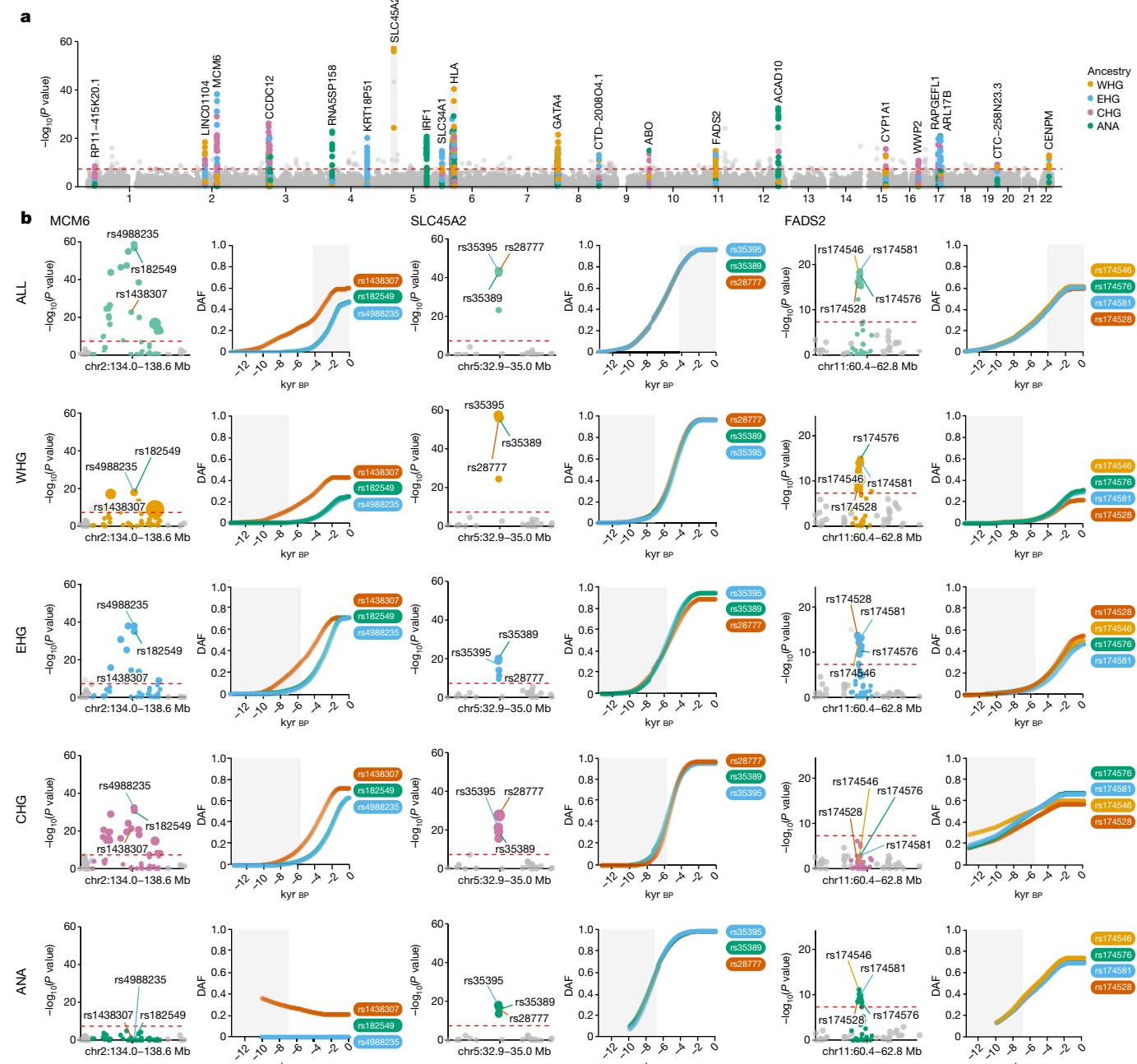

**Fig. 4 | Genome-wide selection scan for trait-associated variants.**
**a**, Manhattan plot of *P* values from selection scan with CLUES, based on a time-series of imputed aDNA genotype probabilities. Twenty-one genome-wide significant selection peaks highlighted in grey and labelled with the gene closest to the most significant SNP within each locus. Within each sweep, SNPs are positioned on the *y* axis and coloured by their most significant marginal ancestry. Outside of the sweeps, SNPs show *P* values from the pan-ancestry analysis and are coloured grey. Red dotted lines indicate genome-wide

significance ($P < 5 \times 10^{-8}$). **b**, Detailed plots for three genome-wide significant sweep loci: (1) *MCM6*, lactase persistence; (2) *SLC45A2*, skin pigmentation; and (3) *FADS2*, lipid metabolism. Rows show results for the pan-ancestry analysis (ALL) plus the four marginal ancestries: WHG, EHG, CHG and ANA. The first column of each locus shows zoomed Manhattan plots of the *P* values for each ancestry and column two shows allele frequency trajectories for the top SNPs across all ancestries (grey shading for the marginal ancestries indicates approximate temporal extent of the pre-admixture population).

## Selection on immunity-associated loci

We also observe evidence of strong selection in several loci associated with immunity and autoimmune disease (Supplementary Note 4). Some of these putative selection events occurred earlier than previously claimed and are probably associated with the transition to agriculture, which may help explain the high prevalence of autoimmune diseases today. Most notably, we detect an 8 Mb-wide selection sweep signal in chromosome 6 (chr6: 25.4–33.5 Mb), spanning the full length of the

human leukocyte antigen (HLA) region. The selection trajectories of the variants within this locus support several independent sweeps, occurring at different times and with differing intensities. The strongest signal of selection at this locus in the pan-ancestry analysis is at an intergenic variant, located between *HLA-A* and *HLA-W* (rs7747253: A; $P = 7.56 \times 10^{-32}$; $s = -0.0178$), associated with protection against chicken-pox (odds ratio (OR) 0.888; ref. 5), increased risk of intestinal infections (OR 1.08; ref. 48) and decreased heel bone mineral density (OR 0.98; ref. 49). This allele rapidly decreased in frequency, beginning about

8,000 years ago (Extended Data Fig. 3), reducing the risk of intestinal infections, at the cost of increasing the risk of chickenpox. By contrast, the signal of selection at *C2* (rs9267677: C; $P = 6.60 \times 10^{-26}$; $s = 0.0441$), also found within this sweep, shows a gradual increase in frequency beginning around 4,000 years ago, before rising more rapidly about 1,000 years ago. In this case, the favoured allele is associated with protection against some sexually transmitted diseases (STDs) (OR 0.786; ref. 48), primarily those caused by human papillomavirus, and with increased psoriasis risk (OR 2.2; ref. 5). This locus provides a good example of the possibility that the high prevalence of autoimmune diseases in present-day populations may, in part, be due to genetic trade-offs; by which selection increased protection against pathogens with the pleiotropic effect of increased susceptibility to autoimmune diseases[50].

These results also highlight the complex temporal dynamics of selection at the HLA locus, which not only plays a role in the regulation of the immune system but is also associated with many non-immune-related phenotypes. The high pleiotropy in this region makes it difficult to determine which selection pressures may have driven these increases in frequencies at different periods of time. However, profound shifts in lifestyle in Eurasian populations during the Holocene have been suggested to be drivers for strong selection on loci involved in immune response. These include a change in diet and closer contact with domestic animals, combined with higher mobility and increasing population density. We further explore the complex pattern of ancestry-specific selection at the HLA locus in our companion paper[51].

We also identify selection signals at the *SLC22A4* (rs35260072: C; $P = 8.49 \times 10^{-20}$; $s = 0.0172$) locus, associated with increased itch intensity from mosquito bites (OR 1.049; ref. 52), protection against childhood and adult asthma (OR 0.902 and 0.909; ref. 48) and asthma-related infections (OR 0.913; ref. 48) and we find that the derived variant has been steadily rising in frequency since around 9,000 years ago (Extended Data Fig. 9). However, in the same *SLC22A4* candidate region as rs35260072, we find that the frequency of the previously reported allele rs1050152: T (which also protects against asthma (OR 0.90; ref. 48) and related infections) plateaued about 1,500 years ago, contrary to previous reports suggesting a recent rise in frequency[7]. Similarly, we detect selection at the *HECTD4* (rs11066188: A; $P = 9.51 \times 10^{-31}$ $s = 0.0198$) and *ATXN2* (rs653178: C; $P = 3.73 \times 10^{-29}$; $s = 0.0189$) loci, both of which have been rising in frequency for about 9,000 years (Extended Data Fig. 4), also contrary to previous reports of a more recent rise in frequency[7]. These SNPs are associated with protection against urethritis and urethral syndrome (OR 0.769 and 0.775; ref. 48), which are often caused by STDs or accumulated urethral damage from having more than five births. Both SNPs are also linked to increased risk of intestinal infectious diseases (OR 1.03 and 1.04), several non-specific parasitic diseases (OR 1.44 and 1.59; ref. 48), schistosomiasis (OR 1.13 and 1.32; ref. 48), helminthiases (OR 1.29 and 1.28; ref. 48), spirochaetes (OR 1.14 and 1.12; ref. 48), pneumonia (OR 1.03 and 1.03; ref. 48) and viral hepatitis (OR 1.15 and 1.15; ref. 5). These SNPs also increase the risk of coeliac disease and rheumatoid arthritis[53]. Thus, several highly pleiotropic disease-associated loci, which were previously thought to be the result of recent adaptation, may have been subject to selection for a much longer period of time.

## Selection on the 17q21.31 locus

We further detect signs of strong selection in a 2 Mb sweep on chromosome 17 (chr. 17: 44.0–46.0 Mb), spanning a locus on 17q21.3, implicated in neurodegenerative and developmental disorders. The locus includes an inversion and other structural polymorphisms with indications of a recent positive selection sweep in some human populations[54,55]. Specifically, partial duplications of the *KANSL1* gene probably occurred independently on the inverted (H2) and non-inverted (H1) haplotypes (Extended Data Fig. 11a) and both are found in high frequencies (15–25%) among current European and Middle Eastern populations, but are much

rarer in Sub-Saharan African and East Asian populations. We used both SNP genotypes and WGS read depth information to determine inversion (H1/H2) and *KANSL1* duplication (d) status in the ancient individuals studied here (Supplementary Note 8).

The H2 haplotype is observed in two of three previously published genomes[56] of Anatolian aceramic-associated Neolithic individuals (Bon001 and Bon004) from around 10,000 BP but data were insufficient to identify *KANSL1* duplications. The oldest evidence for *KANSL1* duplications is observed in an early Neolithic individual (AH1 from 9,900 BP; ref. 57) from present-day Iran, followed by two Mesolithic individuals (NEO281 from 9,724 BP and KK1 (ref. 58) from 9,720 BP), from present-day Georgia, all of whom are heterozygous for the inversion and carry the inverted duplication. The *KANSL1* duplications are also detected in two Neolithic individuals, from present-day Russia (NEO560 from 7,919 BP (H1d) and NEO212 from 7,390 BP (H2d)). With both H1d and H2d having spread to large parts of Europe with Anatolian Neolithic farmer ancestries, their frequency seems unchanged in most of Europe as Steppe-related ancestries become dominant in large parts of the subcontinent (Extended Data Fig. 11c). The fact that both H1d and H2d are found in apparently high frequencies in both early Anatolian farmers and the earliest Steppe-related ancestry groups suggests that any selective sweep acting on the H1d and H2d variants would probably have occurred in populations ancestral to both.

We note that the strongest signal of selection observed in the pan-ancestry analysis at this locus is at *MAPT* (rs4792897: G; $P = 1.33 \times 10^{-18}$; $s = 0.0299$ (Extended Data Fig. 8 and Supplementary Note 4), which codes for the tau protein[59] and is associated with protection against mumps (OR 0.776; ref. 48) and increased risk of snoring (OR 1.04; ref. 60). More generally, polymorphisms in *MAPT* have been associated with increased risk of several neurodegenerative disorders, including Alzheimer's disease and Parkinson's disease[61]. However, we caution that this region is also enriched for evidence of reference bias in our dataset—especially around the *KANSL1* gene—due to complex structural polymorphisms (Supplementary Note 10).

## Selection on pigmentation loci

Our results identify strong selection for lighter skin pigmentation in groups moving northwards and westwards, consistent with the idea that selection is caused by reduced ultraviolet exposure and resulting vitamin D deficiency. We find that the most strongly selected alleles reached near-fixation several thousand years ago, suggesting that this process was not associated with recent sexual selection as previously proposed[62]. In the pan-ancestry analysis, we detect strong selection at the *SLC45A2* locus (rs35395: C; $P = 1.60 \times 10^{-44}$; $s = 0.0215$)[8,63], with the selected allele (responsible for lighter skin) increasing in frequency from around 13,000 years ago, until plateauing around 2,000 years ago (Fig. 4). The predominant hypothesis is that high melanin levels in the skin are important in equatorial regions owing to its protection against ultraviolet radiation, whereas lighter skin has been selected for at higher latitudes (where ultraviolet radiation is less intense) because some ultraviolet penetration is required for cutaneous synthesis of vitamin D[64,65]. Our findings confirm pigmentation alleles as key targets of selection during the Holocene[7,66], particularly on a small proportion of loci with large effect sizes[8].

Our results also provide detailed information about the duration and geographic spread of these processes (Fig. 4), suggesting that an allele associated with lighter skin was selected for repeatedly, probably as a consequence of similar environmental pressures occurring at different times in different regions. In the ancestry-stratified analysis, all marginal ancestries show broad agreement at the *SLC45A2* locus (Fig. 4) but differ in the timing of their frequency shifts. The ANA-associated ancestry background shows the earliest evidence for selection at rs35395, followed by EHG and WHG around 10,000 years ago and CHG about 2,000 years later. In all ancestry backgrounds, except ANA, the selected

haplotypes plateau at high frequency by about 2,000 years ago, whilst the ANA haplotype background reaches near-fixation 1,000 years earlier. We also detect strong selection at the *SLC24A5* locus (rs1426654: A; $P = 2.28 \times 10^{-16}$; $s = 0.0185$), which is also associated with skin pigmentation[63,67]. At this locus, the selected allele increased in frequency even earlier than *SLC45A2* and reached near-fixation around 3,500 years ago. Selection on this locus thus seems to have occurred early on in groups that were moving northwards and westwards and only later in the WHG background after these groups encountered and admixed with the incoming populations.

## Selection among major axes of variation

Beyond patterns of genetic change at the Mesolithic–Neolithic transition, much genetic variability observed today reflects high genetic differentiation in the hunter-gatherer groups that eventually contributed to present-day European genetic diversity[43]. Indeed, many loci associated with cardiovascular disease, metabolism and lifestyle diseases trace their genetic variability before the Neolithic transition to ancient differential selection in ancestry groups occupying different parts of the Eurasian continent (Supplementary Note 7). These may represent selection episodes that preceded the admixture events described above and led to differentiation between ancient hunter-gatherer groups in the late Pleistocene and early Holocene. One of these overlaps with the *SLC24A3* gene, which is a salt-sensitivity gene significantly expressed in obese individuals[68]. Another spans *ROPN1* and *KALRN*, two genes involved in vascular disorders[69]. A further region contains *SLC35F3*, which codes for a thiamine transport[70] and has been associated with hypertension in a Han Chinese cohort[71]. Finally, there is a candidate region containing several genes (*CH25H* and *FAS*) associated with obesity and lipid metabolism[72,73] and another peak with several genes (*ASXL2, RAB10, HADHA* and *GPR113*) involved in glucose homoeostasis and fatty acid metabolism[74–77]. These loci predominantly reflect ancient patterns of extreme differentiation between Eastern and Western Eurasian genomes and may be candidates for selection after the separation of the Pleistocene populations that occupied different environments across the continent (roughly 45,000 years ago[13]).

## Pathogenic structural variants

Rare, recurrent copy-number variants (CNVs) are known to cause neurodevelopmental disorders and are associated with a range of psychiatric and physical traits with variable expressivity and incomplete penetrance[78,79]. To understand the prevalence of pathogenic structural variants over time we examined 50 genomic regions susceptible to recurrent CNVs, known to be the most prevalent drivers of human developmental pathologies[80]. The analysis included 1,442 ancient shotgun genomes passing quality control for CNV analysis (Supplementary Note 10) and 1,093 present-day human genomes for comparison[81,82]. We identified CNVs in ancient individuals at ten loci using a read depth-based approach and digital comparative genomic hybridization[83]. Although most of the observed CNVs (including duplications at 15q11.2 and *CHRNA7* and CNVs spanning parts of the TAR locus and 22q11.2 distal) have not been unambiguously associated with disease in large studies, the identified CNVs include deletions and duplications that have been associated with developmental delay, dysmorphic features and neuropsychiatric abnormalities such as autism (most notably at 1q21.1, 3q29, 16p12.1 and the DiGeorge/VCFS locus but also deletions at 15q11.2 and duplications at 16p13.11). Overall, the carrier frequency in the ancient individuals is similar to that reported in the UKB genomes (1.25% versus 1.6% at 15q11.2 and *CHRNA7* combined and 0.8% versus 1.1% across the remaining loci combined)[84]. These results indicate that large, recurrent CNVs, which can lead to several pathologies, were present at similar frequencies in the ancient and present-day populations included in this study.

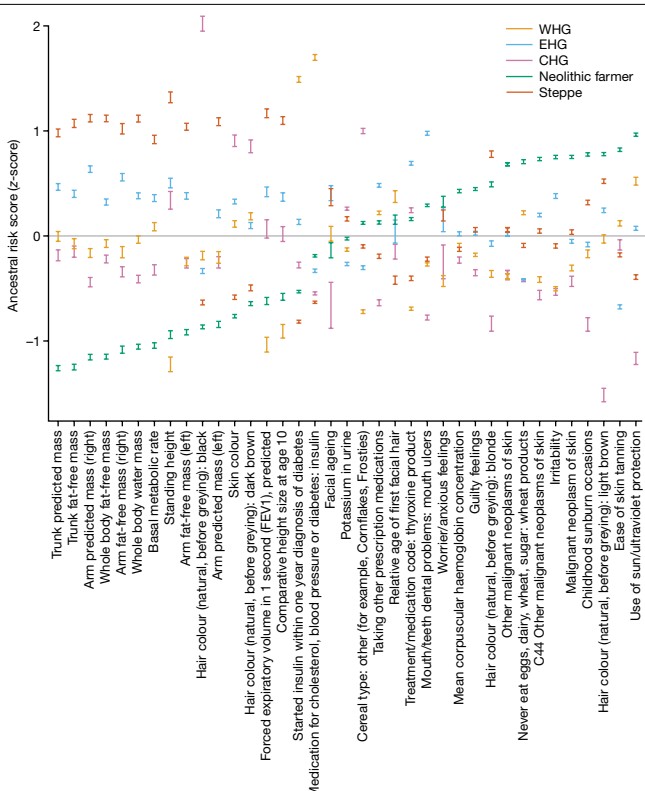

**Fig. 5 | ARSs for 35 complex traits.** Showing the genetic risk that a present-day individual would possess if they were composed entirely of one ancestry. On the basis of chromosome painting of the UKB, for 35 complex traits found to be significantly overdispersed in ancient populations. Confidence intervals (95%) are estimated by bootstrapping present-day samples ($n = 408,884$) and centred on the mean estimate.

## Phenotypic legacy of ancient Eurasians

In addition to identifying evidence of selection for trait-associated variants, we also estimated the contribution from different genetic ancestries (associated with EHG, CHG, WHG, Steppe pastoralists and Neolithic farmers) to variation in complex traits in present-day individuals. We calculated ancestry-specific polygenic risk score—hereafter ancestral risk scores (ARS)—on the basis of chromosome painting of over 400,000 UKB samples using ChromoPainter[11] (Fig. 5 and Supplementary Note 9). This allowed us to identify which ancient ancestry components are over-represented in present-day UK populations at loci significantly associated with a given trait and is analogous to the genetic risk that a present-day individual would possess if they were composed entirely of one of the ancestry groupings defined in this study. This analysis avoids issues related to the portability of polygenic risk scores between populations[85], as our ARSs are calculated from the same individuals used to estimate the effect sizes. Working with many imputed ancient genomes provides high statistical power to use ancient populations as ancestral sources. We focused on 35 phenotypes whose polygenic scores were significantly overdispersed among the ancient populations (Supplementary Note 9), as well as well as three large effect alleles at the *APOE* gene (ApoE2, ApoE3 and ApoE4) known to significantly mediate risk of developing Alzheimer's disease[86]. We emphasize that this approach makes no direct reference to ancient phenotypes but instead describes how these genetic ancestry components contributed to the present-day phenotypic landscape.

We find that for many anthropometric traits—such as trunk predicted mass, forced expiratory volume in 1-second (FEV1) and basal metabolic rate—the ARS for Steppe ancestry was the highest, followed by EHG and

CHG/WHG, whilst Neolithic farmer ancestry consistently scored the lowest for these measurements. Consistent with previous studies, hair and skin pigmentation also showed significant differences, with scores for skin colour for WHG, EHG and CHG higher (that is, darker) than for Neolithic farmer and Steppe-associated ancestries[8,9,27,28]; and scores for traits related to malignant neoplasms of the skin were elevated in Neolithic farmer-associated ancestries. Both Neolithic farmer- and Steppe-associated ancestries have higher scores for blonde and light brown hair, whereas the hunter-gatherer-associated ancestries have higher scores for dark brown hair and CHG-associated ancestries had the highest score for black hair.

In terms of genetic contributions to risk for diseases, the WHG ancestral component had strikingly high scores for traits related to cholesterol, blood pressure and diabetes. The Neolithic farmer component scored the highest for anxiety, guilty feelings and irritability; CHG and WHG ancestry components consistently scored the lowest for these three traits. We found the ApoE4 allele (rs429358: C and rs7412: C, which increases risk of Alzheimer's disease) preferentially painted with a WHG/EHG haplotypic background, suggesting it was probably brought into Western Eurasia by early hunter-gatherers (Supplementary Note 9). This result is in line with the present-day European distribution of this allele, which is highest in northeastern Europe, where the proportion of these ancestries is larger than in other regions of the continent[87]. By contrast, we found the ApoE2 allele (rs429358: T and rs7412: T, which decreases the risk for Alzheimer's disease) on a haplotypic background with affinities to Steppe pastoralists. Our pan-ancestry analysis identified positive selection favouring ApoE2 ($P = 6.99 \times 10^{-3}$; $s = 0.0130$), beginning about 7,000 years ago and plateauing around 2,500 years ago (Supplementary Note 4). However, we did not identify evidence of selection for either ApoE3 (rs429358: T and rs7412: C) or ApoE4, contrary to a recent study with a smaller sample size and unphased genotypes[88]. The selective forces probably favouring ApoE2 in Steppe pastoralists may be associated with protective immune responses against infectious challenges, such as protection against malaria or an unknown viral infection (Supplementary Note 9).

In light of the ancestry gradients within the United Kingdom and across Eurasia (Fig. 2), these results support the suggestion that migration-mediated geographic variation in phenotypes and disease risk is commonplace, and points to a way forward for explaining geographically structured disease prevalence through differential admixture processes between present-day populations. These results also help to clarify the famous discussion of selection in Europe relating to height[7,89]. Our finding that the Steppe- and EHG-associated ancestral components have elevated genetic values for height in the UKB demonstrates that height differences between Northern and Southern Europe may be a consequence of differential ancestry, rather than selection, as claimed in many previous studies[90]. However, our results do not preclude the possibility that height has been selected for in specific populations[91,92].

## Discussion

The fundamental changes in diet resulting from the transitions from hunting and gathering to farming and subsequently to pastoralism, precipitated far-reaching consequences for the physical and mental health of present-day Eurasian populations. These dramatic cultural changes created a heterogeneous mix of selection pressures, probably related to changes in diet and increased population densities, including selection for resistance to new infectious challenges. Owing to the highly pleiotropic nature of each sweep region, it is difficult to ascribe causal factors to any of our selection signals and we did not exhaustively test all non-trait-associated variants. However, our results show that selection during the Holocene has had a substantial impact on present-day genetic disease risk, as well as the distribution of genetic factors affecting metabolic and anthropometric traits. Our analyses have also shown that the ability to detect signatures of natural selection

in present-day human genomes is drastically limited by conflicting selection pressures in different ancestral populations masking the signals. Developing methods to trace selection while accounting for differential admixture allowed us to effectively double the number of genome-wide significant selection peaks and helped clarify the trajectories of several variants related to diet and lifestyle. Furthermore, we have shown that numerous complex traits thought to have been under local selection are better explained by differing genetic contributions of ancient individuals to present-day variation. Overall, our results emphasize how the interplay between ancient selection and the main admixture events occurring in the Mesolithic, Neolithic and Bronze Age have profoundly shaped the patterns of genetic variation observed in present-day humans across Eurasia.

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

## Reporting summary

Further information on research design is available in the Nature Portfolio Reporting Summary linked to this article.

## Data availability

All ancient genomic data used in this study are already published and listed in Supplementary Table 1. Data were aligned to the human reference GRCh37. Modern human genomes were obtained from the 1000 Genomes Project[25], the Simons Genome Diversity Project[81] and the Human Genome Diversity Project[82]. GWAS data were obtained from the GWAS Catalog[24], the FinnGen Study[48] and the UKB[5].

## Code availability

The scripts used to run the chromosome painting (Supplementary Note 2) and calculate ARS in the UKB (Supplementary Note 9) are available at https://github.com/will-camb/mesoneo_selection_paper (https://doi.org/10.5281/zenodo.8301166). The software to perform the ancestral path chromosome painting described in Supplementary Note 3 is available on GitHub at https://github.com/AliPearson/Ancestral-Paths (https://doi.org/10.5281/zenodo.8319452) and the demographic model is available in the stdpopsim library (https://popsim-consortium.github.io/stdpopsim-docs/stable/catalog.html#sec_catalog_homsap_models_ancienteurope_4a21). The analysis pipeline and 'conda' environment necessary to replicate the analysis of allele frequency trajectories of trait-associated variants in Supplementary Note 4 are available at https://github.com/ekirving/mesoneo_paper (https://doi.org/10.5281/zenodo.8289755). The modified version of CLUES used in this study is available from https://github.com/standard-aaron/clues (https://doi.org/10.5281/zenodo.8228252). The pipeline to replicate the analyses for Supplementary Note 7 can be found at https://github.com/albarema/neo (https://doi.org/10.5281/zenodo.8301253). All other analyses relied on available software which has been fully referenced in the manuscript and detailed in the relevant Supplementary notes.

**Acknowledgements** We thank all the former and current staff at the Lundbeck Foundation GeoGenetics Centre and the GeoGenetics Sequencing Core and colleagues across the many institutions detailed below. We are particularly grateful to L. Olsen as project manager for the Lundbeck Foundation GeoGenetics Centre project. We thank UKB for access to the UKB genomic resource. We want to acknowledge the participants and investigators of the FinnGen study. We are thankful to Illumina for collaboration. E.W. thanks St. John's College, Cambridge, for providing a stimulating environment of discussion and learning. The Lundbeck Foundation GeoGenetics Centre is supported by the Lundbeck Foundation (R302-2018-2155 and R155-2013-16338), the Novo Nordisk Foundation (NNF18SA0035006), the Wellcome Trust (214300), Carlsberg Foundation (CF18-0024), the Danish National Research Foundation (DNRF94 and DNRF174), the University of Copenhagen (KU2016 programme), Ferring Pharmaceuticals A/S and a COREX ERC Synergy grant (ID 951385). This research has been conducted using the UKB Resource and the iPSYCH Initiative, funded by the Lundbeck Foundation (R102-A9118 and R155-2014-1724). This work was further supported by the Swedish Foundation for Humanities and Social Sciences grant (Riksbankens Jubileumsfond M16-0455:1) to K.K.. E.K.I.-P. and A.R.-M. were supported by the Lundbeck Foundation (R302-2018-2155) and the Novo Nordisk Foundation (NNF18SA0035006). A.P., R.D. and E.W. were supported by the Wellcome Trust (214300). R.M. was supported by a SSHRC doctoral studentship (G101449). R.A.H. and T.K. were supported by the Carlsberg Foundation (CF19-0712). L.S. was supported by a Sir Henry Wellcome fellowship (220457/Z/20/Z). L.F. was supported by the OAK Foundation (OCAY-15-520). P.H.S. was supported by the Institute of General Medical Sciences (R35GM142916) and a Vallee Scholars Award. R.N. was supported by the National Institutes of Health (R01GM138634). F.R. was supported by a Villum Young Investigator Grant (project no. 00025300), a Novo Nordisk Fonden Data Science Ascending Investigator Award (NNF22OC0076816) and by the European Research Council (ERC) under the European Union's Horizon Europe programme (grant agreements No. 101077592 and 951385).

**Author contributions** E.K.I.-P., A.R.-M., W.B., A.I., A.P. and A.F. contributed equally to this work. P.H.S., D.J.L., R.D., T.K., T.W., M.E.A., M.S., R.N., F.R. and E.W. led the study. A.F., T.W., M.E.A., M.S. and E.W. conceptualized the study. P.H.S., D.J.L., R.D., T.K., T.W., M.E.A., M.S., R.N., F.R. and E.W. supervised the research. M.E.A., K.K., R.D., T.W., R.N. and E.W. acquired funding for research. E.K.I.-P., A.R.-M., A.I., A.P., W.B., A.H.V., L.S., A. J. Stern, K.K., D.J.L., R.D., T.K. M.E.A., M.S., R.N. and F.R. were involved in developing and applying methodology. E.K.I.-P., A.R.-M., A.I., A.P., W.B., A.S.H., R.A.H., T.V., H.M., A.H.V., L.S., A. Ramsøe, A. J. Schork, A. Rosengren, L.Z., P.H.S., T.K., M.E.A., M.S. and F.R. undertook formal analyses of data. E.K.I.-P., A.R.-M., A.I., A.P., A.F., W.B., K.-G.S., A.S.H., R.A.H., T.V., H.M., A.H.V., L.S., A. Ramsøe, A. J. Stern, G.S., A. Ramsøe, A. Rosengren, L.Z., A.K.N.I., L.F., P.H.S., D.J.L., T.K., M.S., F.R. and E.W. drafted the main text. E.K.I.-P., A.R.-M., A.I., A.P., A.F., W.B., K.-G.S., A.S.H., R.A.H., T.V., A. J. Stern, A. Ramsøe, A. Rosengren, L.Z., A.K.N.I., L.F., P.H.S., D.J.L., M.S. and E.W. drafted the Supplementary Information and Tables. E.K.I.-P., A.R.-M., A.I., A.P., A.F., W.B., K.-G.S., A.S.H., R.M., F.D., R.A.H., T.V., H.M., A. Ramsøe, A. J. Schork, L.Z., K.K., A.K.N.I., L.F., P.H.S., D.J.L., R.D., T.K., T.W., M.E.A., M.S., R.N., F.R. and E.W. were involved in reviewing drafts and editing. All co-authors read, commented on and agreed on the submitted manuscript.

**Competing interests** The authors declare no competing interests.

**Additional information**
**Correspondence and requests for materials** should be addressed to Evan K. Irving-Pease, Rasmus Nielsen, Fernando Racimo or Eske Willerslev.

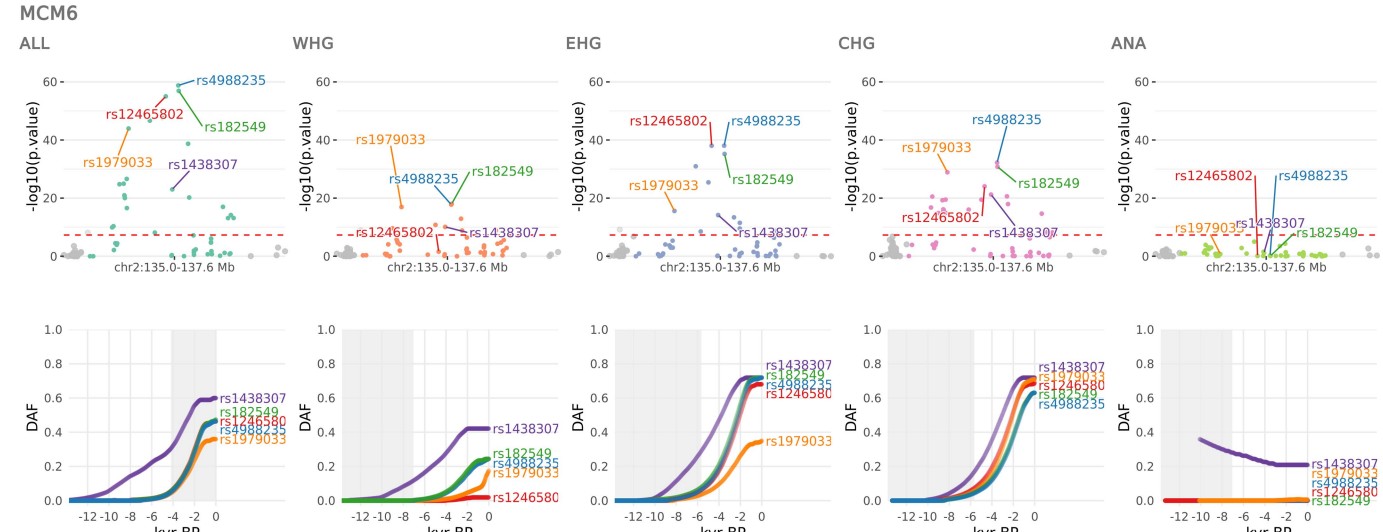

**Extended Data Fig. 1 | Selection at the *MCM6* locus.** CLUES selection results for the most significant sweep locus, showing the pan-ancestry analysis (ALL) plus the four marginal ancestries: Western hunter-gatherers (WHG), Eastern hunter-gatherers (EHG), Caucasus hunter-gatherers (CHG) and Anatolian farmers (ANA). Row one shows zoomed Manhattan plots of the p values for each ancestry and row two shows allele trajectories for the top SNPs across all ancestries (grey shading for the marginal ancestries indicates approximate temporal extent of the pre-admixture population).

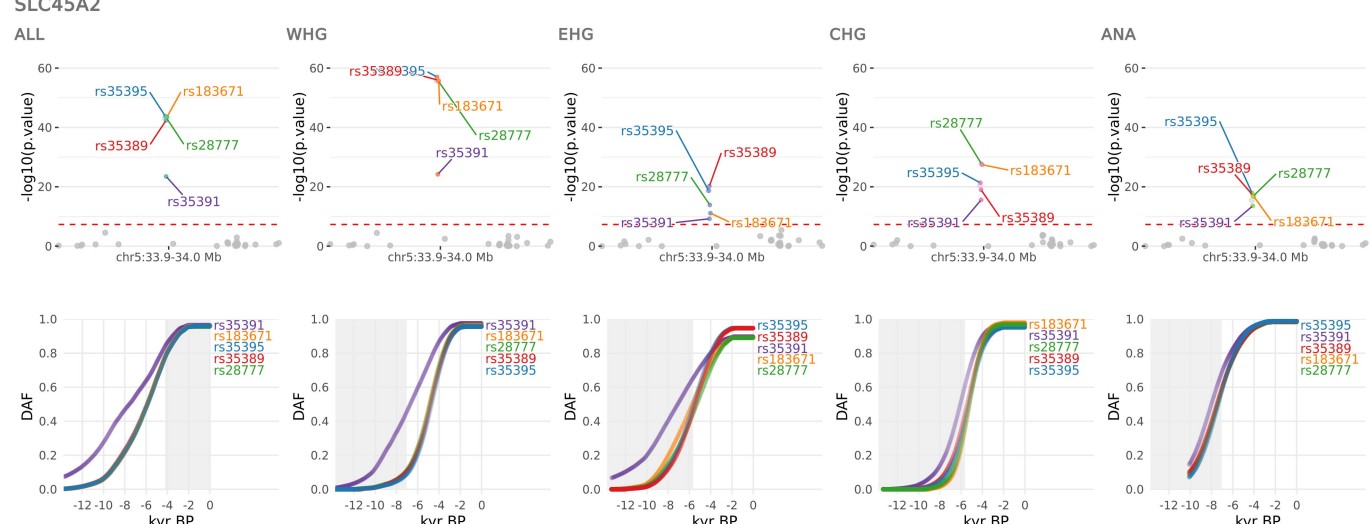

**Extended Data Fig. 2 | Selection at the *SLC45A2* locus.** CLUES selection results for the second most significant sweep locus, showing the pan-ancestry analysis (ALL) plus the four marginal ancestries: Western hunter-gatherers (WHG), Eastern hunter-gatherers (EHG), Caucasus hunter-gatherers (CHG) and Anatolian farmers (ANA). Row one shows zoomed Manhattan plots of the p values for each ancestry and row two shows allele trajectories for the top SNPs across all ancestries (grey shading for the marginal ancestries indicates approximate temporal extent of the pre-admixture population).

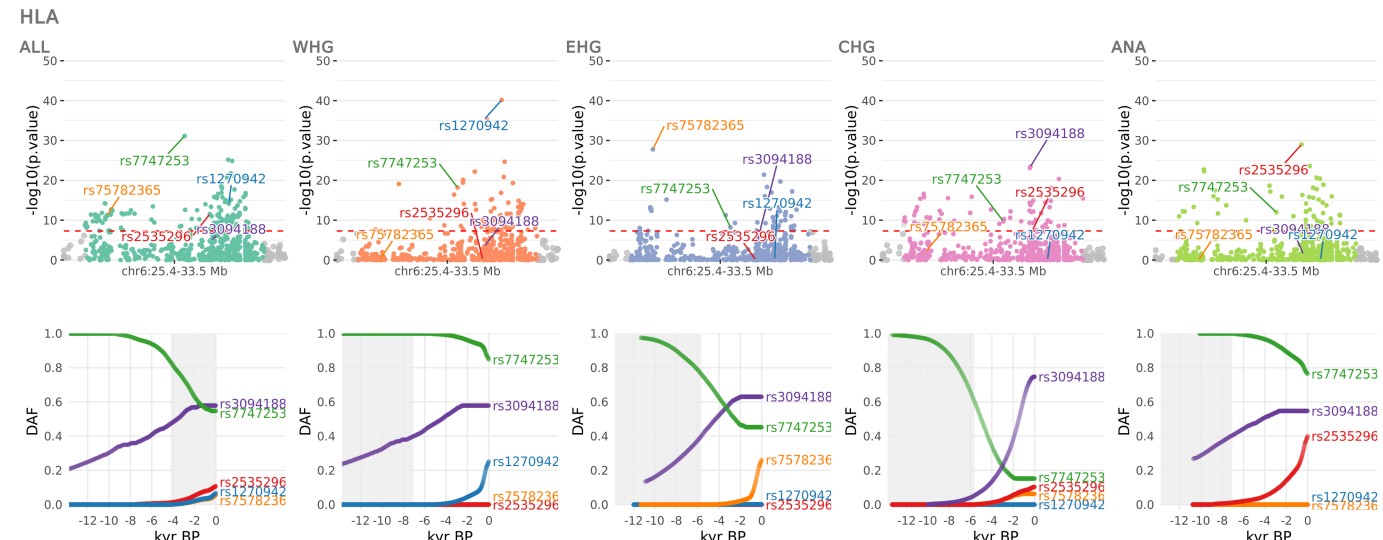

**Extended Data Fig. 3 | Selection at the HLA locus.** CLUES selection results for the third most significant sweep locus, showing the pan-ancestry analysis (ALL) plus the four marginal ancestries: Western hunter-gatherers (WHG), Eastern hunter-gatherers (EHG), Caucasus hunter-gatherers (CHG) and Anatolian farmers (ANA). Row one shows zoomed Manhattan plots of the p values for each ancestry and row two shows allele trajectories for the top SNPs across all ancestries (grey shading for the marginal ancestries indicates approximate temporal extent of the pre-admixture population).

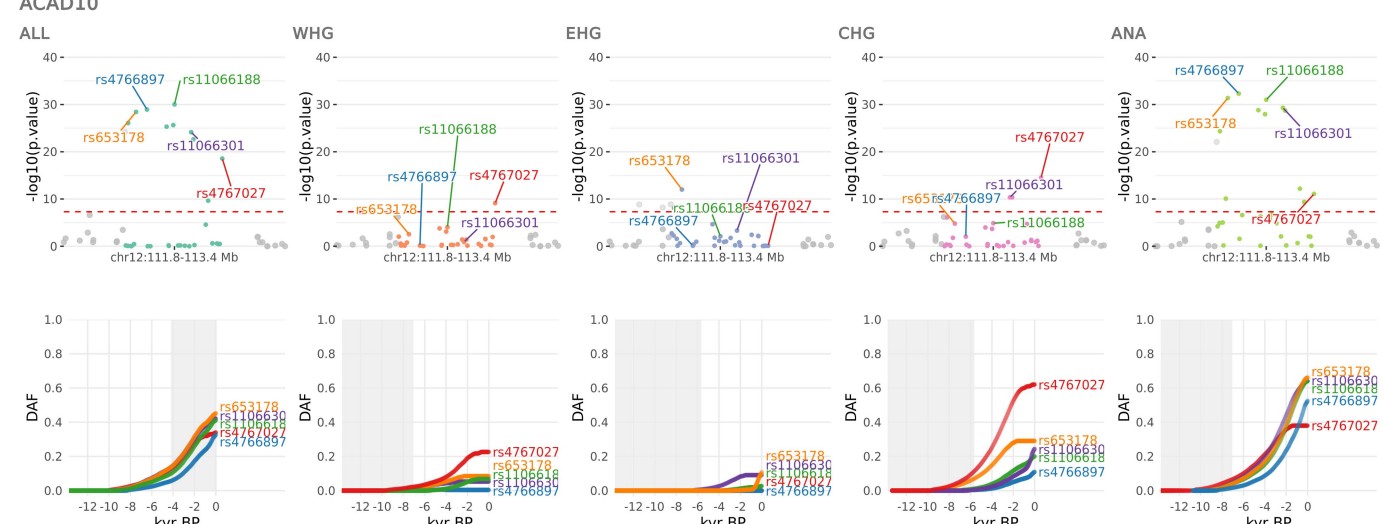

**Extended Data Fig. 4 | Selection at the *ACAD10* locus.** CLUES selection results for the fourth most significant sweep locus, showing the pan-ancestry analysis (ALL) plus the four marginal ancestries: Western hunter-gatherers (WHG), Eastern hunter-gatherers (EHG), Caucasus hunter-gatherers (CHG) and Anatolian farmers (ANA). Row one shows zoomed Manhattan plots of the p values for each ancestry and row two shows allele trajectories for the top SNPs across all ancestries (grey shading for the marginal ancestries indicates approximate temporal extent of the pre-admixture population).

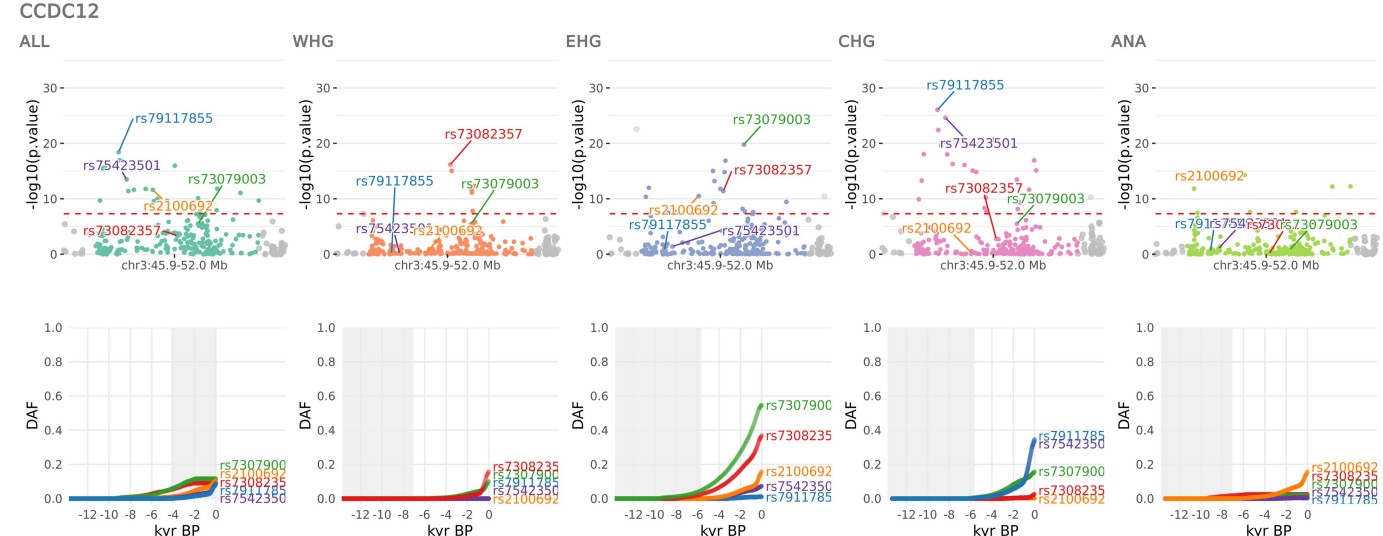

**Extended Data Fig. 5 | Selection at the *CCDC12* locus.** CLUES selection results for the fifth most significant sweep locus, showing the pan-ancestry analysis (ALL) plus the four marginal ancestries: Western hunter-gatherers (WHG), Eastern hunter-gatherers (EHG), Caucasus hunter-gatherers (CHG) and Anatolian farmers (ANA). Row one shows zoomed Manhattan plots of the p values for each ancestry and row two shows allele trajectories for the top SNPs across all ancestries (grey shading for the marginal ancestries indicates approximate temporal extent of the pre-admixture population).

RNA5SP158

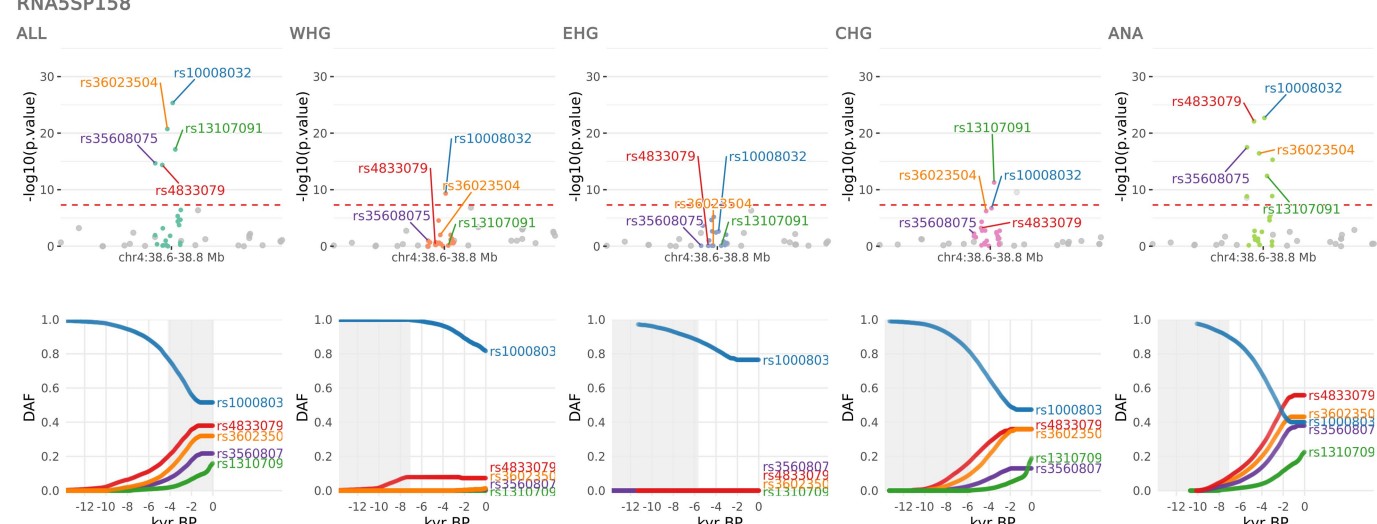

**Extended Data Fig. 6 | Selection at the *RNA5SP158* locus.** CLUES selection results for the sixth most significant sweep locus, showing the pan-ancestry analysis (ALL) plus the four marginal ancestries: Western hunter-gatherers (WHG), Eastern hunter-gatherers (EHG), Caucasus hunter-gatherers (CHG) and Anatolian farmers (ANA). Row one shows zoomed Manhattan plots of the p values for each ancestry and row two shows allele trajectories for the top SNPs across all ancestries (grey shading for the marginal ancestries indicates approximate temporal extent of the pre-admixture population).

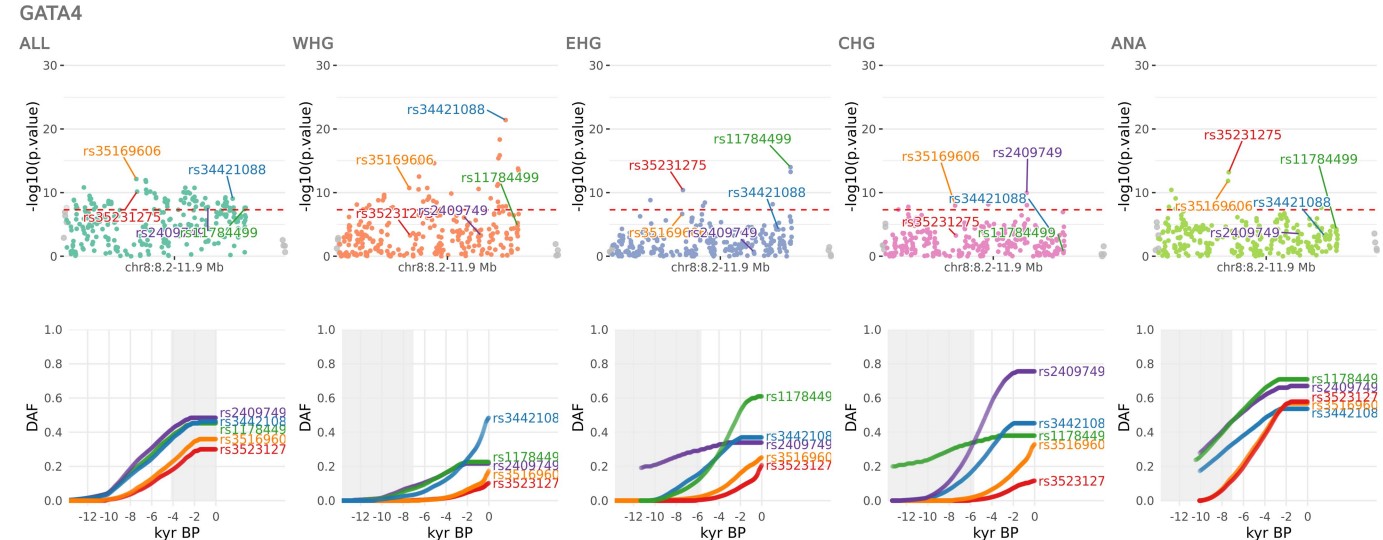

**Extended Data Fig. 7 | Selection at the *GATA4* locus.** CLUES selection results for the seventh most significant sweep locus, showing the pan-ancestry analysis (ALL) plus the four marginal ancestries: Western hunter-gatherers (WHG), Eastern hunter-gatherers (EHG), Caucasus hunter-gatherers (CHG) and Anatolian farmers (ANA). Row one shows zoomed Manhattan plots of the p values for each ancestry and row two shows allele trajectories for the top SNPs across all ancestries (grey shading for the marginal ancestries indicates approximate temporal extent of the pre-admixture population).

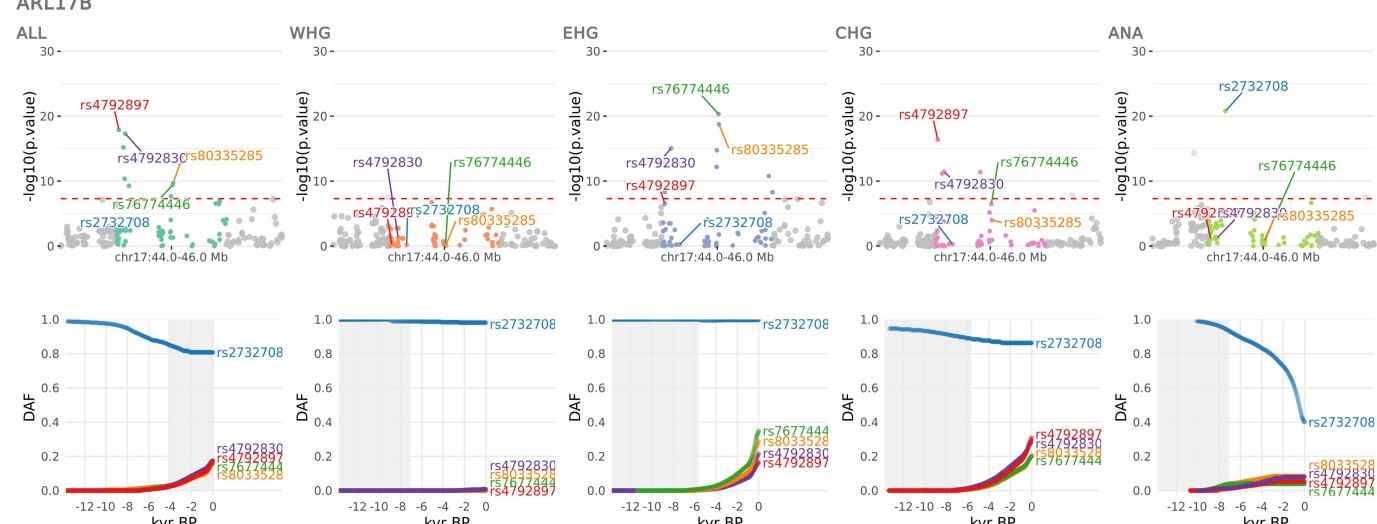

**Extended Data Fig. 8 | Selection at the *ARL17B* locus.** CLUES selection results for the eighth most significant sweep locus, showing the pan-ancestry analysis (ALL) plus the four marginal ancestries: Western hunter-gatherers (WHG), Eastern hunter-gatherers (EHG), Caucasus hunter-gatherers (CHG) and Anatolian farmers (ANA). Row one shows zoomed Manhattan plots of the p values for each ancestry and row two shows allele trajectories for the top SNPs across all ancestries (grey shading for the marginal ancestries indicates approximate temporal extent of the pre-admixture population).

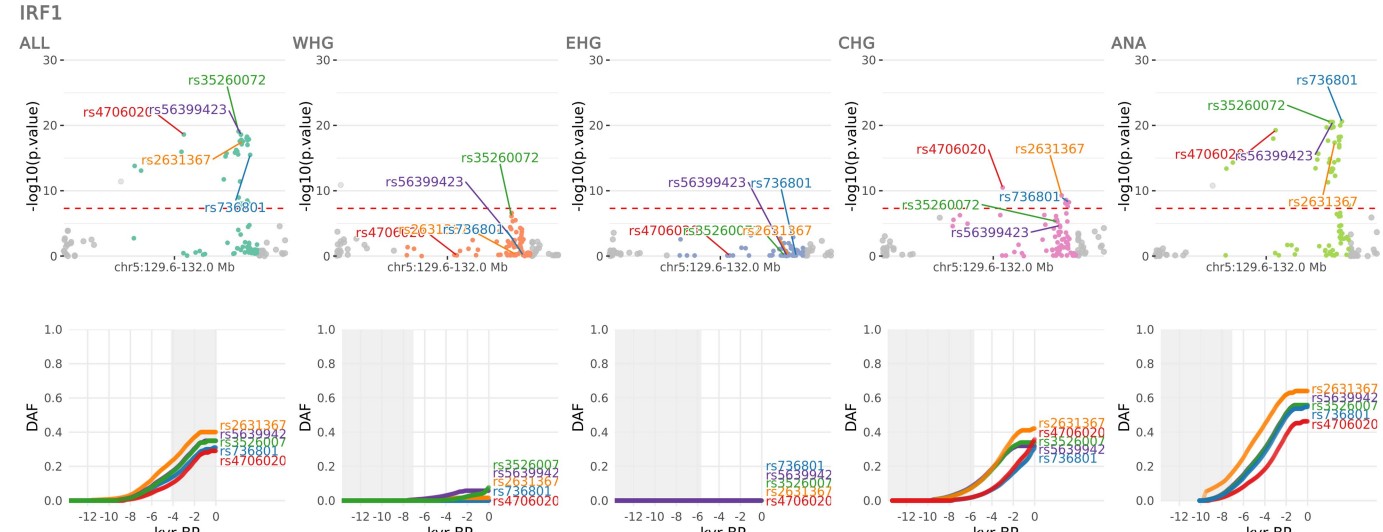

**Extended Data Fig. 9 | Selection at the *IRF1* locus.** CLUES selection results for the ninth most significant sweep locus, showing the pan-ancestry analysis (ALL) plus the four marginal ancestries: Western hunter-gatherers (WHG), Eastern hunter-gatherers (EHG), Caucasus hunter-gatherers (CHG) and Anatolian farmers (ANA). Row one shows zoomed Manhattan plots of the p values for each ancestry and row two shows allele trajectories for the top SNPs across all ancestries (grey shading for the marginal ancestries indicates approximate temporal extent of the pre-admixture population).

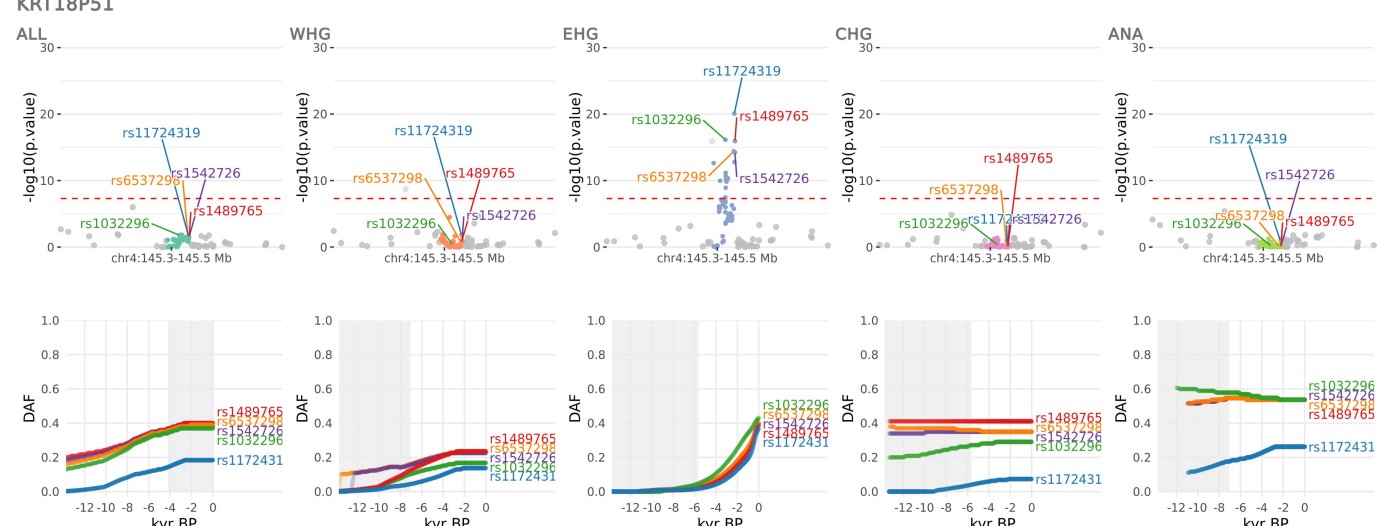

**Extended Data Fig. 10 | Selection at the *KRT18P51* locus.** CLUES selection results for the tenth most significant sweep locus, showing the pan-ancestry analysis (ALL) plus the four marginal ancestries: Western hunter-gatherers (WHG), Eastern hunter-gatherers (EHG), Caucasus hunter-gatherers (CHG) and Anatolian farmers (ANA). Row one shows zoomed Manhattan plots of the p values for each ancestry and row two shows allele trajectories for the top SNPs across all ancestries (grey shading for the marginal ancestries indicates approximate temporal extent of the pre-admixture population).

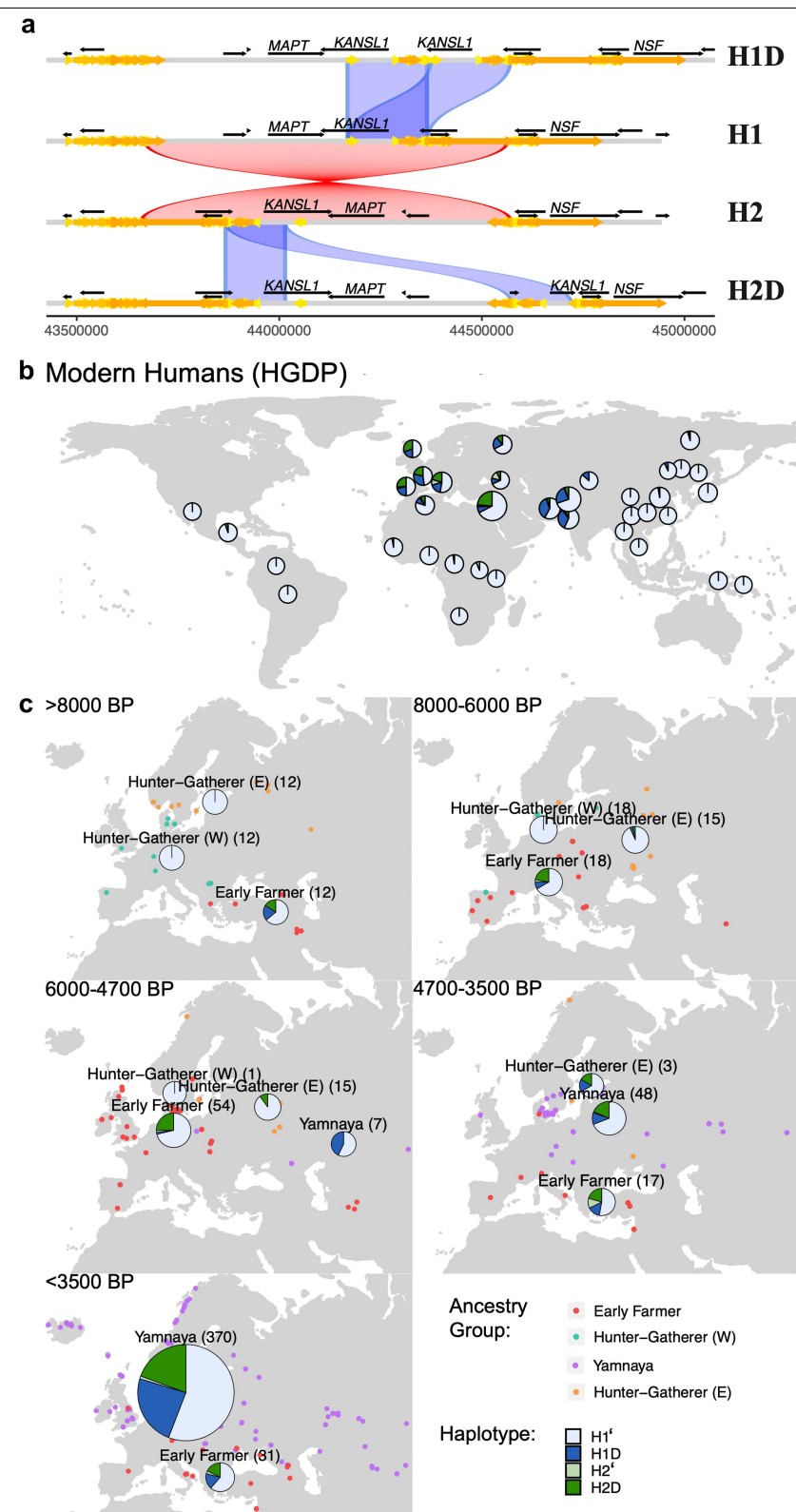

**Extended Data Fig. 11 | The 17q21.31 inversion locus.** A) Haplotypes of the 17q21.31 locus: the ancestral (non-inverted) H1 17q21.31 and the inverted H2 haplotype. Duplications of the *KANSL1* gene have occurred independently on both lineages yielding H1D and H2D haplotypes. B) Frequency of the 17q21.31 inversion and duplication haplotypes across present-day global populations (Human Genome Diversity Project[82]). C) Change in the frequency of the 17q21.31 inversion haplotype through time.

# Reporting Summary

## Statistics

For all statistical analyses, confirm that the following items are present in the figure legend, table legend, main text, or Methods section.

| n/a | Confirmed | |
|---|---|---|
| ☐ | ☒ | The exact sample size (*n*) for each experimental group/condition, given as a discrete number and unit of measurement |
| ☐ | ☒ | A statement on whether measurements were taken from distinct samples or whether the same sample was measured repeatedly |
| ☐ | ☒ | The statistical test(s) used AND whether they are one- or two-sided *Only common tests should be described solely by name; describe more complex techniques in the Methods section.* |
| ☐ | ☒ | A description of all covariates tested |
| ☐ | ☒ | A description of any assumptions or corrections, such as tests of normality and adjustment for multiple comparisons |
| ☐ | ☒ | A full description of the statistical parameters including central tendency (e.g. means) or other basic estimates (e.g. regression coefficient) AND variation (e.g. standard deviation) or associated estimates of uncertainty (e.g. confidence intervals) |
| ☐ | ☒ | For null hypothesis testing, the test statistic (e.g. *F*, *t*, *r*) with confidence intervals, effect sizes, degrees of freedom and *P* value noted *Give P values as exact values whenever suitable.* |
| ☐ | ☒ | For Bayesian analysis, information on the choice of priors and Markov chain Monte Carlo settings |
| ☐ | ☒ | For hierarchical and complex designs, identification of the appropriate level for tests and full reporting of outcomes |
| ☐ | ☒ | Estimates of effect sizes (e.g. Cohen's *d*, Pearson's *r*), indicating how they were calculated |

*Our web collection on statistics for biologists contains articles on many of the points above.*

## Software and code

Policy information about availability of computer code

| | |
|---|---|
| Data collection | No software was used for data collection. |
| Data analysis | The scripts used to run the chromosome painting (Supplementary Note 1b) and calculate ARS in the UK Biobank (Supplementary Note 2f) are available at https://github.com/will-camb/mesoneo_selection_paper (https://doi.org/10.5281/zenodo.8301166). The software to perform the ancestral path chromosome painting described in Supplementary Note 1c is available on GitHub at https://github.com/AliPearson/AncestralPaths (https://doi.org/10.5281/zenodo.8319452), and the demographic model is available in the stdpopsim library (see https://popsim-consortium.github.io/ stdpopsim-docs/stable/catalog.html#sec_catalog_homsap_models_ancienteurope_4a21). The analysis pipeline and 'conda' environment necessary to replicate the analysis of allele frequency trajectories of trait-associated variants in Supplementary Note 2a are available on Github at https://github.com/ekirving/mesoneo_paper (https://doi.org/10.5281/zenodo.8289755). The modified version of CLUES used in this study is available from https://github.com/standard-aaron/clues (https://doi.org/10.5281/zenodo.8228252). The pipeline to replicate the analyses for Supplementary Note 2d can be found at https://github.com/albarema/neo (https://doi.org/10.5281/zenodo.8301253). All other analyses relied upon available software which has been fully referenced in the manuscript and detailed in the relevant supplementary notes, including: bcftools v1.10.2, bedtools v2.29.2, biopython v1.76, clues v36cb7de, conda v4.9.0, numpy v1.17.0, pandas v1.0.4, pysam v0.15.3, python v3.6.7, r-base v3.6.1, r-bedr v1.0.7, r-dplyr v0.8.0.1, r-ggplot2 v3.1.1, r-ggrastr v0.2.1, r-ggrepel v0.8.2, r-ggridges v0.5.1, r-stringr v1.4.0, relate v1.1.3, scipy v1.4.1, and snakemake v5.12.3. |

For manuscripts utilizing custom algorithms or software that are central to the research but not yet described in published literature, software must be made available to editors and reviewers. We strongly encourage code deposition in a community repository (e.g. GitHub). See the Nature Portfolio guidelines for submitting code & software for further information.

# Data

Policy information about availability of data

All manuscripts must include a data availability statement. This statement should provide the following information, where applicable:
- Accession codes, unique identifiers, or web links for publicly available datasets
- A description of any restrictions on data availability
- For clinical datasets or third party data, please ensure that the statement adheres to our policy

All ancient genomic data used in this study are already published and listed in Supplementary Table S1. Data was aligned to the human reference GRCh37. Modern human genomes were obtained from the 1000 Genomes Project (1KGP), the Simons Genome Diversity Project (SGDP) and the Human Genome Diversity Project (HGDP). GWAS data was obtained from the GWAS Catalog, the FinnGen Study, and the UK Biobank (UKB).

# Research involving human participants, their data, or biological material

Policy information about studies with human participants or human data. See also policy information about sex, gender (identity/presentation), and sexual orientation and race, ethnicity and racism.

| | |
|---|---|
| Reporting on sex and gender | Sex was assigned based on sex chromosomes. Sex-specific results were not calculated. |
| Reporting on race, ethnicity, or other socially relevant groupings | Reporting was restricted to a self-identified 'white British' cohort, with PCA outliers removed (details in Bycroft et al., 2018). |
| Population characteristics | *Describe the covariate-relevant population characteristics of the human research participants (e.g. age, genotypic information, past and current diagnosis and treatment categories). If you filled out the behavioural & social sciences study design questions and have nothing to add here, write "See above."* |
| Recruitment | *Describe how participants were recruited. Outline any potential self-selection bias or other biases that may be present and how these are likely to impact results.* |
| Ethics oversight | Use of the UK Biobank resource was approved in 2020. |

Note that full information on the approval of the study protocol must also be provided in the manuscript.

# Field-specific reporting

Please select the one below that is the best fit for your research. If you are not sure, read the appropriate sections before making your selection.

☐ Life sciences  ☐ Behavioural & social sciences  ☒ Ecological, evolutionary & environmental sciences

For a reference copy of the document with all sections, see nature.com/documents/nr-reporting-summary-flat.pdf

# Ecological, evolutionary & environmental sciences study design

All studies must disclose on these points even when the disclosure is negative.

| | |
|---|---|
| Study description | Using an imputed dataset of >1600 complete ancient genome sequences, and new computational methods for locating selection in time and space, we reconstructed the selection landscape of the transition from hunting and gathering, to farming and pastoralism across West Eurasia. |
| Research sample | Our analyses are undertaken on a dataset comprising 1664 imputed diploid ancient genomes, and more than 8.5 million SNPs. These samples represent a considerable transect of Eurasia, ranging longitudinally from the Atlantic coast to Lake Baikal, and latitudinally from Scandinavia to the Middle East. Included are many of the key Mesolithic and Neolithic cultures of Western Eurasia, Ukraine, western Russia, and the Trans-Urals, constituting a thorough temporal sequence of human populations from 11,000 cal. BP to 3,000 cal. BP. |
| Sampling strategy | Sampling was dependent upon the availability of ancient human remains, though a considerable transect of the Mesolithic and Neolithic periods in Eurasia was represented, together with a detailed continuous sequence of human occupation of Denmark specifically. Our assemblage is well-represented with individuals of such key archaeological complexes as the Maglemose, Ertebølle and Funnel Beaker cultures in Scandinavia, the Cardial in the Mediterranean, the Körös and Linear Pottery complexes in SE and Central Europe, and many archaeological cultures in Ukraine, western Russia, and the trans-Ural (e.g. Veretye, Lyalovo, Volosovo, Kitoi). |
| Data collection | We collected 1664 previously published ancient shotgun genomes from 70 prior publications. |
| Timing and spatial scale | Sample chronology was generated by radiocarbon dating, with dates corrected for marine and freshwater reservoir effects. These showed samples ranging from the Upper Palaeolithic (c. 25,700 cal. BP) to the mediaeval period (c. 1200 cal. BP). Most individuals |

(97%, N=309) span 11,000 cal. BP to 3,000 cal. BP, the period broadly associated with the Mesolithic and Neolithic in Eurasia. Our research area can broadly be divided into three large regions: 1) central, western and northern Europe, 2) eastern Europe including western Russia and Ukraine, and 3) the Urals and western Siberia.

Data exclusions — Low coverage and related samples were excluded.

Reproducibility — Data quality and uncertainty (e.g. contamination) was accounted for in computational analyses to assess robustness of inferences, and all methods and data are made available for future replication.

Randomization — Sample groups were defined to reflect archaeological populations, based on phylogenetic inferences, temporal and geographic provenance, and cultural interpretations evidenced by archaeological contexts.

Blinding — Blinding was not applicable to this study.

Did the study involve field work? ☐ Yes ☒ No

# Reporting for specific materials, systems and methods

We require information from authors about some types of materials, experimental systems and methods used in many studies. Here, indicate whether each material, system or method listed is relevant to your study. If you are not sure if a list item applies to your research, read the appropriate section before selecting a response.

## Materials & experimental systems

| n/a | Involved in the study |
|-----|------------------------|
| ☒ | ☐ Antibodies |
| ☒ | ☐ Eukaryotic cell lines |
| ☐ | ☒ Palaeontology and archaeology |
| ☒ | ☐ Animals and other organisms |
| ☒ | ☐ Clinical data |
| ☒ | ☐ Dual use research of concern |
| ☒ | ☐ Plants |

## Methods

| n/a | Involved in the study |
|-----|------------------------|
| ☒ | ☐ ChIP-seq |
| ☒ | ☐ Flow cytometry |
| ☒ | ☐ MRI-based neuroimaging |

## Palaeontology and Archaeology

Specimen provenance — Details of the provenance of all samples analyzed in this study are available in Supplementary Table S1.

Specimen deposition — *Indicate where the specimens have been deposited to permit free access by other researchers.*

Dating methods — *If new dates are provided, describe how they were obtained (e.g. collection, storage, sample pretreatment and measurement), where they were obtained (i.e. lab name), the calibration program and the protocol for quality assurance OR state that no new dates are provided.*

☒ Tick this box to confirm that the raw and calibrated dates are available in the paper or in Supplementary Information.

Ethics oversight — No ethical approval required.

Note that full information on the approval of the study protocol must also be provided in the manuscript.

