## [Peer Review File · Nature]

Manuscript Title: THE SELECTION LANDSCAPE AND GENETIC LEGACY OF ANCIENT EURASIANS

Reviewer Comments & Author Rebuttals

Reviewer Reports on the Initial Version:

Referee expertise:

Referee #1: human evolutionary genetics

Referee #2: aDNA/human evolution

Referee #3: population genetics

Referees' comments:

Referee #1 (Remarks to the Author):

This manuscript integrates several hundred new ancient human DNA samples along with previous human aDNA data to build a dataset of ~1600 imputed ancient genomes. It then applies several new computational methods for the inference of admixture proportions and allele frequency changes suggestive of selection. A major strength of the approach is the ability to control for differences in ancestral background in ancient and modern samples. The resulting catalog of evolutionary pressures on European phenotypes over that past 12,000 years suggests that ancient selection and admixture have played a larger role in modern phenotypes than recent local selection.

Major Comments:

The two paragraphs from lines 123-153 describe the data and fundamental analytic methods used in this study, and as such, they are essential to the plausibility and interpretation of all subsequent results. These approaches include several predictive analyses that integrate new methods and data from this manuscript with recently published methods. While I appreciate the length constraints, extensive supplementary material, and desire to get to the results quickly, more detail on these methods (especially their accuracy and validation) must be provided in the main text.

To illustrate this, here are some of the new claims the reader is asked to accept in these paragraphs without any details in main text (and this is assuming that existing methods like RELATE and CLUES are sufficiently accurate):

- The ancient genomes are accurately imputed and phased.
- The new chromosome painting ancestry inference method is accurate at the haplotype level on both ancient and modern genomes.
- The simulation framework is not sensitive to misspecification/inaccuracies in the four-population admixture model.

- The neural network classifier for ancestral path inference is sufficiently accurate.
- The updates to the CLUES framework enable accurate allele frequency and selection coefficient estimates. (Big kudos for the snakemake and clear github page.)
- The control set of SNPs is appropriate.

I found these approaches to generally be reasonable and the details in the supplementary material to be helpful, but the evaluation was often lacking. To illustrate this, I provide a few non-exhaustive examples where more validation/justification are needed. In the section on the neural network to predict paths backward in time (S1a), the only evaluation provided is the confusion matrix on simulated data in Figure S1a.2. It seems that there is considerable misclassification even on the simulated data. (However, I note that there is not a scale bar, so I can't really evaluate the magnitude of the values in each box.) In S2b, there is no evaluation of the method for inferring allele frequency changes influenced by bias. And there is not a justification for the 0.5 threshold on F_{j} . Out of the context of the distribution of this metric across sites or its effect on the likelihood of inferring selection, this still seems quite low. Similarly in the most of the supplementary sections I did not find the level of evaluation I expected.

Following on this, the bigger point is that it is challenging to evaluate the sensitivity of the results to inaccuracies in each of the modeling and predictive steps listed above. Given that the results of one analysis are often used as inputs to the next, I fear the potential for errors and propagation. Anything the authors can do to better understand this would be extremely valuable in establishing confidence in the results.

I appreciated the comparison to the control group in the GWAS variant selection analyses, but I have two questions about this. First, isn't it important to match the control SNPs on LD as well as MAF, since it is likely associated both with the probability that a variant is a GWAS hit and experienced selection? This is commonly done in tools like SNPSNAP. Second, the finding of many more selection peaks when conditioning on ancestry is interesting. However, there was not any evaluation via simulations of the power to detect different types of selective events when considering ancestry or not. Couldn't this just be due to an increase in power when considering ancestry?

Given the challenges of porting polygenic risk scores across even closely related populations, I was very surprised to see the attempts at PRS-based ancestral trait reconstruction. The authors are aware of these challenges and repeatedly suggest "caution" in the interpretation. This is insufficient as no work has been done to evaluate the feasibility or accuracy of this analysis. Given the known challenges and lack of specific hypotheses guiding these analyses, their value to the manuscript is not clear. I would suggest removing these if stronger justification cannot be provided. (Also, PMID: 29285967 should be cited.) That said, the estimation of the contributions of different ancestral populations to variation in phenotypes in the UK Biobank seems on stronger methodological footing. However, these results are not presented in any detail except to say that they point to a way forward for disentangling ancestry contributions to differences in genetic disease risk (L460). I suggest expanding the presentation of these results instead of the ancient phenotype prediction.

I would also like to see more direct discussion of how the results relate to those of a few recent similar studies. For example:

doi: <https://doi.org/10.1101/2022.07.02.498543> and <https://doi.org/10.1101/2022.08.24.505188> both have traced selection over the past 10,000 years using different methods. PMID: 36316412 argued that admixture can hide selective events.

Minor Comments:

While I appreciate that the details of the construction of the cohort are provided in a companion paper, a few more sentences and perhaps a figure panel describing the geographic locations and ages would be helpful.

Tone down exaggerated statements. For example, on L111 and L119, is this dataset truly “unprecedented”? While this is a wonderful dataset, by now many studies have analyzed hundreds of ancient individuals’ genomes. Thus, I disagree that it is unprecedented.

Similarly, I thought the insights into the timing and different variants potentially involved in selection at the LCT/MCM6 locus were fascinating. But it is not clear to me that this analysis will completely settle the “controversies regarding the timing of this selection” on lactose digestion (L172).

And again, in the discussion of height and selection (L464-470), this study adds valuable new data and hypotheses, but I am not convinced that it is “settled” (and I don’t have a stake in this debate) so I would reframe this section

The sweep “loci” seem extreme large from several of the examples given: multiple Mb for most and 33 Mb for HLA. Can the authors comment on whether this is a resolution issue or likely to reflect selection on multiple variants within these windows (as would be expected for a locus like the HLA)?

At several points a “population structure axis separating” populations is referred to with a reference to Figure 2. I believe that this is in error. Perhaps it should be Figure 4A? Also, the variance explained by the PCs is incredibly small.

Figure 1: I do not see a blue line for pop_4. But perhaps it is not supposed to appear since the numbering starts with 0?

I suspect that this is a PDF conversion issue, but all the figures are blurry.

Referee #2 (Remarks to the Author):

Irving-Pease et al. report analysis searching for natural selection in a large set of >1,600 imputed shotgun genomes, many of which were produced in a "main paper" (Allentoft et al.) to which this paper is a companion paper. They take approaches including inference of selection coefficients on genealogies, decomposition of evolution in different ancestral populations, evolution of polygenic traits and evidence for adaptation on those, and searching for structural variants with evidence of pathogenicity.

The paper is well-written, and several approaches and analysis here are highly interesting and definitely push the field of studying natural selection with ancient DNA forward (for example, the ancestry-decomposed inference of selection coefficients), but it is less clear which new biological insights are learned, and there are several issues that need mention and clarification.

The paper consists of hundreds, perhaps thousands, of claims of selection on single variants and traits, each caveated largely appropriately, but the study is thus different from most papers focusing on presenting tight waterproof evidence for a handful of central claims. Reading the paper, I can see a few claims that I am guessing that the authors are highly confident in, but also others where I am unsure of what level of confidence there actually is.

Of the major claims in the abstract:

-Selection on metabolism, and HLA seems to have been reported previously (e.g. Mathieson et al. 2015, Nature). The link between immune selection and autoimmune disease is entirely speculation, no new advance on the possible link is reported.

-Selection at the FADS cluster and the lactase persistence locus began earlier than previously thought. This seems like novel claims, and the evidence seems strong. At the same time, saying that the debate is "settled" is probably premature. This claim could be supported by additional evidence, how robust is the new timing proposed by the authors, and exactly what timings had been suggested before?

--Differential genetic contributions in height ancestral to present-day Europeans. Yamnaya introduced tall height. This seems largely in line with previous finds (e.g. by Mathieson et al. 2015; Cox and Mathieson 2020, PNAS).

--"Alleles associated with increased risk of some mood-related phenotypes are overrepresented in the farmer ancestry component." "Western hunter-gatherers show a strikingly high contribution of alleles conferring risk of traits related to diabetes." These seem like novel claims, but are the authors confident of the claims, or do they think it could be subject to the caveats about projecting GWAS scores from present-day panels into the past that they bring up in the text? If they want present this as a significant scientific advance, can they expand on the evidence for this, and the robusticity of the analysis. How outlying are the scores for these?

--"a combination of ancient selection and migration, rather than recent local selection, is the primary driver of present-day phenotypic differences in Europe." To me this seems to have been the consensus for some time. Have recent papers suggested it was due to local selection?

There is an interesting find of natural selection on an inversion and duplication of KANSL1, but this is not mentioned in the abstract.

Overall, a lot of the novel results seem to be claims about different timing of selection, than what has been suggested in previous paper. However, the authors don't really provide extensive simulation results or other validation experiments on how robust their timing inference is.

Overall, it is difficult to disentangle the scientific advance of the paper from that of the data that it presents the first selection analyses on, but were really produced by another paper (which will be cited for the data itself). That said, this paper has some very interesting analyses going, in some ways more interesting than the advances presented in the parallel submission with the new data.

Major issues:

The authors conduct their main selection analysis (CLUES) not on the whole genome, but on a selected set of 33k SNPs from the GWAS catalog. They then match those with a neutral set. This seems like a sensible way to test if those SNPs have been subject to selection, but the caveat is that when they then dissect the selection peaks in the data, the actually targeted variant may lay somewhere nearby in the genome. There is thus an extra risk that when they discuss variants subject to selection and associated with particular traits in the text, the natural selection was not on those traits. I presume that this was not done genome-wide since CLUES is not easily scaled to such data, but this particular approach seems to increase the risk of storytelling, as there will always be a well-documented GWAS SNP at the height of every signal peak. Could the authors provide convincing evidence that those top SNPs are in fact the SNPs targeted by selection?

Imputation seems key to the conclusion in the paper, could the authors discuss a bit more about how the results could be robust to imputation?

The authors also seem to discuss one primary trait association for each SNP. Are any of the SNPs discussed in the main text associated with multiple traits?

The CLUES inferred trajectories seem highly constrained, but to which degree is this due to the heatmap colour scheme used in the supplementary figures. Could they colour values down to posterior probability ~ 0.05 more clearly? Also, these trajectories take the imputation and genealogical inference for granted, and thus do not portray the uncertainty associated with those.

On page 34 in the supplement, they report that in the CLUES analysis of aDNA with Ancestral Paintings, they identify quite substantial numbers of outliers also in the control set of SNPs, 346 in the GWAS group and 63 in the Control group. This doesn't seem to be mentioned in the main text.

The authors investigate the correlation of PCs with polygenic scores for traits such as height, but it does not seem appropriate to treat individuals as independent observations (they are related at different degrees), so it seems that some by-chromosome bootstrap or similar could gauge evolutionary uncertainty.

Regarding the possibility of SNPs beginning a frequency rise earlier than the classic lactase persistence candidate, the suggested 12,000 years ago for rise in frequency of rs1438307 is quite a bit further into the past than the majority of data available. Could the authors add confidence intervals to Figure S2a.44. If this is a major claim, then that figure could serve as a main text figure panel too.

Could the authors provide a more intuitive rationale for why conditioning on ancestry in their particular analysis setup provides additional power to detect selection?

Why did the authors opt for a 4-way mixture model with EHG and CHG, instead of WHG, Anatolia, and Yamnaya?

"in chromosome 18, we recover a selection candidate region spanning SMAD7, which is associated with inflammatory bowel diseases such as Crohn's disease 41–43. Taken together these results suggest that the transition to agriculture imposed a substantial amount of selection for humans to adapt to our new diet and that some diseases observed today in modern societies can likely be understood as a consequence of this selection."

-The link between ancient selection and present-day disease seems overly speculative based on the data presented in this paper.

"However, profound shifts in lifestyle in Eurasian populations during the Holocene, including a change in diet and closer contact with domestic animals, combined with higher mobility and increasing population sizes, are likely drivers for strong selection on loci involved in immune response."

-Maybe, but there is no firm data on this yet. Perhaps the authors could say "have been hypothesized to be likely drivers...."

"These results suggest that large, recurrent CNVs that can lead to several pathologies were present at similar frequencies in the ancient and modern populations included in this study. " Can it really be assumed that the ancient sample is representative of the past frequencies? It seems too much to make conclusions about prevalence.

[Minor points, requested clarifications, typos]

Table S2d.3 West Eurasia.cw_hg spans 40 pages. Can it be reduced?

Abstract:

Page 1: "high contribution of alleles conferring risk of traits related to diabetes." wording

Results/discussion:

Samples and data

Page 3: "Unprecedented sample", "unprecedented details" repeat words

Figure 1

Sampling times and pop split times don't line up, especially 180 generations ago

Maybe add borders to distinguish the different parts, legends etc

Selection on diet-associated loci

Page 5: "settling controversies regarding the timing of this selection" too strong word

Genetic trait reconstruction and the phenotypic legacy of ancient Europeans

Page 13: "help to settle the famous discussion of selection in Europe relating to height" again perhaps too bold a claim

Referee #3 (Remarks to the Author):

This paper presents multiple extremely interesting analyses of patterns of genetic variation across several hundred ancient genomes, shedding new light on how natural selection drove rapid changes in allele frequency at a number of loci across the genome during the evolutionary history of modern Europeans. The methods are innovative, and the results provide new insights into the timing of the onset of natural selection for several mutations that are known to have played an important role in adaptation as human migrated into and across Europe (e.g. FADS, LCT), as well as the identification of new candidate selective sweeps that were previously obscured by the effects of admixture. My criticisms below notwithstanding, it represents a real triumph for aDNA in looking back in time to reconstruct human evolutionary history. I think that many of the analyses and results potentially of interest to the broad Nature readership.

However, there are several aspects of the manuscript that need work. I have one major substantive criticism, as well as some frustration that several aspects of the manuscript simply do not appear ready for publication.

#####

My substantive criticism:

the connection between a particular positive selection signal and a given nearby complex trait association is often not clear. In many cases, the actual evidence of a link is extremely weak or altogether absent, but the manuscript is framed as if such evidence exists.

###

For example, psoriasis is mentioned in the abstract as a phenotype that has a high prevalence, and imply that their results may explain why. As far as I can tell, aside from the supplementary tables the only mention of psoriasis in the paper is in this sentence:

"In contrast, the signal of selection at C2 (rs9267677; $p=9.82e-14$; $s=0.04463$), also found within this sweep, and associated with psoriasis risk in UK Biobank ($p=4.1e-291$; $OR=2.2$), shows a gradual increase in frequency beginning c. 4,000 years ago, before rising more rapidly c. 1,000 years ago."

However, I noticed a nearly identical sentence in the supplement, but the phenotype mentioned there is educational attainment:

"In contrast, the signal of selection at C2 (rs9267677; $p=9.82e-14$; $s=0.04463$), also found within this sweep, and associated with educational attainment, shows a gradual increase in frequency

beginning c. 4,000 years ago, before rising more rapidly c. 1,000 years ago; highlighting the complex temporal dynamics of selection at the HLA locus."

Would the authors also be willing to argue in the abstract that this signal of selection may help explain patterns of variation in educational attainment? The strength of the evidence for either conclusion is basically the same.

###

Another example, not directly related to the main, selective sweep focus of the paper, is in the section titled: "Pathogenic structural variants in ancient vs. modern-day humans".

The authors write:

"RISE586 exhibited a hypoplastic tooth, spondylolysis of the L5 vertebrae, incomplete coalescence of the S1 sacral bone, among other minor skeletal phenotypes. The skeletal phenotypes observed in this individual are relatively common (~10%) in European populations and are not specific to 16p13.1 thus do not indicate strong penetrance of this mutation in RISE586. However, these results do highlight our ability to link putatively pathogenic genotypes to phenotypes in ancient individuals."

I do not see how a pathogenic genotypes has been in any way "linked" to phenotypes in ancient individuals. One ancient individual has phenotypes that are common among other ancient individuals, and also carries the deletion/duplication. This is not a result. It just means that the authors were able to genotype an individual for whom they can also measure skeletal traits. I understand that there is some hope that potentially in the long run this sort of paired data can be used to learn more about the relationship between the genotypes and phenotypes of ancient individuals, but this hasn't actually been done here. I think writing that this mutation is not strongly penetrant in this individual is positively misleading. There is no evidence of ANY penetrance or relationship to the phenotype whatsoever.

(I should also note that it is not clear to me whether the the variant of interest in the above section is a deletion or duplication. In line 388, first it is a duplication ("duplications at 16p13.11"), then later in the same line it is a deletion ("An individual harbouring the 16p13.11 deletion"). Maybe there are both? I can't tell...

In general, I think the paper significantly oversells what the results actually tell us about phenotypic variation. The abstract closes with a sentence that begins: "Our results paint a picture of the combined contributions of migration and selection in shaping the phenotypic landscape of present-day Europeans...". But as I've argued above, there is generally little to no evidence linking the reported results to the "phenotypic landscape" of present-day Europeans. For some of these sweeps it is entirely possible that the phenotypes that drove them are not expressed in the modern human environment. The field has been identifying selection signals physically nearby to trait associations for some time now, and that's what most of them still are: two different signals that are close to one

another in the genome but have not other obvious connection. The signals here seem a lot more likely to be "real" than many earlier ones based on iHS or other similar metrics, but the hard work of determining how these sweeps are related to present day phenotypic variation, if at all, lies in the future. To be clear, I think that pointing out nearby phenotypic association or known functions of sweep candidates is fine, but that the overall packaging of the manuscript as if it sheds serious light on this goes too far.

#####

My second major criticism is that the supplement appears incomplete and has many errors and in many places either does not produce enough detail about the methods used, or has text which doesn't fully track.

For example, there is a paragraph starting on line 187 of the supplement that explains why knowing only the first coalescent event is not sufficient for understanding the full ancestry of a given haplotype. While this is no doubt true, this text is not obviously related to the surrounding text. I can surmise that it may be an explanation of why the tool MSMC (which models the first coalescent event and was developed in senior author Richard Durbin's group) isn't an appropriate tool for this task, but MSMC is never explicitly referenced, nor is the "first coalescence event" mentioned anywhere else in the main text or the supplement.

Relatedly, the authors say they adapted CLUES to model time series data. It's hard to tell from the supplemental text whether they've ADDED time series data on top of the existing functionality that uses inferred ARGs (i.e. using aDNA and ARGs jointly), or if it's just that they've taken the CLUES codebase and basically spun off a different (but obviously related) method that uses a time series of aDNA to infer trajectories. The current descriptions in the supplement are extremely cursory, and I think it would be appropriate for the authors to give a more complete description of the methods as actually used.

At line 3050 in supplementary section 2g, the authors write:

"To calculate an ancestry-specific PRS we used an additive model, including a transformation as in Berg & Coop and in line with (Supplementary Note S2c (Allentoft et al. 2022))"

I can't tell what transformation in Berg & Coop they are referring to. I also checked the Allentoft citation and there does not appear to be a section 2c.

Additional pieces of the supplement that still need some work:

Supplementary text 1a switches back and forth between first person singular and first person plural.

Figure S2c.2, it appears the row labels have been removed, presumably by accident.

For Figure S2c.3, the traits are referenced only by their numbers in the "UK Biobank coding system". I think it is not unreasonable for readers to expect a figure with human readable trait labels on it.

For Figure S2c.6, the figure caption reads "Principal component analysis on West Eurasian samples coloured by individual polygenic scores.", but no part of the figure indicates what trait the polygenic scores are for. Searching the text, it seems like it is for height, but this should be clearly indicated in the figure.

Tables S2d.1 S2d.2 and S2d.3 have no figure captions. I can mostly guess at what the column headings are, but readers shouldn't have to. The supplementary text refers to Tables S2d.1 and S2d.2, but there does not appear to be a reference to Table S2d.3 anywhere in the text.

Note that this list is not exhaustive, and I do not think that the authors merely need to respond to the specific examples I point out. Rather, I think the authors need to take a serious pass through the supplement again, including sections that I do not explicitly note here, and make sure that it is actually ready for publication.

#####

Lastly, I have a few comments on the "polygenic selection" analyses relying on polygenic scores:

1) I could not find the actual quantitative results of this analysis. There are figures in the main text that show the traits that pass a bonferroni multiple testing threshold, and figures in the supplement that show a heatmap of some summaries of the analysis (as noted above, however, these figures are not human readable), but readers should have access to the actual results. Relatedly, there is some basic information about the empirical randomization scheme that I could not find: e.g. how many null replicates were sampled to generate the empirical p values?

2) The statement at line 464 that "these analyses help to settle the famous discussion of selection in Europe relating to height" is far too strong. There are at least three potentially distinct signals of selection on height that have been reported in or near Europe. Field et al 2016 reported a signal of recent selection for increased height in Britain within the last 2000 years, based on analyses using the singleton density score (Howe et al 2022 also supported this result using sibling based effect sizes which are free of confounding). There is also a reported signal of selection for decreased height in Sardinia, supported by effect sizes from the Biobank of Japan (Chen et al 2020). Then, there is the signal that's being reported here, which is similar to one reported by Mathieson et al 2015. I think the authors should be clearer about the broader context and complex history of this particular question.

Author Rebuttals to Initial Comments:

*Referee expertise:*

*Referee #1: human evolutionary genetics*

*Referee #2: aDNA/human evolution*

*Referee #3: population genetics*

*Referees' comments:*

*Referee #1 (Remarks to the Author):*

*This manuscript integrates several hundred new ancient human DNA samples along*
*with previous human aDNA data to build a dataset of ~1600 imputed ancient*
*genomes. It then applies several new computational methods for the inference of*
*admixture proportions and allele frequency changes suggestive of selection. A major*
*strength of the approach is the ability to control for differences in ancestral background*
*in ancient and modern samples. The resulting catalog of evolutionary pressures on*
*European phenotypes over that past 12,000 years suggests that ancient selection and*
*admixture have played a larger role in modern phenotypes than recent local selection.*

**Response:** We thank the reviewer for their constructive feedback, and agree that the
ancestral path decomposition represents a major strength of the paper. We have since
made modifications and improvements to the novel chromosome painting model, to
increase the accuracy of the inference.

*Major Comments:*

*The two paragraphs from lines 123-153 describe the data and fundamental analytic*
*methods used in this study, and as such, they are essential to the plausibility and*
*interpretation of all subsequent results. These approaches include several predictive*
*analyses that integrate new methods and data from this manuscript with recently*
*published methods. While I appreciate the length constraints, extensive*
*supplementary material, and desire to get to the results quickly, more detail on these*
*methods (especially their accuracy and validation) must be provided in the main text.*

**Response:** We agree that length constraints in the main text make it difficult to
provide a complete justification for all of the methods used in the paper; however, we
have added additional details and citations to the manuscript to address these
concerns (detailed below).

*To illustrate this, here are some of the new claims the reader is asked to accept in*
*these paragraphs without any details in main text (and this is assuming that existing*
*methods like RELATE and CLUES are sufficiently accurate):*

*- The ancient genomes are accurately imputed and phased.*

**Response:** We have prepared a new paper which provides comprehensive validation
and benchmarking of the imputation and phasing of the ancient samples used in this
manuscript (da Mota et al. 2022 *bioRxiv*; <https://doi.org/10.1101/2022.07.19.500636>).
The validation of the imputation accuracy is performed using 42 down-sampled high-
coverage ancient genomes, and the validation of the phasing accuracy is performed
using the first ever ancient human trio. For a 1x ancient genome, we estimate an
imputation error rate of 1.9% and a phasing switch error rate of 2.0%, which is
comparable to modern genomes at equivalent coverage.

**Changes:** We have updated the main text to cite the new preprint, and have added
the following summary text.

lines 130-133:

This dataset comprises 1,664 imputed diploid ancient genomes and more than
8.5 million SNPs, with an estimated imputation error rate of 1.9% and a
phasing switch error rate of 2.0% for 1X genomes. Full details of the validation
and benchmarking of the imputation and phasing of this dataset are provided in
reference (da Mota et al. 2022).

- *The new chromosome painting ancestry inference method is accurate at the*
*haplotype level on both ancient and modern genomes.*
- *The simulation framework is not sensitive to misspecification/inaccuracies in the four-*
*population admixture model.*
- *The neural network classifier for ancestral path inference is sufficiently accurate.*

**Response:** We have prepared a second additional paper which comprehensively
describes the validation and benchmarking of an improved version of the novel
chromosome painting method used in our updated aDNA time-series analyses
(Pearson & Durbin 2023 *bioRxiv*; <https://doi.org/10.1101/2023.03.06.529121>). In this
paper we show that our method outperforms a leading alternative, *GNOMix*
(Hilmansson et al. 2021 *bioRxiv*; <https://doi.org/10.1101/2021.09.19.460980>), under
most tested scenarios. Using simulations, we estimate an average accuracy of 94.6%
for the four ancestral paths leading to present-day Europeans. We also show that our
method is robust to a range of simulated demographic scenarios and model
misspecification.

**Changes:** We have updated the main text to cite the new preprint, and have added
the following summary text.

lines 210-213:

Using simulations, we show that our novel chromosome painting method has
an average accuracy of 94.6% for the four ancestral paths leading to present-
171 day Europeans, and is robust to model misspecification. Details of the novel
chromosome painting method used on this dataset are provided in
Supplementary Note 1c, and further described in reference (Pearson and
Durbin 2023).

- *The updates to the CLUES framework enable accurate allele frequency and*
*selection coefficient estimates.*

**Changes:** To validate the accuracy of the updates to the CLUES method, we have
performed a set of new simulations. These simulations show that CLUES accurately
infers selection coefficients and allele frequency trajectories for sample sizes smaller
than those we used in our empirical analyses. We have also updated the way we
convert the log-likelihood ratio statistic into a p-value, which has increased our power
to detect SNPs under selection. Full details of the simulation design and
benchmarking accuracy are described in Supplementary Note 2b.

*(Big kudos for the snakemake and clear github page.)*

**Response:** We believe that open and reproducible methods are critical to the
advancement of science, and thank the reviewer for their acknowledgement.

**Changes:** We have added additional URLs for the chromosome painting and ARS
analysis of the UK Biobank, the implementation of the novel chromosome painting
method and for the demographic model used to train the classifier.

lines 616-622:

The scripts used to run the chromosome painting (Supplementary Note 1b)
and calculate ARS in the UK Biobank (Supplementary Note 2f) are available at
https://github.com/will-camb/mesoneo_selection_paper. The software to
perform the ancestral path chromosome painting described in Supplementary
Note 1c is available on GitHub at [https://github.com/AliPearson/](https://github.com/AliPearson/AncestralPaths)
[AncestralPaths](https://github.com/AliPearson/AncestralPaths), and the demographic model is available in the stdpopsim
library (see [https://popsim-consortium.github.io/stdpopsim-docs/stable/](https://popsim-consortium.github.io/stdpopsim-docs/stable/catalog.html#sec_catalog_homsap_models_ancienteurope_4a21)
[catalog.html#sec_catalog_homsap_models_ancienteurope_4a21](https://popsim-consortium.github.io/stdpopsim-docs/stable/catalog.html#sec_catalog_homsap_models_ancienteurope_4a21)).

- *The control set of SNPs is appropriate.*

**Response:** The selection process for the control SNPs is detailed in Supplementary
Note 2a. Control SNPs were ascertained by selecting all biallelic SNPs within the
imputed dataset and excluding any that fell within +/- 50 kb of a GWAS SNP or a gene
region. Control SNPs were grouped into bins based on their derived allele frequency
(DAF), rounded to the nearest 1%, and paired randomly (without replacement) with
GWAS SNPs in the same chromosome and DAF bin.

Note: we respond to the query about LD pairing below.

**Changes:** We have updated the main text to cross-reference the relevant chapter in
the supplement.

lines 223-224:

An equal number of putatively neutral, frequency-paired variants were used as
a control set (Supplementary Note 2a).

*I found these approaches to generally be reasonable and the details in the*
*supplementary material to be helpful, but the evaluation was often lacking. To illustrate*
*this, I provide a few non-exhaustive examples where more validation/justification are*
*needed.*

**Changes:** We have revised the supplement to improve clarity, and have added
additional analyses, including a validation of the revised CLUES method
(Supplementary Note 2b). We now also cite two additional papers, which
systematically validate and benchmark the imputation pipeline and the novel
chromosome painting method (detailed below).

*In the section on the neural network to predict paths backward in time (S1a), the only*
*evaluation provided is the confusion matrix on simulated data in Figure S1a.2. It*
*seems that there is considerable misclassification even on the simulated data.*
*(However, I note that there is not a scale bar, so I can't really evaluate the magnitude*
*of the values in each box.)*

**Changes:** We apologise for the error in the confusion matrix and have replaced this
figure in the supplement with a more detailed breakdown of the classification
accuracy, showing confusion matrices for five different simulated populations. We
have also updated the manuscript to use a revised and improved version of the
chromosome painting model, which now has an average accuracy of 94.6% for the
four ancestral paths leading to present-day Europeans.

*In S2b, there is no evaluation of the method for inferring allele frequency changes*
*influenced by bias. And there is not a justification for the 0.5 threshold on F_j . Out of*
*the context of the distribution of this metric across sites or its effect on the likelihood of*
*inferring selection, this still seems quite low. Similarly in the most of the supplementary*
*sections I did not find the level of evaluation I expected.*

**Changes:** We have expanded the discussion in the supplement justifying the choice
of threshold for the F_j statistic, and added simulations which show that the F_j statistic
is well calibrated to detect problematic SNPs (see Figure S2c.2).

lines 2932-2948 (Supplement):

The distribution of F_j can be divided into 3 parts, the region $F_j < 0$, maps back to
$-1 < R_j < 0$, the region $0 < F_j \leq 1$ maps back to $R_j > 0$, and the region $F_j > 1$ maps
back to $R_j < -1$. When filtering sites we exclude those sites where the indirect
and direct impacts have the same direction, making it impossible to
differentiate between the two, i.e., $R_j > 0$. Therefore when the filtering is based
on F_j values, we only filter the sites with F_j values within $0 < F_j \leq 1$. When F_j the
value is closer to 1, it becomes harder to distinguish the confounding signal
from the true temporal signal (e.g., selection), so we use a cutoff c to remove
the SNPs where it is unworkable to differentiate them. The value of F_j is used
to keep the sites which we believe to be under (relatively large) selection and
screen the rest, while the numerator and denominator of R_j , as they reflect the
absolute strengths of the temporal and confounding signals, are more suitable

for detecting unknown selection signals upon sites. To filter out sites in
selection analyses that may be affected by biases, we use a fixed threshold of
$0.5 < F_j \leq 1$. Choosing a filtering threshold of F_j values above empirical threshold
0.5 removes those sites where indirect effects mediated through the ancient
DNA characteristics (biases caused by ancient characteristics) are greater than
the true change of allele frequency measured as the direct effect. Their change
of frequencies are more likely caused by ancient signals rather than selections.

*Following on this, the bigger point is that it is challenging to evaluate the sensitivity of*
*the results to inaccuracies in each of the modeling and predictive steps listed above.*
*Given that the results of one analysis are often used as inputs to the next, I fear the*
*potential for errors and propagation. Anything the authors can do to better understand*
*this would be extremely valuable in establishing confidence in the results.*

**Response:** To build confidence in the robustness of our analyses, we have performed
additional benchmarking and validation for the phasing and imputation (da Mota et al.
2022 *bioRxiv*; <https://doi.org/10.1101/2022.07.19.500636>), the ancestral path
chromosome painting (Pearson & Durbin 2023 *bioRxiv*;
<https://doi.org/10.1101/2023.03.06.529121>), and the inference of allele frequency
trajectories and selection coefficients using the modified version of CLUES
(Supplementary Note 2b). These new analyses show that the overall rate of error in
each analysis step is low and that accuracy outperforms comparable approaches.
Furthermore, to mitigate the potential of bias caused by the propagation of errors from
one analysis step to another, we have implemented several additional controls.

To test for potential effects of imputation bias on our selection analyses, we ran
additional models for each of the top SNPs in the pan-ancestry analysis, using
genotype likelihoods (GL) called directly from the aDNA sequencing reads. Because
our chromosome painting model requires phased haplotypes, this replication test was
limited to pan-ancestry models only. When comparing the imputed and GL selection
models, we observed that the posterior likelihood densities of the allele frequency
trajectories are highly correlated, as are the selection coefficients. Due to the smaller
sample sizes in the GL callset we had less power to reject neutrality, and the inferred
log-likelihood ratio test statistics were consistently lower than in the imputed models,
but all remained high. Overall, we did not detect any substantive bias when comparing
these two sets of models.

To test for potential effects of painting bias on our selection analyses, we performed
neutral simulations using the same demographic model used to train the classifier
(Supplementary Note 2a). We applied the chromosome painting model to the
simulated VCF, and used CLUES to infer allele frequency trajectories and selection
coefficients for frequency paired simulated SNPs, in both a pan-ancestry analysis and
stratified by each of the four ancestral paths. We then applied the same thresholds
used in the empirical analysis to detect selective sweep loci. We observed zero
genome-wide significant selective sweep loci across all five analyses. At a SNP level
we observed 22 false-positive SNPs (0.75%) across all analyses (Supplementary
Figure S2a.26). Due to the stochastic occurrence of false-positive SNPs along the
chromosome, no false-positive sweep loci were detected. Detection of a false-positive

sweep locus would require a cluster of at least 6 genome-wide significant SNPs within
a +/- 1 Mb locus. From this analysis we conclude that the low rate of error in our
chromosome painting model is unlikely to bias the inference of sweep loci, but may
produce a small number of randomly occurring false-positive SNPs.

To control for errors specific to ancient DNA damage, we developed two new quality-
control metrics for filtering sites with evidence of potential bias. Firstly, we developed a
novel statistic (F_3) for detecting correlations between characteristics of aDNA damage
(i.e., depth of coverage, read length and error rate) and changes in allele frequencies
over time (Supplementary Note 2c). The purpose of the F_3 statistic is to identify SNPs
where the observed time-series of aDNA genotypes may be biased by age
dependent preservation characteristics. We also developed a second metric, intended
to detect reference and mapping bias when analysing ancient and modern DNA
together (Supplementary Note 2a). Due to the characteristics of aDNA damage, some
sites may be enriched for mapping bias, which favours observations of the reference
allele. This can result in systematic differences between allele frequencies calculated
from aDNA and modern data. To control for this, we developed a test to filter out sites
exhibiting substantial differences between present-day allele frequencies inferred from
ancient and modern data respectively.

Finally, to test for potential effects from unmodeled phenomena, we ascertained a set
of putatively neutral "control" SNPs. These SNPs were drawn at random from regions
of the genome at least 50 kb from a GWAS SNP or gene region, and frequency paired
with each GWAS SNP based on their derived allele frequency (DAF). We then ran the
Control SNPs through the same selection pipeline as the GWAS SNPs. Our ancestry
stratified results detected 19 genome-wide significant selective sweeps in the GWAS
group, and 2 in the Control group. Upon further investigation, one of the two sweeps
identified in the Control group (chr19:57.5-57.5 Mb) contains genome-wide significant
SNPs that were not reported in the GWAS Catalog, but are reported in the UK
Biobank (i.e., rs959939, rs2102540, rs56830277 and rs12978492 for phenotype code
'20024_1121'). Interpretation of the remaining sweep is less clear, as it is entirely
possible that a non-GWAS locus may be a target of selection. Overall, these results
indicate that error propagation between analysis steps is not a major source of bias,
as the Control SNPs are themselves subject to the same phasing, imputation, painting
and selection analyses as the GWAS SNPs and yet we find a 20-to-1 ratio of sweep
loci that contain significant GWAS trait associations.

**Changes:** We have added citations in the main text to the two new manuscripts which
describe the validation and benchmarking of our imputation and chromosome painting
approaches in more detail. We have also added a new chapter to the supplement
detailing a comprehensive set of simulations to benchmark the modifications to our
CLUES method (Supplementary Note 2b).

*I appreciated the comparison to the control group in the GWAS variant selection*
*analyses, but I have two questions about this. First, isn't it important to match the*
*control SNPs on LD as well as MAF, since it is likely associated both with the*

*probability that a variant is a GWAS hit and experienced selection? This is commonly*
*done in tools like SNPSNAP.*

**Response:** When conducting an enrichment analysis for a genome-wide association
study it is important to pair SNPs using linkage disequilibrium to properly calibrate
background expectations. However, our use-case for ascertaining a set of control
SNPs is different, as we are specifically looking for evidence of selection, rather than
evidence of enrichment for particular biological annotations. We chose not to pair our
control SNPs using the flanking patterns of LD because the strength and extent of LD
is directly influenced by selection (i.e., LD is a dependent variable in many selection
tests). Our concern is that if we LD-paired our control SNPs we would be enriching for
sites with evidence of selection, due to the high occurrence of selection in the GWAS
set.

*Second, the finding of many more selection peaks when conditioning on ancestry is*
*interesting. However, there was not any evaluation via simulations of the power to*
*detect different types of selective events when considering ancestry or not. Couldn't*
*this just be due to an increase in power when considering ancestry?*

**Response:** Our analysis suggests that there are several reasons why we detect more
selection when conditioning on ancestry; one of which is increased statistical power. In
cases where the selected allele is not segregating in all ancestral backgrounds (e.g., if
it is private to one ancestral path), stratification by ancestry increases our statistical
power to detect selection, as it allows us to separate haplotypes in which the selected
allele is absent (i.e., where selection can have no observable effect). Similarly, in
cases where selection is influenced by epistasis, stratification by ancestry may allow
391 us to separate haplotypes that contain only a subset of the adaptive markers.

However, the main reason we detect more selection is due to the effects of multiple
waves of admixture. Our pan-ancestry analysis spans three major waves of admixture
(Figure 3), which coincide with dramatic changes in subsistence strategy, as well as
large movements of people into new environmental niches. In cases where selection is
acting in only one population, admixture can confound analyses based on time-series
data, especially when the admixing populations have substantially different allele
frequencies. Stratifying by ancestry controls for this effect, as it allows us to model
changes in allele frequency independently of changes in admixture fraction.

*Given the challenges of porting polygenic risk scores across even closely related*
*populations, I was very surprised to see the attempts at PRS-based ancestral trait*
*reconstruction. The authors are aware of these challenges and repeatedly suggest*
*“caution” in the interpretation. This is insufficient as no work has been done to*
*evaluate the feasibility or accuracy of this analysis. Given the known challenges and*
*lack of specific hypotheses guiding these analyses, their value to the manuscript is not*
*clear. I would suggest removing these if stronger justification cannot be provided.*
*(Also, PMID: 29285967 should be cited.)*

**Response:** We have accepted this recommendation, and removed the PRS-based
trait reconstruction analyses from the manuscript. Whilst some simulation studies have

been published on this topic (Carlson et al. 2022 *PLOS Genetics*;
<https://doi.org/10.1371/journal.pgen.1010170>; Yair & Coop 2022 *Proc. R. Soc. B*;
<https://doi.org/10.1098/rstb.2020.0416>), and two papers have attempted empirical
validation (Cox et al. 2021 *AJPA*; <https://doi.org/10.1002/ajpa.24426>; Marciniak et al.
2022 *PNAS*; <https://doi.org/10.1073/pnas.2106743119>), we agree that additional
simulations would be beneficial for estimating the loss of predictive accuracy in
populations that are only partially ancestral to the GWAS cohort. However, a new
simulation study is outside the scope of this already large manuscript, so we have
elected to remove this analysis instead.

**Changes:** We have updated the main text to remove these results and have deleted
the Supplementary Notes titled “Over-dispersion in polygenic scores across ancient
populations” and “Correlation between components of variation in population structure
and components of variation in SNP-trait association”.

*That said, the estimation of the contributions of different ancestral populations to*
*variation in phenotypes in the UK Biobank seems on stronger methodological footing.*
*However, these results are not presented in any detail except to say that they point to*
*a way forward for disentangling ancestry contributions to differences in genetic*
*disease risk (L460). I suggest expanding the presentation of these results instead of*
*the ancient phenotype prediction.*

**Response:** We agree that the ancestral risk scores calculated from the chromosome
painting of the UK Biobank provides a robust estimate of the phenotypic legacy of
these ancestral populations, as it avoids the issue of portability when directly
predicting polygenic scores in ancient individuals.

**Changes:** We have greatly expanded the presentation of these results in the main
text, under the subheading “The phenotypic legacy of ancient Eurasians” on line 497.

*I would also like to see more direct discussion of how the results relate to those of a*
*few recent similar studies. For example:*

*doi: <https://doi.org/10.1101/2022.07.02.498543> and*

*<https://doi.org/10.1101/2022.08.24.505188> both have traced selection over the past*
*10,000 years using different methods.*

*PMID: 36316412 argued that admixture can hide selective events.*

**Response:** We did not cite the papers by either Lee et al. (2022 *bioRxiv*;
<https://doi.org/10.1101/2022.08.24.505188>) or Kerner et al. (2023 *Cell Genomics*;
<https://doi.org/10.1016/j.xgen.2022.100248>) in the previous version of our manuscript
because both these papers post-date our initial submission in May 2022 (prior to the
splitting up of the original manuscript into more focused papers). In both cases, it is
difficult to directly compare our results, due to substantial differences in methodology,
sampling and reporting.

Lee et al. (2022) used pseudohaploid data from the 1240k capture array, so their true
sample size (measured in count of genotypes) is considerably less than their reported
1,291 individuals, which they further subdivide into three time periods. For example, in

our analysis of selection at the LCT locus using the 1240k dataset (Supplementary
Figure S2a.56), we observed that there were 838 pseudohaploid calls for rs4988235,
but only 476 for rs1438307, when using the same 1,291 samples as Lee et al. (2022).
We explicitly chose not to subdivide our selection scan of 1,015 diploid ancient
genomes into multiple epochs, due to the risk of overfitting to small sample sizes. The
Let et al. selection scan is an updated version of the mixture model used in Mathieson
et al. (2015, *Nature*; <https://doi.org/10.1038/nature16152>) which relies on differences
in allele frequencies postdating admixture. As such, they are best powered to detect
rapid episodes of selection following admixture between populations with large
differences in allele frequencies. In our analyses, we used local ancestry inference to
identify selection signals in a manner that is independent of changes in admixture
proportions, so we are well-powered to detect selection in a much broader range of
demographic scenarios. We also have reservations about the use of the 1240k
capture array data to detect selection, due to well-established allelic bias caused by
the capture chemistry (Rohland et al., *bioRxiv*;
<https://doi.org/10.1101/2022.01.13.476259>).

Kerner et al. (2023) also used pseudohaploid data from the 1240k capture array, but
they took additional quality control steps to filter their results based on a comparison to
shotgun sequenced data. Strikingly they found that nine of the top 10 variants in their
capture dataset had a frequency trajectory inconsistent with their shotgun dataset;
indicating systematic problems with using 1240k capture data for selection scans. We
also note that the selection test used by Kerner et al. (2023) is based on choosing
variants with an estimated selection coefficient above the 99th quantile from their
simulations, and is therefore only powered to detect cases of extremely high selection.
One of the novel findings from our study is that when you characterise a locus based
on the SNP with the most extreme divergence from neutrality (e.g., rs4988235 at the
LCT locus) you can easily overlook a much longer history of selection at that locus.

**Changes:** We have added a citation to PMID 36316412.

lines 245-247:

This suggests that admixture between ancestral populations has masked
evidence of selection at many trait associated loci in Eurasian populations
(Souilmi et al. 2022).

*Minor Comments:*

*While I appreciate that the details of the construction of the cohort are provided in a*
*companion paper, a few more sentences and perhaps a figure panel describing the*
*geographic locations and ages would be helpful.*

**Changes:** We have added a new Figure 1, which shows sampling locations and ages
of the West Eurasian samples used in our aDNA time-series selection analysis, as
well as five density plots of the sample ages, grouped by sampling region.

*Tone down exaggerated statements. For example, on L111 and L119, is this dataset*
*truly “unprecedented”?* *While this is a wonderful dataset, by now many studies have*
*analyzed hundreds of ancient individuals’ genomes. Thus, I disagree that it is*
*unprecedented.*

**Changes:** We have removed all usage of the word “unprecedented”

lines 128-130:

Our analyses are undertaken on the largest collection of shotgun-sequenced
ancient genomes published to date; presented in the accompanying study
‘Population Genomics of Stone Age Eurasia’ (Allentoft et al. 2022).

lines 139-141:

This dataset allows us to characterise in fine detail the changes in selective
pressures exerted by major transitions in human culture and environment.

lines: 435-438

Additionally, our results provide detailed information about the duration and
geographic spread of these processes (Fig. 4) suggesting that an allele
associated with lighter skin was selected for repeatedly, probably as a
consequence of similar environmental pressures occurring at different times in
different regions.

*Similarly, I thought the insights into the timing and different variants potentially*
*involved in selection at the LCT/MCM6 locus were fascinating. But it is not clear to me*
*that this analysis will completely settle the “controversies regarding the timing of this*
*selection” on lactose digestion (L172).*

**Response:** To support our novel result that a microRNA variant near the LCT locus
has been under selection for thousands of years prior to the emergence of the lactase
persistence allele we replicated our analysis using genotypes from the 1240k capture
array, downloaded from v52.2 of the Allen Ancient DNA Resource. We limited our
analysis to the 1,291 West Eurasian samples used by Le et al. (2022 *bioRxiv*;
<https://doi.org/10.1101/2022.08.24.505188>), and binned genotypes into one-thousand-
540 year bins, then plotted a weighted loess regression (Supplementary Figure S2a.56.).
This analysis independently replicates our finding of earlier selection at the microRNA
variant, using a different set of samples, genotyped with a different sequencing
technology, and without relying on either imputation or chromosome painting.

**Changes:** We have reworded the main text to remove the claim that our results settle
the controversies of the timing of this selection.

lines 250-252:

We find strong changes in selection associated with lactose digestion after the
introduction of farming, but prior to the expansion of the Steppe pastoralists
into Europe around 5,000 years ago (Allentoft et al. 2015; Haak et al. 2015),
the timing of which is a long standing controversy (Enattah et al. 2008; Itan et
al. 2009; Ségurel and Bon 2017; Segurel et al. 2020).

And again, in the discussion of height and selection (L464-470), this study adds valuable new data and hypotheses, but I am not convinced that it is “settled” (and I don’t have a stake in this debate) so I would reframe this section

Changes: We have reworded this sentence.

lines 554-559:

These results also help to clarify the famous discussion of selection in Europe relating to height (Mathieson et al. 2015; Cox et al. 2019; Rosenstock et al. 2019). Our finding that the ‘Steppe’ ancestral components (Yamnaya/EHG) have consistently high genetic values for height in the UK Biobank demonstrates that height differences between Northern and Southern Europe may be a consequence of differential ancestry, rather than selection, as claimed in many previous studies (Field et al. 2016). However, our results do not preclude the possibility that height has been selected for in specific populations (Chen et al. 2020; Howe et al. 2022).

The sweep “loci” seem extreme large from several of the examples given: multiple Mb for most and 33 Mb for HLA. Can the authors comment on whether this is a resolution issue or likely to reflect selection on multiple variants within these windows (as would be expected for a locus like the HLA)?

Response: Regarding the reported sizes of the sweep loci, we have changed the way that these are calculated, and now report smaller loci for the majority of the sweeps. Previously we were using the software Manhattan Harvester (Haller et al. 2019 *BMC Bioinformatics*; <https://doi.org/10.1186/s12859-019-2600-4>), which was reporting the very wide loci. We have now updated our approach to use a hierarchical clustering algorithm, with a maximum branch length of 1Mb. This results in more compact loci, but with fewer genome-wide significant SNPs (Supplementary Note 2a).

In the case of the HLA, our results indicate that this locus has been subject to multiple independent sweeps, occurring at different times and with differing intensities (line: 326). We further explore the complex pattern of ancestry specific selection at the HLA in our companion paper, “Elevated genetic risk for Multiple Sclerosis originated in Steppe Pastoralist populations” (Barrie et al. 2022 *bioRxiv*; <https://doi.org/10.1101/2022.09.23.509097>). In that paper, we show that polygenic selection at the HLA locus has increased genetic risk for multiple sclerosis and reduced genetic risk for rheumatoid arthritis. These polygenic selection signals are principally centred around the HLA locus, are independent of each other, and occur at strikingly different times over the last 13,000 years.

Our results also suggest that multiple targets of selection are more common in genome-wide sweep loci than previously thought. We observe multiple cases where the most significant SNP in a sweep locus varies between ancestral backgrounds, consistent with selection favouring more than one haplotype (although small differences could also be attributed to error in the painting model). We further observe that sweep loci shared across ancestries are often only partially overlapping. For the

602 ancestry stratified analysis, the reported boundaries of each sweep locus is obtained
by merging overlapping loci inferred in each marginal ancestry. As such, flanking
regions of these merged loci may only be genome-wide significant in a subset of the
ancestries where we detect that sweep. Lastly, we also detect cases where the
inferred timing of selection differs substantially among genome-wide significant SNPs
within the locus (e.g., the lactase persistence allele and the microRNA variant
discussed in the main text).

**Changes:** We have updated the main text to report the smaller sweep loci, and have
added the following citation to our new preprint.

lines 349-351:

We further explore the complex pattern of ancestry specific selection at the
HLA locus in our companion paper, “Elevated genetic risk for Multiple Sclerosis
originated in Steppe Pastoralist populations” (Barrie et al. 2022).

*At several points a “population structure axis separating” populations is referred to with*
*a reference to Figure 2. I believe that this is in error. Perhaps it should be Figure 4A?*
*Also, the variance explained by the PCs is incredibly small.*

**Changes:** We have removed this passage of text and Figure 4A.

*Figure 1: I do not see a blue line for pop_4. But perhaps it is not supposed to appear*
*since the numbering starts with 0?*

**Changes:** We have updated this figure with a new version that reflects the updated
model used in our revised manuscript.

*I suspect that this is a PDF conversion issue, but all the figures are blurry.*

**Response:** The blurriness of the figures was caused by a PDF conversion issue
during the upload to the submission system. All figures in the submitted version were
of high resolution.

*Referee #2 (Remarks to the Author):*

*Irving-Pease et al. report analysis searching for natural selection in a large set of*
*>1,600 imputed shotgun genomes, many of which were produced in a "main paper"*
*(Allentoft et al.) to which this paper is a companion paper. They take approaches*
*including inference of selection coefficients on genealogies, decomposition of*
*evolution in different ancestral populations, evolution of polygenic traits and evidence*
*for adaptation on those, and searching for structural variants with evidence of*
*pathogenicity.*

*The paper is well-written, and several approaches and analysis here are highly*
*interesting and definitely push the field of studying natural selection with ancient DNA*
*forward (for example, the ancestry-decomposed inference of selection coefficients),*
*but it is less clear which new biological insights are learned, and there are several*
*issues that need mention and clarification.*

**Response:** We thank the reviewer for their assessment that our paper is highly
interesting and pushes the field forward. To clarify which new biological insights have
been learned, we have made several changes to the main text (detailed below).

*The paper consists of hundreds, perhaps thousands, of claims of selection on single*
*variants and traits, each caveated largely appropriately, but the study is thus different*
*from most papers focusing on presenting tight waterproof evidence for a handful of*
*central claims. Reading the paper, I can see a few claims that I am guessing that the*
*authors are highly confident in, but also others where I am unsure of what level of*
*confidence there actually is.*

*Of the major claims in the abstract:*

*-Selection on metabolism, and HLA seems to have been reported previously (e.g.*
*Mathieson et al. 2015, Nature). The link between immune selection and autoimmune*
*disease is entirely speculation, no new advance on the possible link is reported.*

**Response:** We have prepared a new companion paper titled "Elevated genetic risk for
Multiple Sclerosis originated in Steppe Pastoralist populations" (Barrie et al. 2022
*bioRxiv*; <https://doi.org/10.1101/2022.09.23.509097>), in which we formally test for
polygenic selection for two autoimmune diseases. In that paper, we show that
statistically significant polygenic selection at the HLA locus has increased genetic risk
for multiple sclerosis and reduced genetic risk for rheumatoid arthritis. These
polygenic selection signals are principally centred around the HLA locus, are
independent of each other, happen at strikingly different times over the last 13,000
676 years, and occur on different ancestral backgrounds.

Regarding the metabolism results, we report two major advances. Firstly, we show
that at the FADS locus much of the selection associated with a more vegetarian diet
occurred in Neolithic populations before they arrived in Europe, then continued during
the Neolithic, contrary to previous reports. Secondly, at the LCT locus we show that
selection predates the emergence of the lactase persistence (LP) allele by thousands

of years, and appears to be favouring a microRNA variant (rs1438307) with strikingly
different metabolic effects. We further show that the high LD between the LP allele
and the microRNA variant may explain the recently observed correlation between
frequency rises in the LP allele and archaeological proxies for famine and increased
pathogen exposure (Evershed et al. 2022; *Nature*; <https://doi.org/10.1038/s41586-022-05010-7>).

**Changes:** We have updated the main text to cite our new preprint, and have reworded
the results in this manuscript to make it clearer when we are hypothesising about a
connection to autoimmune disease.

lines 349-351:

We further explore the complex pattern of ancestry specific selection at the
HLA locus in our companion paper, "Elevated genetic risk for Multiple Sclerosis
originated in Steppe Pastoralist populations" (Barrie et al. 2022).

*-Selection at the FADS cluster and the lactase persistence locus began earlier than
previously thought. This seems like novel claims, and the evidence seems strong. At
the same time, saying that the debate is "settled" is probably premature. This claim
could be supported by additional evidence, how robust is the new timing proposed by
the authors, and exactly what timings had been suggested before?*

**Response:** To support our novel result that a microRNA variant near the LCT locus
has been under selection for thousands of years prior to the emergence of the lactase
persistence allele we replicated our analysis using genotypes from the 1240k capture
array, downloaded from v52.2 of the Allen Ancient DNA Resource. We limited our
analysis to the 1,291 West Eurasian samples used by Le et al. (2022 *bioRxiv*;
<https://doi.org/10.1101/2022.08.24.505188>), and binned genotypes into one-thousand-
711 year bins, then plotted a weighted loess regression (Supplementary Figure S2a.56.).
This analysis independently replicates our finding of earlier selection at the microRNA
variant, using a different set of samples, genotyped with a different sequencing
technology, and without relying on either imputation or chromosome painting.

**Changes:** We have reworded the main text to remove the claim that our results settle
the controversies of the timing of this selection.

lines 250-252:

We find strong changes in selection associated with lactose digestion after the
introduction of farming, but prior to the expansion of the Steppe pastoralists
into Europe around 5,000 years ago (Allentoft et al. 2015; Haak et al. 2015),
the timing of which is a long standing controversy (Enattah et al. 2008; Itan et
al. 2009; Séguérel and Bon 2017; Segurel et al. 2020).

*--Differential genetic contributions in height ancestral to present-day Europeans.
Yamnaya introduced tall height. This seems largely in line with previous finds (e.g. by
Mathieson et al. 2015; Cox and Mathieson 2020, PNAS).*

**Response:** We agree that this is broadly in line with the previous studies cited by the
reviewer. These results can therefore be seen firstly as a validation that our ancestral
risk score (ARS) analysis is consistent with prior findings for a phenotype which is well
studied. The novelty of this analysis is that nobody has previously directly quantified
the impact of Yamnaya ancestry on height in modern populations, which the ARS
does. Instead of merely inferring that the genetically taller Yamnaya introduced height
increasing alleles into modern populations, we have used local ancestry inference to
explicitly quantify this, and calculated ancestry specific polygenic risk scores.

*--"Alleles associated with increased risk of some mood-related phenotypes are*
*overrepresented in the farmer ancestry component." "Western hunter-gatherers show*
*a strikingly high contribution of alleles conferring risk of traits related to diabetes. "*
*These seem like novel claims, but are the authors confident of the claims, or do they*
*think it could be subject to the caveats about projecting GWAS scores from present-*
*day panels into the past that they bring up in the text? If they want present this as a*
*significant scientific advance, can they expand on the evidence for this, and the*
*robusticity of the analysis. How outlying are the scores for these?*

**Response:** We have removed the section of the manuscript in which we reported
polygenic scores for ancient individuals, due to concerns about the portability of
present-day GWAS effect sizes to populations that are only partially ancestral to the
GWAS cohort. However, our ancestral risk score (ARS) analysis avoids the issue of
portability entirely, by calculating the genetic risk that a modern individual would
possess if they were composed entirely of one ancient ancestry. We believe that these
results are robust, and have expanded our discussion of the ARS analysis in the main
text.

**Changes:** We have updated the main text to make it clearer that this analysis is not
affected by the portability issue.

Lines 500-508:

We calculated ancestry-specific polygenic risk scores—hereafter ancestral risk
scores (ARS)—based on chromosome painting of >400,000 UKB genomes
using ChromoPainter (Lawson et al. 2012) (Fig. 6, Supplementary Note 2f).
This allowed us to identify which ancient ancestry components are over-
represented in present-day UK populations at loci significantly associated with
a given trait, and is analogous to the genetic risk that a modern individual
would possess if they were composed entirely of one ancestry. This analysis
avoids issues with the portability of polygenic risk scores between populations
(Martin et al. 2017), as our ancestral risk scores are calculated from the same
individuals used to estimate the effect sizes.

*--"a combination of ancient selection and migration, rather than recent local selection,*
*is the primary driver of present-day phenotypic differences in Europe." To me this*
*seems to have been the consensus for some time. Have recent papers suggested it*
*was due to local selection?*

**Response:** Several recent papers have suggested local selection as a driver of
present-day phenotypic differences. For example, Chen et al. (2020 *AJHG*
<https://doi.org/10.1016/j.ajhg.2020.05.014>) reports evidence for local selection for
reduced height in Sardinians, and Howe et al. (2022 *Nature Genetics*;
<https://doi.org/10.1038/s41588-022-01062-7>) reports evidence for recent selection for
increased height, increased number of children, and reduced HDL-cholesterol.

*There is an interesting find of natural selection on an inversion and duplication of*
*KANSL1, but this is not mentioned in the abstract.*

**Response:** Our evidence of selection at the *KANSL1* locus is complicated. We find
the *KANSL1* duplications to be present in elevated frequencies in some of our earliest
samples, suggesting that selection may predate the time resolution of our study. We
also detect a recent selective sweep which straddles the inversion and *KANSL1*
duplications, but as we note in the main text, this region is also enriched for evidence
of reference bias in our dataset, due to the complex structural polymorphisms which
affect short-read mapping.

*Overall, a lot of the novel results seem to be claims about different timing of selection,*
*than what has been suggested in previous paper. However, the authors don't really*
*provide extensive simulation results or other validation experiments on how robust*
*their timing inference is.*

**Response:** To validate the accuracy of the updates to the CLUES method, we have
performed a set of new simulations. These simulations show that CLUES accurately
infers selection coefficients and allele frequency trajectories for sample sizes smaller
than used in our empirical analyses. Full details of the simulation design and
benchmarking accuracy are described in Supplementary Note 2b.

*Overall, it is difficult to disentangle the scientific advance of the paper from that of the*
*data that it presents the first selection analyses on, but were really produced by*
*another paper (which will be cited for the data itself). That said, this paper has some*
*very interesting analyses going, in some ways more interesting than the advances*
*presented in the parallel submission with the new data.*

**Response:** We have further revised both the "main" paper and this manuscript to
make the separation between the two papers clearer. These were originally submitted
as one paper, which we were asked to split up by the editor. We believe this selection
manuscript represents a substantial scientific advance that is independent of the data
generated in the main paper. Our ancestry stratified time-series selection analysis,
chromosome painting of ancient and present-day populations, and ancestral risk score
analysis of the UK Biobank represent novel methodological approaches that advance
the field of ancient DNA. Furthermore, the results stemming from these analyses make
substantial contributions to our understanding of the strength and timing of selection at
key dietary and immune loci, as well as characterising how differential ancestry has
affected present-day anthropometric and disease traits in the British population.

*Major issues:*

*The authors conduct their main selection analysis (CLUES) not on the whole genome,*
*but on a selected set of 33k SNPs from the GWAS catalog. They then match those*
*with a neutral set. This seems like a sensible way to test if those SNPs have been*
*subject to selection, but the caveat is that when they then dissect the selection peaks*
*in the data, the actually targeted variant may lay somewhere nearby in the genome.*
*There is thus an extra risk that when they discuss variants subject to selection and*
*associated with particular traits in the text, the natural selection was not on those*
*traits. I presume that this was not done genome-wide since CLUES is not easily*
*scaled to such data, but this particular approach seems to increase the risk of*
*storytelling, as there will always be a well-documented GWAS SNP at the height of*
*every signal peak. Could the authors provide convincing evidence that those top SNPs*
*are in fact the SNPs targeted by selection?*

**Response:** Establishing causality between a selection signal and a particular
phenotype is extremely difficult, if not impossible, outside of an experimental evolution
study. We agree with the reviewer that it is entirely possible that the variants directly
targeted by selection may lay somewhere nearby in the genome. However, this
problem would also be present in a genome-wide scan of all SNPs, as the truly
adaptive variant may be an INDEL or structural variant in LD with a nearby SNP.
Furthermore, it is entirely plausible that the truly adaptive phenotypes in recent human
evolution are not well characterised in GWAS. Even in the case of putatively
monogenic loci, establishing causality is not straightforward. For example, our results
reveal strong evidence of at least two sweeps at the LCT/MCM6 locus, containing
variants with strikingly different metabolic phenotypes. Nevertheless, our study has not
sought to establish causality between a selection signal and a phenotype, and there is
no need to directly invoke causality to link directional changes in allele frequencies to
present-day phenotypic variation. In cases where we have strong evidence that a trait
associated variant has changed in frequency, those changes will have affected
present-day expression of that trait, regardless of the causal phenotype that drove the
selective sweep.

**Changes:** To avoid any implication of causality between our selection analysis and
our reporting of trait associations, we have moderated the language used when
referring to trait associations, and have added further caveats to the discussion.

lines 566-570:

Due to the highly pleiotropic nature of each sweep region, it is difficult to ascribe
causal factors to any of our selection signals. However, our results show that
selection during the Holocene has had a substantial impact on present-day genetic
disease risk, as well as the distribution of genetic factors affecting metabolic and
anthropometric traits.

*Imputation seems key to the conclusion in the paper, could the authors discuss a bit*
*more about how the results could be robust to imputation?*

**Response:** We have prepared a new paper which provides comprehensive validation
and benchmarking of the imputation and phasing of the ancient samples used in this
manuscript (da Mota et al. 2022 *bioRxiv*; <https://doi.org/10.1101/2022.07.19.500636>).
The validation of the imputation accuracy is performed using 42 down-sampled high-
coverage ancient genomes, and the validation of the phasing accuracy is performed
using the first ever human aDNA trio. For a 1x ancient genome, we estimate an
imputation error rate of 1.9% and a phasing switch error rate of 2.0%, which is
comparable to modern genomes at equivalent coverage.

To test for potential effects of imputation bias on our selection analyses, we ran
additional models for each of the top SNPs in the pan-ancestry analysis, using
genotype likelihoods (GL) called directly from the aDNA sequencing reads. Because
our chromosome painting model requires phased haplotypes, this replication test was
limited to pan-ancestry models only. When comparing the imputed and GL selection
models, we observed that the posterior likelihood densities of the allele frequency
trajectories are highly correlated, as are the selection coefficients. Due to the smaller
sample sizes in the GL callset we have less power to reject neutrality, and the inferred
log-likelihood ratio test statistics were consistently lower than in the imputed models,
but all remained high. Overall, we did not detect any substantive bias when comparing
these two sets of models.

**Changes:** We have updated the main text to cite the new preprint, and have added
the following summary text.

lines 130-133:

This dataset comprises 1,664 imputed diploid ancient genomes and more than
8.5 million SNPs, with an estimated imputation error rate of 1.9% and a
phasing switch error rate of 2.0% for 1X genomes. Full details of the validation
and benchmarking of the imputation and phasing of this dataset are provided in
reference (da Mota et al. 2022).

*The authors also seem to discuss one primary trait association for each SNP. Are any*
*of the SNPs discussed in the main text associated with multiple traits?*

**Changes:** Due to the pleiotropic nature of the human genome, many of the SNPs with
evidence of selection have more than one association reported in the GWAS Catalog.
We have updated the main text to include additional associations of the named
variants, including associations from FinnGen and UK Biobank which are not reported
in the GWAS Catalog. We provide a full list of all GWAS Catalog associations in
Supplementary Note 2a and in Supplementary Table S2a.3.

*The CLUES inferred trajectories seem highly constrained, but to which degree is this*
*due to the heatmap colour scheme used in the supplementary figures. Could they*
*colour values down to posterior probability ~0.05 more clearly? Also, these trajectories*
*take the imputation and genealogical inference for granted, and thus do not portray*
*the uncertainty associated with those.*

**Response:** The new CLUES benchmarking simulations (Supplementary Note 2b) show that
the width of the allele frequency posterior trajectory is a function of sampling density and
effective population size. In particular, as sampling density increases, the posterior density
becomes more concentrated, because the model has more information about the true allele
frequency trajectory. These simulations further show that even in cases where drift is very
high (i.e., $N_e=1,000$), a sampling density of 250 diploids, sampled over 500 generations, is
sufficient for the true allele frequency to mostly fall within the 95% bound of the posterior
interval. The main reason why our CLUES trajectories appear highly constrained is because
our imputed dataset contains a very high sampling density of 1,015 diploids sampled over
529 generations. With regards to uncertainty from the imputation, this is taken into account
in the model, as imputed genotype-probabilities are used as input into CLUES, rather than
hard-called genotypes. Uncertainty in genealogical inference is also taken into account in
CLUES through the importance sampling framework described in the original CLUES paper,
although inferred genealogies were not used in the aDNA analysis and thus do not
contribute to the inferred trajectory plots.

*On page 34 in the supplement, they report that in the CLUES analysis of aDNA with*
*Ancestral Paintings, they identify quite substantial numbers of outliers also in the*
*control set of SNPs, 346 in the GWAS group and 63 in the Control group. This doesn't*
*seem to be mentioned in the main text.*

**Response:** We initially chose not to report the individual number of SNPs that achieve
genome-wide significance in the main text, for either the GWAS or the Control groups,
because we apply a secondary filtering step to detect clusters of significant SNPs
which are consistent with a selective sweep. Reporting these clusters gives a more
accurate indication of the number of independent selection signals detected, as the
density of SNPs varies across the genome, and because we require at least 6
genome-wide significant SNPs to call a sweep region. In the pan-ancestry analysis,
we identified 51 genome-wide significant SNPs in the Control group (0.15%), but none
were consistent with a selective sweep, because they were randomly distributed
across the genome.

**Changes:** We have updated the main text to include the counts of significant SNPs for
both groups.

lines 238-241:

In contrast, when using imputed aDNA genotype probabilities, we identified 11
genome-wide significant selective sweeps in the GWAS group ($n=476$ SNPs),
and none in the control group ($n=51$ SNPs), consistent with selection acting on
trait-associated variants (Supplementary Note 2a, Supplementary Figs. S2a.3
to S2a.25).

*The authors investigate the correlation of PCs with polygenic scores for traits such as*
*height, but it does not seem appropriate to treat individuals as independent*
*observations (they are related at different degrees), so it seems that some by-*
*chromosome bootstrap or similar could gauge evolutionary uncertainty.*

**Response:** We have removed this section from the manuscript due to concerns about
the portability of using present-day GWAS effect sizes to infer polygenic scores for
populations which are only partially ancestral to the GWAS cohort.

*Regarding the possibility of SNPs beginning a frequency rise earlier than the classic*
*lactase persistence candidate, the suggested 12,000 years ago for rise in frequency of*
*rs1438307 is quite a bit further into the past than the majority of data available. Could*
*the authors add confidence intervals to Figure S2a.44. If this is a major claim, then*
*that figure could serve as a main text figure panel too.*

**Response:** We are confident in the robustness of these results, and have
independently replicated the signal using publicly available data from the 1240k
capture array data. This new analysis—using a different set of samples, genotyped
with a different sequencing technology—shows the same pattern, in which rs1438307
rises in frequency thousands of years before rs4988235 (Figure S2a.56). The
maximum likelihood trajectories from the CLUES models for both rs1438307 and
rs4988235 are depicted in Figure 4b in the main text.

**Changes:** We have added 95% confidence intervals to Figure S2a.55 and S2a.56.

*Could the authors provide a more intuitive rationale for why conditioning on ancestry in*
*their particular analysis setup provides additional power to detect selection?*

**Response:** Our analysis suggests that there are several reasons why we detect more
selection when conditioning on ancestry; one of which is increased statistical power. In
cases where the selected allele is not segregating in all ancestral backgrounds (e.g., if
it is private to one ancestral path), stratification by ancestry increases our power to
detect selection, as it allows us to separate haplotypes in which the selected allele is
absent (i.e., where selection can have no observable effect). Similarly, in cases where
selection is influenced by epistasis, stratification by ancestry may allow us to separate
haplotypes that contain only a subset of the adaptive markers.

However, the main reason we detect more selection is due to the effects of multiple
waves of admixture. Our pan-ancestry analysis spans three major waves of admixture
(Figure 3), which coincide with dramatic changes in subsistence strategy, as well as
large movements of people into new environmental niches. In cases where selection is
acting in only one population, admixture can confound analyses based on time-series
data, especially when the admixing populations have substantially different allele
frequencies. Stratifying by ancestry controls for this effect, as it allows us to model
changes in allele frequency independently of changes in admixture fraction.

*Why did the authors opt for a 4-way mixture model with EHG and CHG, instead of*
*WHG, Anatolia, and Yamnaya?*

**Response:** Evidence suggests that the Yamnaya population was formed from the
admixture of EHG and CHG populations. Given that we have a number of
representative samples from both these populations as well as Yamnaya samples we
included two paths leading to the Yamnaya population. This allows insight into

selection events happening on an EHG vs CHG ancestry background rather than
simply on a Yamnaya background, especially in cases where selection may have
occurred before the admixture events that formed the Yamnaya population.

*"in chromosome 18, we recover a selection candidate region spanning SMAD7, which*
*is associated with inflammatory bowel diseases such as Crohn's disease 41–43.*

*Taken together these results suggest that the transition to agriculture imposed a*
*substantial amount of selection for humans to adapt to our new diet and that some*
*diseases observed today in modern societies can likely be understood as a*
*consequence of this selection."*

*-The link between ancient selection and present-day disease seems overly speculative*
*based on the data presented in this paper.*

**Changes:** We have softened the language used to describe these results.

lines 313-316:

Taken together these results suggest that the transition to agriculture imposed
a substantial amount of selection for humans to adapt to a new diet and
lifestyle, and that the prevalence of some diseases observed today in present-
1031 day societies may be a consequence of these selective processes.

*"However, profound shifts in lifestyle in Eurasian populations during the Holocene,*
*including a change in diet and closer contact with domestic animals, combined with*
*higher mobility and increasing population sizes, are likely drivers for strong selection*
*on loci involved in immune response."*

*-Maybe, but there is no firm data on this yet. Perhaps the authors could say "have*
*been hypothesized to be likely drivers...."*

**Changes:** We have softened the language used here.

lines 346-349:

However, profound shifts in lifestyle in Eurasian populations during the
Holocene have been hypothesised to be drivers for strong selection on loci
involved in immune response. These include a change in diet and closer
contact with domestic animals, combined with higher mobility and increasing
population density.

*"These results suggest that large, recurrent CNVs that can lead to several pathologies*
*were present at similar frequencies in the ancient and modern populations included in*
*this study. " Can it really be assumed that the ancient sample is representative of the*
*past frequencies? It seems too much to make conclusions about prevalence.*

**Response:** We believe that it is reasonable to assume that the frequency of CNVs observed
in our ancient samples is representative of past prevalence. To control for potential bias from
low sequencing depth, and other aDNA characteristics, we estimate CNV prevalence after
performing quality control filtering of our ancient genomes. It is possible that the underlying
samples are themselves biased with respect to CNV prevalence (e.g., if a CNV pathology
reduced the likelihood of survival into adulthood), but we have no evidence to suggest this is

the case. Nevertheless, these results are specifically worded to report that we observe
similar frequencies “in the ancient and modern populations included in this study”.

*[Minor points, requested clarifications, typos]*

*Table S2d.3 West Eurasia.cw_hg spans 40 pages. Can it be reduced?*

**Changes:** We have moved this table (Supplementary Table S2d.3), and all other long
tables, into a separate Supplementary Tables spreadsheet.

*Abstract:*

*Page 1: “high contribution of alleles conferring risk of traits related to diabetes.”*

wording

**Changes:** We have reworded this sentence.

lines 63-67:

Alleles associated with increased risk of some mood-related phenotypes are
overrepresented in the farmer-associated component, entering Europe from
Anatolia around 11,000 years ago, while risk alleles for diabetes and
Alzheimer's disease are highly enriched for tracts with affinities to ancient
Western Hunter-gatherers.

*Results/discussion:*

*Samples and data*

*Page 3: “Unprecedented sample”, “unprecedented details” repeat words*

**Changes:** We have removed all usage of the word “unprecedented”

lines 128-130:

Our analyses are undertaken on the largest collection of shotgun-sequenced
ancient genomes published to date; presented in the accompanying study
‘Population Genomics of Stone Age Eurasia’ (Allentoft et al. 2022).

lines 139-141:

This dataset allows us to characterise in fine detail the changes in selective
pressures exerted by major transitions in human culture and environment.

lines: 435-438

Additionally, our results provide detailed information about the duration and
geographic spread of these processes (Fig. 4) suggesting that an allele
associated with lighter skin was selected for repeatedly, probably as a
consequence of similar environmental pressures occurring at different times in
different regions.

*Figure 1*

*Sampling times and pop split times don't line up, especially 180 generations ago*

*Maybe add borders to distinguish the different parts, legends etc*

**Changes:** We have replaced this figure with a new version (Figure 2) that describes
the improved model used in the current results.

*Selection on diet-associated loci*

*Page 5: "settling controversies regarding the timing of this selection" too strong word*

**Changes:** We have reworded the main text to remove the claim that our results settle
the controversies of the timing of this selection.

lines 250-252:

We find strong changes in selection associated with lactose digestion after the
introduction of farming, but prior to the expansion of the Steppe pastoralists
into Europe around 5,000 years ago (Allentoft et al. 2015; Haak et al. 2015),
the timing of which is a long standing controversy (Enattah et al. 2008; Itan et
al. 2009; Séguérel and Bon 2017; Segurel et al. 2020).

*Genetic trait reconstruction and the phenotypic legacy of ancient Europeans*

*Page 13: "help to settle the famous discussion of selection in Europe relating to*
*height" again perhaps too bold a claim*

**Changes:** We have reworded this sentence.

lines: 554-559

These results also help to clarify the famous discussion of selection in Europe
relating to height (Mathieson et al. 2015; Cox et al. 2019; Rosenstock et al.
2019). Our finding that the 'Steppe' ancestral components (Yamnaya/EHG)
have consistently high genetic values for height in the UK Biobank
demonstrates that height differences between Northern and Southern Europe
may be a consequence of differential ancestry, rather than selection, as
claimed in many previous studies (Field et al. 2016). However, our results do
not preclude the possibility that height has been selected for in specific
populations (Chen et al. 2020; Howe et al. 2022).

*Referee #3 (Remarks to the Author):*

*This paper presents multiple extremely interesting analyses of patterns of genetic*
*variation across several hundred ancient genomes, shedding new light on how natural*
*selection drove rapid changes in allele frequency at a number of loci across the*
*genome during the evolutionary history of modern Europeans. The methods are*
*innovative, and the results provide new insights into the timing of the onset of natural*
*selection for several mutations that are known to have played an important role in*
*adaptation as human migrated into and across Europe (e.g. FADS, LCT), as well as*
*the identification of new candidate selective sweeps that were previously obscured by*
*the effects of admixture. My criticisms below notwithstanding, it represents a real*
*triumph for aDNA in looking back in time to reconstruct human evolutionary history. I*
*think that many of the analyses and results potentially of interest to the broad Nature*
*readership.*

*However, there are several aspects of the manuscript that need work. I have one*
*major substantive criticism, as well as some frustration that several aspects of the*
*manuscript simply do not appear ready for publication.*

**Response:** We thank the reviewer for their assessment that our paper is extremely
interesting and that it represents a real triumph for aDNA. To address the issues
raised by the reviewer, we have made multiple improvements to the original
manuscript, which we outline in detail below.

#####

*My substantive criticism:*

*the connection between a particular positive selection signal and a given nearby*
*complex trait association is often not clear. In many cases, the actual evidence of a*
*link is extremely weak or altogether absent, but the manuscript is framed as if such*
*evidence exists.*

**Response:** We have updated the main text to moderate the language used when
referring to selection signals and the trait associations of the top SNPs (details below).

###

*For example, psoriasis is mentioned in the abstract as a phenotype that has a high*
*prevalence, and imply that their results may explain why. As far as I can tell, aside*
*from the supplementary tables the only mention of psoriasis in the paper is in this*
*sentence:*

*"In contrast, the signal of selection at C2 (rs9267677; $p=9.82e-14$; $s=0.04463$), also*
*found within this sweep, and associated with psoriasis risk in UK Biobank ($p=4.1e-$*
*291; $OR=2.2$), shows a gradual increase in frequency beginning c. 4,000 years ago,*
*before rising more rapidly c. 1,000 years ago."*

*However, I noticed a nearly identical sentence in the supplement, but the phenotype*
*mentioned there is educational attainment:*

*"In contrast, the signal of selection at C2 (rs9267677; p= 9.82e-14; s= 0.04463), also*
*found within this sweep, and associated with educational attainment, shows a gradual*
*increase in frequency beginning c. 4,000 years ago, before rising more rapidly c.*
*1,000 years ago; highlighting the complex temporal dynamics of selection at the HLA*
*locus."*

*Would the authors also be willing to argue in the abstract that this signal of selection*
*may help explain patterns of variation in educational attainment? The strength of the*
*evidence for either conclusion is basically the same.*

**Response:** As we were interested in understanding how natural selection has
influenced the evolution of human traits, we compiled an exhaustive list of all trait
associations reported in the GWAS Catalog. A consequence of this approach is that
some of the trait associations were for phenotypes which have debatable
interpretation outside of the specific environmental context, and socioeconomic status
of the cohort, in which they were measured. We chose not to feature associations for
traits like this in the main text, as these are more likely to be enriched for uncorrected
stratification, and their interpretive value in ancient populations is unclear (see Irving-
Pease et al. 2020 *Front. Genet.*; <https://doi.org/10.3389/fgene.2021.703541>). In the
specific case of rs9267677, we note that the odds-ratio for the association with
educational attainment is less than 1.02 (Lee et al. 2018 *Nature Genetics*;
<https://doi.org/10.1038/s41588-018-0147-3>), whereas the odds-ratio for the
association with psoriasis is 2.2; consistent with selection at rs9267677 explaining a
substantially larger fraction of present-day variation in psoriasis risk than it does for
EA.

**Changes:** We have removed the specific reference to psoriasis in the abstract.

lines 49-52:

A substantial amount of selection is also found in the HLA region and other loci
associated with immunity; possibly due to increased exposure to pathogens
during the Neolithic, which may have contributed to the currently high
prevalence of auto-immune diseases.

**###**

*Another example, not directly related to the main, selective sweep focus of the paper,*
*is in the section titled: "Pathogenic structural variants in ancient vs. modern-day*
*humans".*

*The authors write:*

*"RISE586 exhibited a hypoplastic tooth, spondylolysis of the L5 vertebrae, incomplete*
*coalescence of the S1 sacral bone, among other minor skeletal phenotypes. The*
*skeletal phenotypes observed in this individual are relatively common (~10%) in*
*European populations and are not specific to 16p13.1 thus do not indicate strong*

*penetrance of this mutation in RISE586. However, these results do highlight our ability*
*to link putatively pathogenic genotypes to phenotypes in ancient individuals."*

*I do not see how a pathogenic genotypes has been in any way "linked" to phenotypes*
*in ancient individuals. One ancient individual has phenotypes that are common among*
*other ancient individuals, and also carries the deletion/duplication. This is not a result.*
*It just means that the authors were able to genotype an individual for whom they can*
*also measure skeletal traits. I understand that there is some hope that potentially in*
*the long run this sort of paired data can be used to learn more about the relationship*
*between the genotypes and phenotypes of ancient individuals, but this hasn't actually*
*been done here. I think writing that this mutation is not strongly penetrant in this*
*individual is positively misleading. There is no evidence of ANY penetrance or*
*relationship to the phenotype whatsoever.*

*(I should also note that it is not clear to me whether the the variant of interest in the*
*above section is a deletion or duplication. In line 388, first it is a duplication*
*("duplications at 16p13.11"), then later in the same line it is a deletion ("An individual*
*harbouring the 16p13.11 deletion"). Maybe there are both? I can't tell...*

**Changes:** We have removed the paragraph that implied an association between the
pathogenic CNV and the observed skeletal traits.

*In general, I think the paper significantly oversells what the results actually tell us*
*about phenotypic variation. The abstract closes with a sentence that begins: "Our*
*results paint a picture of the combined contributions of migration and selection in*
*shaping the phenotypic landscape of present-day Europeans...". But as I've argued*
*above, there is generally little to no evidence linking the reported results to the*
*"phenotypic landscape" of present-day Europeans.*

**Response:** The results presented in our manuscript have two broad approaches. The
first approach focuses on identifying evidence of selection for trait associated variants.
In our ancestry stratified time-series analysis, we identified 21 genome-wide significant
loci with evidence of strong selection. However, linking these loci to phenotypic
outcomes is complicated, because each loci is highly pleiotropic and most traits of
interest are highly polygenic. For large effect loci, like LCT, SLC45A2 and FADS,
single-locus results can inform directly on the phenotypes of present-day Europeans,
but for many other loci the picture is more complicated. This is why we undertook the
second major approach of the paper, which focused on understanding the present-day
genetic legacy of Mesolithic hunter-gatherer, Neolithic farmer and Bronze Age
pastoralist populations. Using our ancient genomes as 'donors' to chromosome paint
the UK Biobank, we identified the local ancestry composition of different complex
traits, in the same genomes used to perform the GWAS. We used this information to
develop ancestral risk scores (ARS) for 35 complex traits (Figure 6), which represent
the differing contributions of ancestral populations to present-day phenotypes in more
than 400,000 British people. Our results reveal major differences in the contributions
of ancestral risk to present-day people for a range of anthropometric, metabolic and
disease traits. To better highlight the significance of these findings, we have made
substantial modifications to the main text to include additional discussion of these

results under the subheading “The phenotypic legacy of ancient Eurasians” on line
497.

*For some of these sweeps it is entirely possible that the phenotypes that drove them*
*are not expressed in the modern human environment. The field has been identifying*
*selection signals physically nearby to trait associations for some time now, and that's*
*what most of them still are: two different signals that are close to one another in the*
*genome but have not other obvious connection. The signals here seem a lot more*
*likely to be "real" than many earlier ones based on iHS or other similar metrics, but the*
*hard work of determining how these sweeps are related to present day phenotypic*
*variation, if at all, lies in the future. To be clear, I think that pointing out nearby*
*phenotypic association or known functions of sweep candidates is fine, but that the*
*overall packaging of the manuscript as if it sheds serious light on this goes too far.*

**Response:** Establishing causality between a selection signal and a particular
phenotype is extremely difficult, if not impossible, outside of an experimental evolution
study. We agree with the reviewer that it is entirely possible, if not likely, that the truly
adaptive phenotypes in recent human evolution are not well characterised in GWAS.
Even in the case of putatively monogenic loci, establishing causality is complicated.
For example, our results show strong evidence of at least two sweeps at the
LCT/MCM6 locus, containing variants with strikingly different metabolic phenotypes.
Nevertheless, our study has not sought to establish causality between a selection
signal and a phenotype, and there is no need to invoke causality to link changes in
allele frequencies to present-day phenotypic variation. In cases where we have strong
evidence that a trait associated variant has changed in frequency, those changes will
have affected present-day expression of that trait, regardless of the causal
phenotype(s) that drove the selective sweep.

**Changes:** To avoid any implication of causality between our selection analysis and
our reporting of trait associations, we have moderated the language used when
referring to trait associations, and have added further caveats to the discussion.

lines 566-570:

Due to the highly pleiotropic nature of each sweep region, it is difficult to ascribe
causal factors to any of our selection signals. However, our results show that
selection during the Holocene has had a substantial impact on present-day genetic
disease risk, as well as the distribution of genetic factors affecting metabolic and
anthropometric traits.

#####

*My second major criticism is that the supplement appears incomplete and has many*
*errors and in many places either does not produce enough detail about the methods*
*used, or has text which doesn't fully track.*

*For example, there is a paragraph starting on line 187 of the supplement that explains*
*why knowing only the first coalescent event is not sufficient for understanding the full*
*ancestry of a given haplotype. While this is no doubt true, this text is not obviously*

*related to the surrounding text. I can surmise that it may be an explanation of why the*
*tool MSMC (which models the first coalescent event and was developed in senior*
*author Richard Durbin's group) isn't an appropriate tool for this task, but MSMC is*
*never explicitly referenced, nor is the "first coalescence event" mentioned anywhere*
*else in the main text or the supplement.*

**Response:** To address the lack of detail regarding our novel chromosome painting
model, we have prepared a separate manuscript which comprehensively describes the
methodology, validation and benchmarking of the method (Pearson & Durbin 2023
*bioRxiv*; <https://doi.org/10.1101/2023.03.06.529121>). We have also added additional
discussion to the supplementary text to address the specific issue raised by the
reviewer.

*Relatedly, the authors say they adapted CLUES to model time series data. It's hard to*
*tell from the supplemental text whether they've ADDED time series data on top of the*
*existing functionality that uses inferred ARGs (i.e. using aDNA and ARGs jointly), or if*
*it's just that they've taken the CLUES codebase and basically spun off a different (but*
*obviously related) method that uses a time series of aDNA to infer trajectories. The*
*current descriptions in the supplement are extremely cursory, and I think it would be*
*appropriate for the authors to given a more complete description of the methods as*
*actually used.*

**Response:** Our new version of the CLUES software can be run in three possible
modes, using either (i) ARGs only; (ii) aDNA time-series only; or (iii) aDNA time-series
and ARGs jointly. In this analysis, we used ARGs only for the 1000G populations; and
aDNA time-series only for the ancient populations. This is described in Supplementary
Note 2a (e.g., “We also ran CLUES in an alternative mode, excluding the modern
ARG data, and replacing them with aDNA time series data”), where we also provide a
complete list of the command line arguments used. We have also provided a GitHub
repository that contains all the pipeline code, and a conda environment, to fully
reproduce our CLUES analyses (see https://github.com/ekirving/mesoneo_paper/).
We also provide another GitHub repository that contains the revised version of
CLUES, and includes a tutorial on how to use it (see [https://github.com/standard-](https://github.com/standard-aaron/clues)
[aaron/clues](https://github.com/standard-aaron/clues)). To validate the accuracy of our updates to the CLUES method, we have
performed a set of new simulations. These simulations show that CLUES accurately
infers selection coefficients and allele frequency trajectories for sample sizes smaller
than used in our empirical analyses. Full details of the simulation design and
benchmarking accuracy are described in Supplementary Note 2b.

*At line 3050 in supplementary section 2g, the authors write:*

*"To calculate an ancestry-specific PRS we used an additive model, including a*
*transformation as in Berg & Coop and in line with (Supplementary Note S2c (Allentoft*
*et al. 2022))"*

*I can't tell what transformation in Berg & Coop they are referring to. I also checked the*
*Allentoft citation and there does not appear to be a section 2c.*

**Changes:** The Berg & Coop transformation is a conversion of the scores to standard
deviations from the pan-ancestry mean (i.e., the mean of all ancestries analysed). The PRS
are therefore shown in z-score units. We have also fixed the incorrect cross-reference.

lines 3620-3622 (Supplement):

To calculate an ancestry-specific PRS we used an additive model, including a
transformation as in Berg & Coop, which converts scores to standard
deviations from a pan-ancestry mean (i.e. z-scores).

*Additional pieces of the supplement that still need some work:*

*Supplementary text 1a switches back and forth between first person singular and first*
*person plural.*

**Response:** We have reviewed the supplement, and made various changes to improve
consistency, clarity and improve robustness of the analysis.

*Figure S2c.2, it appears the row labels have been removed, presumably by accident.*

*For Figure S2c.3, the traits are referenced only by their numbers in the "UK Biobank*
*coding system". I think it is not unreasonable for readers to expect a figure with human*
*readable trait labels on it.*

*For Figure S2c.6, the figure caption reads "Principal component analysis on West*
*Eurasian samples coloured by individual polygenic scores.", but no part of the figure*
*indicates what trait the polygenic scores are for. Searching the text, it seems like it is*
*for height, but this should be clearly indicated in the figure.*

**Changes:** We have removed this chapter from the supplement, and the corresponding
results from the main text, due to reviewer concerns about the portability of present-
1410 day effect size estimates in populations that are only partially ancestral to the
1411 discovery cohort.

*Tables S2d.1 S2d.2 and S2d.3 have no figure captions. I can mostly guess at what the*
*column headings are, but readers shouldn't have to. The supplementary text refers to*
*Tables S2d.1 and S2d.2, but there does not appear to be a reference to Table S2d.3*
*anywhere in the text.*

**Changes:** We have moved these tables into a separate Supplementary Tables
spreadsheet, and better annotated the column headers (see Supplementary Table
S2d.1 - S2d.3).

*Note that this list is not exhaustive, and I do not think that the authors merely need to*
*respond to the specific examples I point out. Rather, I think the authors need to take a*
*serious pass through the supplement again, including sections that I do not explicitly*
*note here, and make sure that it is actually ready for publication.*

**Changes:** We have reviewed the supplement, and made various changes to improve
consistency, clarity and improve robustness of the analysis.

#####

*Lastly, I have a few comments on the "polygenic selection" analyses relying on*
*polygenic scores:*

*1) I could not find the actual quantitative results of this analysis. There are figures in*
*the main text that show the traits that pass a bonferroni multiple testing threshold, and*
*figures in the supplement that show a heatmap of some summaries of the analysis (as*
*noted above, however, these figures are not human readable), but readers should*
*have access to the actual results.*

**Response:** We have removed some of the analyses which were missing the raw
quantitative results, and now provide a separate Supplementary Tables spreadsheet for all
the remaining results.

*Relatedly, there is some basic information about the empirical randomization scheme*
*that I could not find: e.g. how many null replicates were sampled to generate the*
*empirical p values?*

**Changes:** We have removed this chapter from the supplement, and the corresponding
results from the main text, due to reviewer concerns about the portability of present-day
effect size estimates in populations that are only partially ancestral to the discovery cohort.

*2) The statement at line 464 that "these analyses help to settle the famous discussion*
*of selection in Europe relating to height" is far too strong. There are at least three*
*potentially distinct signals of selection on height that have been reported in or near*
*Europe. Field et al 2016 reported a signal of recent selection for increased height in*
*Britain within the last 2000 years, based on analyses using the singleton density score*
*(Howe et al 2022 also supported this result using sibling based effect sizes which are*
*free of confounding). There is also a reported signal of selection for decreased height*
*in Sardinia, supported by effect sizes from the Biobank of Japan (Chen et al 2020).*
*Then, there is the signal that's being reported here, which is similar to one reported by*
*Mathieson et al 2015. I think the authors should be clearer about the broader context*
*and complex history of this particular question.*

**Response:** The signal of polygenic adaptation reported in Field et al. (2016, *Science*;
<https://doi.org/10.1126/science.aag0776>) was based on GWAS effect sizes from GIANT and
R15-sibs, both of which were confounded by stratification along the North-South gradient
where signals of selection were reported (Berg et al. 2019, *eLife*;
<https://doi.org/10.7554/eLife.39725>). The signals reported in Chen et al. (2020 *AJHG*
<https://doi.org/10.1016/j.ajhg.2020.05.014>) and Howe et al. (2022 *Nature Genetics*;
<https://doi.org/10.1038/s41588-022-01062-7>) relate to specific populations, and make no
broader claim about a North-South gradient.

**Changes:** We have reworded the presentation of our results on height to include
citations to these three papers.

lines: 554-559

These results also help to clarify the famous discussion of selection in Europe
relating to height (Mathieson et al. 2015; Cox et al. 2019; Rosenstock et al.
2019). Our finding that the 'Steppe' ancestral components (Yamnaya/EHG)
have consistently high genetic values for height in the UK Biobank
demonstrates that height differences between Northern and Southern Europe
may be a consequence of differential ancestry, rather than selection, as
claimed in many previous studies (Field et al. 2016). However, our results do
not preclude the possibility that height has been selected for in specific
populations (Chen et al. 2020; Howe et al. 2022).

Reviewer Reports on the First Revision:

Referees' comments:

Referee #1 (Remarks to the Author):

I thank the authors for the comprehensive response to my review. I am largely satisfied. I have just a couple suggestions for clarifications in the text and supplement:

- In response to a comment from R2, the authors added (L238-241) the number of SNPs that make up each of the significant sweep signals. This results in the rather confusing statement that they identified "none in the control group (n=51 SNPs)". I suggest rephrasing to clarify that these SNPs were significant, but did not meet the secondary criteria to for being a sweep.

- I found the the description provided in the response letter of how this approach differs from other recent methods to be extremely helpful in framing its contribution. I would suggest adding this to the supplement.

Referee #2 (Remarks to the Author):

The manuscript by Irving-Pease et al. has been substantially revised, and one major advance of the previous submission has now instead been moved to a separate paper (Pearson and Durbin 2023, bioRxiv). The paper provides several highly interesting observations, but similarly to other reviewers I think conclusions tend to not be appropriately caveated.

I have several concerns about the revised paper:

1. Other manuscripts

The first is the questions about the delimitations of companion papers and dual publication. This Irving-Pease et al. manuscript is part of a network of papers (Irving-Pease et al., Allentoft et al., da Mota et al., Barrie et al., Pearson and Durbin). Arguably the greatest advance of the previous manuscript was the deconvolution of ancestry which allowed selection in different ancestral populations to be reconstructed. This is now published in a separate preprint by Pearson and Durbin, which I assume is the primary publication of the method. The main text of this revised paper still claims that this method is "novel"-but surely it is not if it is presented in the preprint. This should be corrected and what is novel in this paper should be delimited from what is presented in other publications.

The current submission thus reports neither new ancient genomes (which are reported in the Allentoft et al. 'main paper' submission), or strongly novel approaches aside from the incorporation of ancient genome allele frequencies in CLUES (which is a very welcome advance). Also the fascinating signals discussed in the HLA locus are presented in a separate Barrie et al. preprint. Figure 1 is a description of data that is previously published, and presented new by Allentoft et al.

All these aspects reduce the advance that this paper represents, and in my view causes some degree of confusion in the scientific literature.

2. Restricting selection scans to 33 thousand functional SNPs

As mentioned in previous reviews, this paper takes a different approach to previous leading selection papers (e.g. Mathieson et al. 2015, Field et al. 2018) and doesn't scan the entire genome of millions of loci, but only scans 33 thousand SNPs.

In the revised version, the authors respond to these concerns by talking about "causality" and removing any mention of such causality. But to me the point is not about causality, but about whether their approach has identified the SNPs with the most evidence for selection in their data or missed them, a question which could be decoupled from phenotypes altogether. Their analysis of the LCT region seems to prove this point, as when they re-scan the entire broader locus they find that the most evidence for selection is in a SNP not included in the first scan. I think the caveat of other selected SNPs possibly being nearby should be mentioned clearly.

Other points:

Figure 2 shows proportions of ancestry across Eurasia and in Britain, but methods such as CHROMOPAINTER have been demonstrated to be sensitive to demographic history, for example variable rates of genetic drift in different populations (Lawson, van Dorp, and Falush 2018). What are the confidence intervals for these estimates, e.g. of WHG and EHG ancestry in East Asia?

The paper highlight in the abstract differential Neolithic farmer ancestry across the UK, with higher proportions in the south and east, but this has already been shown by Patterson et al. 2021 and Olalde et al. 2018, and the former study addressed exactly the question of why there are these differences in Britain.

No state of the art study claims white British are homogeneous. See e.g. Leslie et al. 2015 in which analysis of the POBI data revealed fine-scale structure, and more recently Saada et al. 2020 who analysed the UK Biobank.

They claim that the most strongly selected pigmentation alleles reached fixation several thousand years ago, but to me it is hard to see what is novel compared to evidence presented e.g. by Ju and Mathieson 2021.

"The selective forces likely favouring ApoE2 in Steppe pastoralists may be associated with protective immune responses against infectious challenges, such as protection against severe childhood malaria or infection with an unknown coronavirus" -why specifically a coronavirus? This seems sensationalistic.

Overall, I think this paper provides an interesting and valuable analysis of the new substantial data to-be-presented in the Allentoft et al. preprint, but has reduced novelty due to the expansive network of multiple publications.

Referee #4 (Remarks to the Author):

The authors present a rigorous analysis of selection on complex traits in Europe over the last 12,000 years using ~1,600 imputed ancient genomes. They also describe the role of ancient admixture in shaping the genetic variation underlying complex traits in present-day Europeans. I found the manuscript very interesting and for the most part, clearly written. This is not easy given the limited space. The authors also seem to have done a commendable job of responding to the reviewers' comments constructively and addressing them.

In particular, reviewer 3's comment on the weak link between selection hits and trait-associated variants was well-made. As the authors wrote, connecting selection hits to traits is not trivial and remains an unsolved problem in the field. This does not kill the study as long as the results are properly caveated, which they are. The authors have also toned down the language throughout the paper sufficiently so as not to imply that a signal of selection on a variant implies selection on a trait that the variant or some other variant nearby might be associated with.

I do have some minor comments:

1. The differences in ancestral risk scores seen in Fig. 6 could yet be due to subtle inflation in effect sizes due to stratification in the original GWAS given that there are ancestry gradients in UKB in the same direction. For example, they found height ARS to be higher for Steppe compared to WHG, which is consistent with people in the north being taller and having more Steppe ancestry compared to people in the south who have more WHG ancestry. This could either be a real signal — that the ancestry gradient actually contributes to the north-south cline in height — or it could be because the GWAS effect sizes are influenced by uncorrected stratification in the UKB. The authors recognize this as well in Supplementary note S2f and I don't necessarily think this is the case. But I wonder, given the strongly worded statement in lines 554, if the authors should also state this caveat in the main text.
2. In Figs. 3 and S1c1, please clarify that the numbers along the paths are effective population sizes (presumably).
3. I couldn't find the generation time used to convert the time in generations in Figs. 3 and S1c1 and the time in years in Fig. 4.

Presumably between 25-30 years given the scales but good to be explicit.

*Referees' comments:*

*Referee #1 (Remarks to the Author):*

*I thank the authors for the comprehensive response to my review. I am largely satisfied. I*
*have just a couple suggestions for clarifications in the text and supplement:*

*- In response to a comment from R2, the authors added (L238-241) the number of SNPs that*
*make up each of the significant sweep signals. This results in the rather confusing statement*
*that they identified "none in the control group (n=51 SNPs)". I suggest rephrasing to clarify*
*that these SNPs were significant, but did not meet the secondary criteria to for being a*
*sweep.*

**Changes:** We have reworded this sentence to improve clarity.

lines 228-232:

In contrast, when using imputed aDNA genotype probabilities, we identified 11
genome-wide significant selective sweeps in the GWAS group (n=476 SNPs
with $p < 5e-8$), and no sweeps in the control group, despite some SNPs
exhibiting evidence of selection (n=51). These results are consistent with
selection preferentially acting on trait-associated variants (Supplementary Note
2a, Supplementary Figs. S2a.3 to S2a.25).

*- I found the the description provided in the response letter of how this approach differs from*
*other recent methods to be extremely helpful in framing its contribution. I would suggest*
*adding this to the supplement.*

**Changes:** We have added an additional section to Supplementary Node 2a
contrasting the methodological and sampling approach used in this paper with the
recent selection papers by Lee et al. (2022) and Kerner et al. (2023).

lines 2964-2994 (Supplement):

Other recent papers have also modelled selection in West Eurasia during the
Holocene (Le et al. 2022; Kerner et al. 2023); however, it is difficult to directly
compare results due to substantial differences in methodology and sampling.
Lee et al. (2022) use an updated version of the mixture model developed in
Mathieson et al. (2015), which relies on differences in allele frequencies
postdating admixture. As such, they are best powered to detect rapid episodes
of selection following admixture between populations. The selection test used
by Kerner et al. (2023) is based on choosing variants with an estimated
selection coefficient above the 99th quantile from their simulations, and is
therefore best-powered to detect cases of strong selection. In our analyses, we
used a selection test that is well-powered to detect both weak and strong
selection, and we used local ancestry inference to deconvolute the effects of

changes in admixture proportions through time, allowing us to detect selection
in a broader range of demographic scenarios.

Another key difference is in sampling. Both Lee et al. (2022) and Kerner et al.
(2023) used pseudohaploid data from the 1240k capture array, which is
affected by allelic bias, due to the capture chemistry (Rohland et al. 2022;
Davidson et al. 2023). It remains unclear how sensitive selection results from
the 1240k array are to systematic bias in the recovery of some alleles;
however, Kerner et al. (2023) found that nine of the top 10 variants in their
capture dataset had a frequency trajectory inconsistent with their shotgun
dataset. This suggests that allelic bias from the 1240k capture chemistry may
be a major confounder for tests of selection. In comparison to shotgun data,
Rohland et al. (2022) found that 61.7% of the SNPs on the 1240k capture array
exhibit evidence of allelic bias ($n=757,587$ with
``PassFilterForMetaAnalysisBias==0``).

A compounding factor may also be systematic differences in capture efficiency
between sites, which results in much smaller sample sizes than the reported
number of ancient individuals. For example, in our analysis of selection at the
LCT locus using the 1240k dataset (Supplementary Figure S2a.56)—which
used the same 1,291 samples as Lee et al. (2022)—we observed that there
were 838 pseudohaploid calls for rs4988235, but only 476 for rs1438307,
indicating capture efficiency varies greatly between sites, as well as between
alleles at the same site. In comparison, our imputed callset contains 1,015
diploid genotypes for all modelled SNPs, and we show via replication (using
genotype-likelihoods) that imputation does not substantively bias our inference
of allele frequency trajectories or selection coefficients.

*Referee #2 (Remarks to the Author):*

*The manuscript by Irving-Pease et al. has been substantially revised, and one major*
*advance of the previous submission has now instead been moved to a separate paper*
*(Pearson and Durbin 2023, bioRxiv). The paper provides several highly interesting*
*observations, but similarly to other reviewers I think conclusions tend to not be appropriately*
*caveated.*

*I have several concerns about the revised paper:*

*1. Other manuscripts*

*The first is the questions about the delimitations of companion papers and dual publication.*
*This Irving-Pease et al. manuscript is part of a network of papers (Irving-Pease et al.,*
*Allentoft et al., da Mota et al., Barrie et al., Pearson and Durbin). Arguably the greatest*
*advance of the previous manuscript was the deconvolution of ancestry which allowed*
*selection in different ancestral populations to be reconstructed. This is now published in a*
*separate preprint by Pearson and Durbin, which I assume is the primary publication of the*
*method. The main text of this revised paper still claims that this method is "novel"-but surely*
*it is not if it is presented in the preprint. This should be corrected and what is novel in this*
*paper should be delimited from what is presented in other publications.*

**Response:** The preprint by Pearson & Durbin, which describes the novel method for local
ancestry inference, has been fully reincorporated into the supplement of this paper. The
preprint has not been submitted to any other journal, and we have moved the entirety of the
content into Supplementary Note 1c, replacing the previous benchmarking analysis — the
results of which remain unchanged.

*The current submission thus reports neither new ancient genomes (which are reported in the*
*Allentoft et al. 'main paper' submission), or strongly novel approaches aside from the*
*incorporation of ancient genome allele frequencies in CLUES (which is a very welcome*
*advance). Also the fascinating signals discussed in the HLA locus are presented in a*
*separate Barrie et al. preprint. Figure 1 is a description of data that is previously published,*
*and presented new by Allentoft et al.*

*All these aspects reduce the advance that this paper represents, and in my view causes*
*some degree of confusion in the scientific literature.*

**Response:** Our original submission, dated May 2022, consisted of one large manuscript that
contained all of the results and analyses presented in Allentoft et al. (the “main paper”) and
Irving-Pease et al. (the “selection paper”). It was at the suggestion of Editor that
these were split into two separate papers, to allow more focused presentation of the results.

We believe this selection manuscript represents a substantial scientific advance that is
independent of the data generated in the main paper (especially now that the novel
LAI method has been reincorporated). In terms of methodological novelty, we present
(i) a new method for performing local ancestry inference; (ii) a new method for inferring
allele frequency trajectories from time-series data; (iii) a novel pipeline for

deconvoluting admixture in a selection test; and (iv) a new statistical model to
distinguish direct effects of age on allele frequency from indirect effects mediated by
read depth, read length, and/or error rates. We also apply existing methods in novel
ways, by using ancient populations as donors to “chromosome paint” the UK Biobank,
and by inferring ancestry-specific polygenic risk scores, for which we coin the new
term “Ancestral Risk Scores”. More importantly, we used these novel methodologies to
make substantial biological insights into the strength and timing of selection at key
dietary and immune loci, as well as characterising how differential ancestry has
affected present-day anthropometric and disease traits in the British population.

Figure 1 shows a map of sampling locations and ages, and was added at the request
of Reviewer 1. Whilst all of these samples are described in other publications, this
figure accurately reflects the breadth and depth of the sampling used in our time-
series selection analyses. The corresponding panel of Figure 1 in the Allentoft et al.
paper only shows the 317 novel genomes presented in that study, and excludes the
majority of the samples used in our analyses.

*2. Restricting selection scans to 33 thousand functional SNPs*

*As mentioned in previous reviews, this paper takes a different approach to previous leading*
*selection papers (e.g. Mathieson et al. 2015, Field et al. 2018) and doesn't scan the entire*
*genome of millions of loci, but only scans 33 thousand SNPs.*

*In the revised version, the authors respond to these concerns by talking about "causality"*
*and removing any mention of such causality. But to me the point is not about causality, but*
*about whether their approach has identified the SNPs with the most evidence for selection in*
*their data or missed them, a question which could be decoupled from phenotypes altogether.*
*Their analysis of the LCT region seems to prove this point, as when they re-scan the entire*
*broader locus they find that the most evidence for selection is in a SNP not included in the*
*first scan. I think the caveat of other selected SNPs possibly being nearby should be*
*mentioned clearly.*

**Response:**

Our study design was based on the hypothesis that natural selection would systematically
favour variants with GWAS trait associations, when compared to a control set of non-trait
associated variants. The results from our pan-ancestry analysis confirm this hypothesis, and
show a >9-fold enrichment for evidence of genome-wide significant selection among the
GWAS set of variants. In comparison to Mathieson et al. (2015, *Nature*;
<https://doi.org/10.1038/nature16152>), it is a striking confirmation of our hypothesis (and of
the greater sensitivity of our methods) that we are able to identify 75% more sweep loci (21
vs. 12) when analysing only 6% as many SNPs (66,682 vs. 1,055,209).

We agree with the reviewer that our experimental design does not guarantee that we have
identified the SNPs with the lowest p-values in our sweep loci. If we were to expand our
analysis to include all 8.5 million SNPs in the imputed callset (a 127-fold increase in the size
of our study), we would likely identify many non-trait associated SNPs with strong evidence
of selection. However, this would not change the conclusions of our paper, which are

focused on the phenotypic consequences of selection, not on finding the SNP with the
lowest p-value.

In the case of the LCT locus, our results demonstrate that exclusively characterising a
selective sweep by the SNP with lowest p-value can obscure important biological signals.
We comprehensively scanned all SNPs within the LCT sweep region, which confirmed our
prior finding that the lactase persistence SNP (rs4988235) exhibits the strongest evidence of
selection at this locus ($p=1.68e-59$). We then analysed the trajectories of all genome-wide
significant SNPs ($p < 5e-8$) within the locus, and ranked them by their *earliest* evidence of
selection. This ranking revealed that the majority of selected SNPs began rising in frequency
thousands of years earlier than the lactase persistence allele, despite all having larger p-
values. This suggests that the LCT locus has experienced at least two separate sweeps, and
that focusing on the SNP with the strongest evidence of selection can obscure selection
signals occurring at deeper time depths.

**Changes:** We have added an additional caveat to the discussion explicitly stating that
we did not test all non-trait associated variants.

lines 557-562:

Due to the highly pleiotropic nature of each sweep region, it is difficult to
268 ascribe causal factors to any of our selection signals, and we did not
exhaustively test all non-trait associated variants. However, our results show
that selection during the Holocene has had a substantial impact on present-day
genetic disease risk, as well as the distribution of genetic factors affecting
metabolic and anthropometric traits.

*Other points:*

*Figure 2 shows proportions of ancestry across Eurasia and in Britain, but methods such as*
*CHROMOPAINTER have been demonstrated to be sensitive to demographic history, for*
*example variable rates of genetic drift in different populations (Lawson, van Dorp, and*
*Falush 2018). What are the confidence intervals for these estimates, e.g. of WHG and EHG*
*ancestry in East Asia?*

**Response:**

A feature of CHROMOPAINTER, as used here, is that it defines ancestry with respect to a
user-defined reference panel. It is therefore sensitive to the particular details of the model
only as far as the different panels extract information regarding different times and
populations. Lawson, van Dorp, and Falush (2018, *Nature Communications*;
<https://doi.org/10.1038/s41467-018-05257-7>) described a way to check a population history
by contrasting SNP-based and haplotype-based signals. Our analyses pass that test
because our novel LAI method (Supplementary Note 1c) was validated against SNP-based
statistics, and the results concur with CHROMOPAINTER, indicating no reason to expect
gross model-misspecification.

In general, the confidence intervals for country level means are low, due to the relatively
 large sample sizes in UKB. However, we believe that the important results are not the
 country level means, but the clines across modern populations. To illustrate this, we have
 calculated 95% confidence intervals for each country, by bootstrapping across individuals
 (iterations=1000). We report the results for East Asia below, for EHG and WHG, and include
 the count of individuals for each country.

 Given that these ancestries are old, we expect their within-country variance to be low. For
 the UK, this is in the region of +/-5% of the mean, depending on the ancestry, and we
 observe similar results for countries in East Asia where the count of samples is greater than
 30. These narrow confidence intervals are partially due to the effectiveness of our selection
 of individuals for a given country; which is based on density-based clustering of the first 18
 PCs of individuals from the UKB born in that country, to select individuals of a 'typical
 ancestral background' (Supplementary Note 1a). These results give us confidence that the
 clines we see in the average ancestry proportion, as reported in the paper, are real.

Country	Ancestry	Mean	Lower CI	Upper CI	Count
Singapore	WHG	0.031469	0.030554	0.032342	86
Singapore	EHG	0.058902	0.057996	0.059811	86
Japan	WHG	0.022166	0.021667	0.022685	242
Japan	EHG	0.038967	0.038264	0.039696	242
Hong Kong	WHG	0.032907	0.032467	0.033335	448
Hong Kong	EHG	0.059084	0.058650	0.059505	448
China	WHG	0.029814	0.029314	0.030325	371
China	EHG	0.059305	0.058829	0.059805	371
Philippines	WHG	0.037516	0.036993	0.038088	310
Philippines	EHG	0.063313	0.062698	0.063908	310
Thailand	WHG	0.035764	0.034564	0.036930	87
Thailand	EHG	0.065922	0.064664	0.067081	87
Indonesia	WHG	0.034818	0.032543	0.037047	35
Indonesia	EHG	0.064188	0.062570	0.065861	35

Cambodia	WHG	0.033638	0.030287	0.036965	7
Cambodia	EHG	0.062782	0.058422	0.067542	7
Macau (Macao)	WHG	0.032064	0.027794	0.036126	6
Macau (Macao)	EHG	0.059363	0.055302	0.062830	6
Taiwan	WHG	0.030333	0.028481	0.032074	23
Taiwan	EHG	0.059557	0.057596	0.061452	23
Mongolia	WHG	0.013266	0.010869	0.015703	6
Mongolia	EHG	0.084820	0.078868	0.091473	6
South Korea	WHG	0.025614	0.023928	0.027248	24
South Korea	EHG	0.057325	0.055454	0.059393	24
North Korea	WHG	0.025231	0.021230	0.028626	5
North Korea	EHG	0.053330	0.049883	0.056928	5

**Changes:** We have added an additional caveat to the main text.

lines 152-155:

Overall, these results refine global patterns of spatial distributions of ancient
ancestries amongst present-day individuals. Whilst the absolute admixture
proportions are dependent on the reference samples used, as well as the
treatment of pre- or post-admixture drift, the geographical variation and
associations should be consistent.

*The paper highlight in the abstract differential Neolithic farmer ancestry across the UK, with*
*higher proportions in the south and east, but this has already been shown by Patterson et al.*
*2021 and Olalde et al. 2018, and the former study addressed exactly the question of why*
*there are these differences in Britain.*

**Response:**

Neither Olalde et al. (2018, *Nature*; <https://doi.org/10.1038/nature25738>) nor Patterson et al.
(2022, *Nature*; <https://doi.org/10.1038/s41586-021-04287-4>) estimated genetic ancestry
proportions in modern individuals from the UK, and we believe we are the first to do this.
Furthermore, while both studies used three-way admixture models to show regional variation
in ancestry proportions in the past, both were limited by sparse sampling to broad regional
comparisons (e.g., comparing England and Wales to Scotland). The novelty of our results is

in showing that these differences persist into the present-day, can be detected on a fine-
scale basis (e.g., between counties), and exist for several genetic ancestries not previously
studied.

**Changes:** We now cite Olalde et al. (2018), in addition to Patterson et al. (2022), in
the main text.

lines 166-171:

This regional pattern was already evident in the Pre-Roman Iron Age and
persists to the present day even though immigrating Anglo-Saxons had
relatively less affinities to Neolithic farmers than the Iron-Age individuals of
southwest Briton. Although this Neolithic farmer/Steppe-related dichotomy
mirrors the modern 'Anglo-Saxon'/'Celtic' ethnic divide, its origins are older,
resulting from continuous migration from a continental population relatively
enriched in Neolithic farmer ancestries, starting as early as the Late Bronze
Age (Patterson et al. 2022; Olalde et al. 2018).

*No state of the art study claims white British are homogeneous. See e.g. Leslie et al. 2015 in*
*which analysis of the POBI data revealed fine-scale structure, and more recently Saada et*
*al. 2020 who analysed the UK Biobank.*

**Response:**

We agree with the reviewer that state-of-the-art studies which specifically examine British
population structure do not claim that it is homogenous; however, many studies treat the
'white British' subset of the UK Biobank as a relatively homogenous population, because it
occupies a restricted PCA space with outliers removed. Many studies restrict to this subset
as a first stage of their analysis, particularly those involving GWAS or PRS calculation —
e.g., Sakaue et al. (2021, *Nature Genetics*; <https://doi.org/10.1038/s41588-021-00931-x>) and
Tanigawa et al. (2022, *PLOS Genetics*; <https://doi.org/10.1371/journal.pgen.1010105>).
Furthermore, while Leslie et al. (2015, *Nature*; <https://doi.org/10.1038/nature14230>) and
Saada et al. (2020, *Nature Communications*; <https://doi.org/10.1038/s41467-020-19588-x>)
both find geographically-based clusters of individuals from the UK, based on haplotype-
sharing or identity-by-descent, neither is able to offer more than speculative historical
reasons for the clustering. Our results demonstrate that there are systematic ancestry
differences within the 'white British' subset which have not previously been described, and
add to the consensus that care is needed to account for population structure.

**Changes:** We have removed the statement that the white British population is
"traditionally considered relatively homogenous".

lines 174-176:

These results demonstrate clear ancestry differences within an 'ethnic group'
(white British), highlighting the need to account for subtle population structure
when using resources such as the UK Biobank genomes (Zaidi and Mathieson
2020).

*They claim that the most strongly selected pigmentation alleles reached fixation several*
*thousand years ago, but to me it is hard to see what is novel compared to evidence*
*presented e.g. by Ju and Mathieson 2021.*

**Response:** Our results replicate the signal reported in Ju and Mathieson (2021, *PNAS*
<https://doi.org/10.1073/pnas.2009227118>) that selection has acted on skin pigmentation by
favouring a limited subset of large-effect alleles, and we duly cite their paper in the main text.
The novelty of our analysis is in the deconvolution of ancestry, which allows us to trace the
timing of these changes in each of the four ancestral paths leading to present-day
Europeans. We show that selection occurred early on in groups that were moving
northwards and westwards, and only later in the Western hunter-gatherer background after
these groups encountered and admixed with the incoming populations.

*"The selective forces likely favouring ApoE2 in Steppe pastoralists may be associated with*
*protective immune responses against infectious challenges, such as protection against*
*severe childhood malaria or infection with an unknown coronavirus" -why specifically a*
*coronavirus? This seems sensationalistic.*

**Response:** The link between ApoE isoforms and coronaviruses is discussed in
Supplementary Note 2f:

lines 4875-4889 (Supplement):

The impacts of ApoE isoforms on severe acute respiratory virus 2 (SARS-CoV-
2) infection risk, disease progression, and mortality are under investigation.
One study has linked ApoE2 to a decrease in the risk of SARS-CoV-2 infection
but not to the severity of the disease⁶⁰. Several other papers have linked
ApoE4 to an increased risk of infection and more severe disease⁶¹⁻⁶³. In
addition, ApoE4 has been linked to microvascular damage in the brain and
increased neuroinflammation, with some pathways overlapping those activated
in Alzheimer's⁶⁴, suggesting SARS-CoV-2 infection might work as a dementia
disease accelerator, specifically in those suffering from - or predisposed to -
Alzheimer's dementia. No studies appear to have investigated the link between
ApoE isotypes and other human coronaviruses according to PubMed searches
(HCoV-229E, HCoV-OC43, HCoV-HKU1, HCoV-NL63, MERS-CoV, and
SARS-CoV-1). Taken together, these somewhat incomplete results suggest
that it is possible ApoE2 might have reduced the risk of infection with a SARS-
CoV-2-like coronavirus and might thus have been positively selected for in
regions of high endemicity of this (and possibly several) coronaviruses.
However, this suggestion is highly speculative due to the lack of data.

**Changes:** We have updated the main text reference to an "unknown viral infection",
rather than an "unknown coronavirus", as ApoE isoforms have been associated with
multiple infectious diseases.

lines 536-539:

The selective forces likely favouring ApoE2 in Steppe pastoralists may be
associated with protective immune responses against infectious challenges,

such as protection against severe childhood malaria or an unknown viral
infection (Supplementary Note 2f, Supplementary Table S2f.3).

*Overall, I think this paper provides an interesting and valuable analysis of the new*
*substantial data to-be-presented in the Allentoft et al. preprint, but has reduced novelty due*
*to the expansive network of multiple publications.*

**Response:** We thank the reviewer for their constructive feedback and we hope that our
responses have clarified the novelty of our methods, results and conclusions. It is our view
that the network of related publications has helped us present these results in a more
coherent and focused manner, and made them more accessible to a broader audience.

**Referee #4 (Remarks to the Author):**

*The authors present a rigorous analysis of selection on complex traits in Europe over the last*
*12,000 years using ~1,600 imputed ancient genomes. They also describe the role of ancient*
*admixture in shaping the genetic variation underlying complex traits in present-day*
*Europeans. I found the manuscript very interesting and for the most part, clearly written. This*
*is not easy given the limited space. The authors also seem to have done a commendable job*
*of responding to the reviewers' comments constructively and addressing them.*

*In particular, reviewer 3's comment on the weak link between selection hits and trait-*
*associated variants was well-made. As the authors wrote, connecting selection hits to traits*
*is not trivial and remains an unsolved problem in the field. This does not kill the study as long*
*as the results are properly caveated, which they are. The authors have also toned down the*
*language throughout the paper sufficiently so as not to imply that a signal of selection on a*
*variant implies selection on a trait that the variant or some other variant nearby might be*
*associated with.*

**Response:** We thank the reviewer for their positive feedback.

*I do have some minor comments:*

*1. The differences in ancestral risk scores seen in Fig. 6 could yet be due to subtle inflation*
*in effect sizes due to stratification in the original GWAS given that there are ancestry*
*gradients in UKB in the same direction. For example, they found height ARS to be higher for*
*Steppe compared to WHG, which is consistent with people in the north being taller and*
*having more Steppe ancestry compared to people in the south who have more WHG*
*ancestry. This could either be a real signal — that the ancestry gradient actually contributes*
*to the north-south cline in height — or it could be because the GWAS effect sizes are*
*influenced by uncorrected stratification in the UKB. The authors recognize this as well in*
*Supplementary note S2f and I don't necessarily think this is the case. But I wonder, given the*
*strongly worded statement in lines 554, if the authors should also state this caveat in the*
*main text.*

**Changes:** We have added an additional caveat to the main text.

lines 549-550:

However, our results do not preclude the possibility that height has been
selected for in specific populations (Chen et al. 2020; Howe et al. 2022), nor do
they prove that UK Biobank effect sizes are free from uncorrected stratification.

*2. In Figs. 3 and S1c1, please clarify that the numbers along the paths are effective*
*population sizes (presumably).*

**Changes:** We have amended the figure caption to make this clearer.

lines 206-209:

Fig 3. A schematic of the model of population structure in Europe, used to
simulate genomes to train the local ancestry neural network classifier. Moving
down the figure is forwards in time and the population split times and admixture
487 times are given in generations ago. Each branch is labelled with the effective
population size of the population. Coloured lines represent the populations
declared in the simulation that extend through time.

*3. I couldn't find the generation time used to convert the time in generations in Figs. 3 and*
*S1c1 and the time in years in Fig. 4.*

*Presumably between 25-30 years given the scales but good to be explicit.*

**Changes:** We have updated Supplementary Note 2a with the generation time used.

lines 1668-1670:

We converted the calendrical ages of the samples into generations by
assuming a generation time of 28 years (Moorjani et al. 2016).

Reviewer Reports on the Second Revision:

Referees' comments:

Referee #2 (Remarks to the Author):

The authors have added caveats for the majority of the points I raised, and while I don't necessarily share all their preferences, these caveats are satisfactory. I congratulate them on the major joint effort and contribution that this manuscript represents.

My remaining comment is that the admixture proportions outside of Europe from the worldwide CHROMOPAINTER analysis (Figure 2) remain quite extraordinary claims, somewhat disconnected from the rest of the manuscript.

Do the authors indeed claim that the EHG, early Holocene eastern European hunter-gatherers, contributed ~6% of the ancestry in present-day Philippines, and similarly for other countries in East Asia? This seems very important for our understanding of prehistory if true, but should then be confirmed further and put in context of other ancient DNA and modern DNA studies of the regions that suggested simpler models. Or do they think that they do not necessarily imply a direct contribution of these ancient populations due to uncertainties in the source panels and model?

In relation to this point, obtaining confidence intervals by bootstrapping across individuals seems quite clearly incorrect, as it doesn't account for evolutionary variance. Uncertainty for admixture proportions is usually obtained by bootstrapping across chromosomes or loci. I don't necessarily request a new analysis, if indeed the authors do not necessarily believe the admixture proportions to be robust and will clarify this throughout the text and figure.

Author Rebuttals to Second Revision:

Referee #2 (Remarks to the Author):

The authors have added caveats for the majority of the points I raised, and while I don't necessarily share all their preferences, these caveats are satisfactory. I congratulate them on the major joint effort and contribution that this manuscript represents.

My remaining comment is that the admixture proportions outside of Europe from the worldwide CHROMOPAINTER analysis (Figure 2) remain quite extraordinary claims, somewhat disconnected from the rest of the manuscript.

Do the authors indeed claim that the EHG, early Holocene eastern European hunter-gatherers, contributed ~6% of the ancestry in present-day Philippines, and similarly for other countries in East Asia? This seems very important for our understanding of prehistory if true, but should then be confirmed further and put in context of other ancient DNA and modern DNA studies of the regions that suggested simpler models. Or do they think that they do not necessarily imply a direct contribution of these ancient populations due to uncertainties in the source panels and model?

In relation to this point, obtaining confidence intervals by bootstrapping across individuals seems quite clearly incorrect, as it doesn't account for evolutionary variance. Uncertainty for admixture proportions is usually obtained by bootstrapping across chromosomes or loci. I don't necessarily request a new analysis, if indeed the authors do not necessarily believe the admixture proportions to be robust and will clarify this throughout the text and figure.

Response: We have added an additional caveat to the main text to address this issue. It was not our intention to imply that Eastern Hunter Gatherers (EHG) migrated into the Philippines and admixed directly with local hunter gatherer groups there, or anywhere else in East Asia. Our CHROMOPAINTER results are best interpreted as depicting shared genetic affinities between present-day populations and the ancestral source populations used for the local ancestry inference. In East Asia, our ancestral source populations are less directly related to present-day individuals than they are in West Eurasia, and therefore, the results should not be interpreted as literal movements of people. In East Asia, EHG ancestry is the best match among our source populations for a closely related ancestry present across the region in variable quantities.

Changes:

lines 142-147:

We caution, however, that absolute admixture proportions should be interpreted with caution in regions where our ancient source populations are less directly related to present-day individuals, such as in Africa and East Asia. Whilst these values are dependent on the reference samples used, as well as the treatment of pre- or post-admixture drift, the relative geographical variation and associations should remain consistent.